



# Vertical distribution of planktonic foraminifera in the Subtropical South Atlantic: depth hierarchy of controlling factors

Douglas Lessa[1,2], Raphaël Morard[1], Lukas Jonkers[1], Igor M. Venancio[3], Runa Reuter[1], Adrian Baumeister[1], Ana Luiza Albuquerque[2], Michal Kucera[1]

[1] MARUM - Center for Marine Environmental Sciences, University of Bremen, D-28359 Bremen, Germany

[2] Programa de Pós-Graduação em Geoquímica Ambiental, Universidade Federal Fluminense, Niterói, Brazil, 24.020-141

[3] Center for Weather Forecasting and Climate Studies (CPTEC), National Institute for Space Research (INPE), Rodovia Pres. Dutra, km 39, 12.630-000 Cachoeira Paulista, SP, Brazil

*Correspondence to*: Douglas Villela de Oliveira Lessa (dvolessa@id.uff.br)

**Abstract.** Temperature appears to be the best predictor of species composition of planktonic foraminifera communities, making it possible to use their fossil assemblages to reconstruct sea surface temperature (SST)
variation in the past. However, the role of other environmental factors potentially modulating the spatial and vertical distribution of planktonic foraminifera species is poorly understood. This is especially relevant for environmental factors affecting the subsurface habitat. If such factors play a role, changes in the abundance of deeper dwelling species may not solely reflect SST variation. In order to constrain the effect of subsurface parameters on species composition, we here characterize the vertical distribution of living planktonic
foraminifera community across the subtropical South Atlantic Ocean, where SST variability is small but the subsurface water mass structure changes dramatically. Four planktonic foraminifera communities could be identified across the top 700 m of the E-W transect. Gyre and Agulhas Leakage faunas were predominantly composed of *Globigerinoides ruber*, *Globigerinoides tenellus*, *Trilobatus sacculifer*, *Globoturborotalita rubescens*, *Globigerinella calida*, *Tenuitella iota* and *Globigerinita glutinata*, and only differed in terms of
relative abundances (community composition). Upwelling fauna was dominated by *Neogloboquadrina pachyderma*, *Neogloboquadrina incompta*, *Globorotalia crassaformis* and *Globorotalia inflata*. Thermocline fauna was dominated by *Tenuitella fleisheri*, *Globorotalia truncatulinoides* and *Globorotalia scitula* in the western side, and by *G. scitula* in the eastern side of the basin. The largest part of the standing stock was consistently found in the surface layer, but SST was not the main predictor of species composition, neither for
the total fauna at each station nor in analyses considering each depth layer separately. Instead, we identified a consistent vertical pattern in parameters controlling species composition at different depths, in which the parameters appear to reflect different aspects of the pelagic habitat. Whereas productivity appears to dominate in the mixed layer (0-60 m), physical-chemical parameters are important at depth immediately below (60-100 m), followed by parameters related to the degradation of organic matter (100-300 m), and parameters describing the
dissolved oxygen availability (>300 m). These results indicate that the seemingly straightforward relationship between assemblage composition and SST in sedimentary assemblages reflects vertically and seasonally integrated processes that are only indirectly linked to SST. This also implies that fossil assemblages of planktonic foraminifera should also contain a signature of subsurface processes, which could be used for paleoceanographic reconstructions.


Key words: planktonic foraminifera, South Atlantic, vertical distribution, ecology, micropaleontology, plankton multinets

## 1.    Introduction

The composition of planktonic foraminifera communities in the water column changes in response to key
properties of their habitat such as water temperature, salinity and food availability (Bé, 1977; Ottens, 1992; Ufkes et al., 1998; Schiebel and Hemleben, 2017). In contrast, the composition of sedimentary assemblages, integrating the seasonal shell flux across years to centuries consistently appears to be best predicted by





temperature alone (Morey et al, 2005) and this relationship has been exploited by paleoceanographers to reconstruct past temperature (e.g. Kucera et al., 2005). However, the reason for the dominant temperature control on sedimentary assemblage composition remains poorly understood. If species composition primarily reflects other factors, and the covariance of these factors with temperature differed in the past then variability of fossil assemblages may not reflect solely temperature variation, violating a key assumption of the transfer function methods used to convert past assemblage composition to temperature (Telford and Birks, 2005; Juggins et al, 2015). Indeed, data from plankton tows and sediment traps show that the living community likely responds to multiple processes in the water column (Ortiz et al, 1995; Field et al., 2006; Storz et al, 2009; Jonkers and Kucera, 2015). This does not mean that the planktonic foraminifera community does not respond to temperature, but it implies that this relationship may be indirect, mediated by temperature control on productivity or other aspects of the foraminifera habitat. The lack of understanding on how exactly the environment shapes planktonic foraminifera communities is a common issue for the choice of variables used to construct forward models (Žarić et al, 2005; Lombard et al, 2009; Fraile et al, 2009; Kretschmer et al, 2016) as well as transfer function models (e.g. Morey et al, 2005; Siccha et al, 2009; Telford et al., 2013).

Attempts to disentangle the processes that affect planktonic foraminifera community composition must consider the vertical dimension of species habitat. Observations from vertically resolved plankton nets indicate a vertical span of the living community up to 1000 m with the majority of the standing stock concentrated in the upper 300 m (Bé, 1960). Because their habitat stretches across such a large range of depths, communities at different depths likely reflect a vertical hierarchy of controlling processes: light intensity and photosynthesis are only relevant in the photic zone, whereas sinking organic matter budget acts at depth, where degradation activity become important factors leading to oxygen limitation and lower water pH. In such a scenario, there will not be a universal predictor of community composition. Instead, the community will reflect an integrated signal of depth-stratified processes affecting composition in different depth layers. Planktonic foraminifera counts from stratified plankton net samples coupled with observations of physical and chemical parameters in the water column provide the most direct means to tackle this issue. Unfortunately, most previous studies based on foraminifera census counts from plankton nets do not present direct measurements of relevant environmental parameters or lack appropriate depth ranges (Bé, 1960; Ottens, 1992; Kemle-von Mücke and Oberhänsli et al., 1999; Lončarić, 2006; Sousa et al., 2014; Rebotim et al., 2017). In this study, we analyze stratified plankton net samples collected in the subtropical South Atlantic, along a transect covering the gradient from productive coastal waters off South Africa to the oligotrophic waters of the subtropical gyre off Brazil. The transect has been specifically chosen because it samples a region where SST variability is small but the subsurface water mass structure changes dramatically. We make use of the availability of in-situ vertically resolved measurements of environmental parameters and aim to determine which processes explain compositional variability at different depth layers across the transect.

## 2. Material and Methods

The M124 cruise (Karstensen et al., 2016) took place between February 29$^{th}$ to March 16$^{th}$ 2016 on board the RV Meteor, sampling across the subtropical South Atlantic (26 to 34 °S and 18 °E to 36 °W) in an east – west (zonal) transect (Fig. 1a). Planktonic foraminifera were collected at 17 stations using a Multiple Plankton Sampler (MPS, HydroBios, Kiel) with 50 x 50 cm opening, 100 μm mesh size and 5 cod-ends, which allowed sampling five different depths per haul. Two MPS casts were performed at each station, a shallow cast with depth intervals 100–80 m, 80–60 m, 60–40 m, 40–20 m, 20–0 m and a deep cast with depth intervals 700–500 m, 500–300 m, 300–200 m, 200–100 m, 100–0 m. This sampling scheme provides a nine-level resolution of the water column. At the shallow station 192, only a shallow MPS cast was done.

For all deployments, the MPS was slacked with all the nets closed to avoid contamination. The slacking was done at a speed of 0.5 m/s and stopped when the rope length equaled the lowest depth plus 10 to 20 meters to account for the angle of the rope. The MPS was hoisted at a speed of 0.5 m/s and the successive closing/opening depth level were automatically triggered by an in-house software running under MATLAB 2011b based on the absolute depth determined by the pressure sensor of the MPS. Rough sea was encountered at station 265 and the hoist speed was lowered at 0.3 m/s to reduce the tension on the nets. The triggering was activated 2.5 meters before the MPS reached a given depth to account for the time needed for the net to open/close. For 2 MPS deployments at stations 202 and 306, the opening and closing of the nets was done manually because of connectivity problems between the software and the controlling unit of the MPS. After each haul, the nets were carefully rinsed using seawater. The collected plankton was recovered in the cod-ends and brought to the lab and empty cod-ends were mounted on the MPS for the next cast. At the end of each station the MPS was carefully rinsed with soft water and the nets were inspected to ensure that they were not damaged during the deployment.





In addition to the MPS sensors that measure the pressure and activate the opening/closing of the nets, a CTD was mounted on the MPS to measure physical and chemical characteristics of the water column (Temperature, Salinity, Oxygen, Chlorophyll-a, PH) during the cast. The CTD was set on a recording mode to make measurement every second. The CTD was switched on before starting the operation and was running during the
whole station. We also obtained sea surface height (SSH) data from SSALTO/DUACS for each station during the sampling day. SSH data was used to recognize the presence of eddies related to Agulhas Leakage in our stations and possible relationship between planktonic foraminifera species and eddies environment.

The recovered plankton samples were transferred from the cod-ends to glass cups. The cod-ends were carefully rinsed with filtered sea water several times to ensure that all planktonic particles were recovered. After having
recovered each sample, the cod-ends were rinsed thoroughly using freshwater and cleaned in an ultrasonic bath to remove the finest planktonic particles that may have clogged the mesh. The samples were swirled to concentrate the planktonic foraminifera in the middle of the dish and separate them from other zooplankton and organic particles. The planktonic foraminifera were pipetted out on a filter and transferred onto micro paleontological slides with a brush. All small patches of organic matter were also checked to pick exhaustively
all foraminifera. When each cardboard was fully covered with foraminifera, the cardboard slides were air-dried for at least one hour and stored at –80°C. For all stations, except for the station 192, all the samples have been processed during one working day. The samples of the station 192 were only partially processed on board and the plankton residues were placed into sampling bag and frozen. All samples were kept frozen during their transport back to the Bremen University in Germany.

The picked planktonic foraminifera were counted and identified to species level following Schiebel and Hemleben (2017). Living and dead specimens were recognized by their cytoplasm content and considered separately and specimens that showed adult morphology were separated from pre-adult ontogenetic stages. Specimens with cytoplasm in the last whorl were considered as living, whereas tests with none (or almost none) cytoplasm in the last whorl, given by a distinctive white coloration of the test wall, were considered dead. This
distinction is likely overestimating the number of living specimens, because the presence of cytoplasm itself does not guarantee that a specimen was alive during collection. This simplification should result in a slightly deeper estimate of the living depth, caused by dead specimens with cytoplasm being found beneath their original habitat (see Rebotim et al, 2017). The ontogenetic classification of specimens followed the morphological differences between pre-adult and adult stages as summarized by Brummer et al (1986). Species with relatively
small morphological differences among the ontogenetic stages (e.g. *Globigerinella calida*, *Globigerina bulloides*, *Globorotalia scitula* and *Globorotalia truncatulinoides*) were classified as "pre-adult" when their identification was performed at a magnification higher than 100x and surface features typically found in adults (e.g. spines, pustules, large pores) were lacking. In many small sized species (e.g. *Tenuitella fleisheri*, *Turborotalita clarkei*, *Turborotalita quinqueloba*, *Globigerinita glutinata* and *Globigerinoides tenellus*), pre-
adults could not be identified due to the high morphologic similarity among them and pre-adults of big species and such specimens were grouped together in the category "unidentified juveniles". Initial ontogenetic stages of *Globigerinoides ruber* white and *Globigerinoides elongatus* were lumped together as "*G. ruber* juveniles", because the diagnostic trait of the two species is only observed among adult specimens.

The counts were used to calculate concentrations (shells per cubic meter) for each station through
the formula

(1)

$$Ci\ (shells/m^3) = Ni/V$$

Where "$C_i$" is the concentration of the species "i" "$N_i$" is the number of counted specimens for the specie "i" and "$V$" is the filtered volume by the plankton net (in m³), which was obtained by multiplying the haul depth and the multinet opening area.

Concentration values of the "living" category (standing stocks) were used to calculate the average living depth
(ALD) and vertical dispersion (VD) using the following equations proposed by Rebotim et al. (2017) in order to determine the preferential depth habitat and the estimated potential vertical range of species.

$$ALD = \frac{\sum(Ci * Di)}{\sum Ci}$$

(2)



(3)

$$VD = \frac{\sum((ALD - Di) * Ci)}{\sum Ci}$$

Where $Ci$ is the concentration of a specie or the total number of foraminifera (shells.m$^{-3}$) and $Di$ is the middle value of the depth interval $i$. ALD and VD were calculated only for species with at least five counted shells at a station.

In addition to depth habitat calculations, we performed multivariate analysis in order to trace faunal groups and environmental variables, which determined the spatial and vertical distribution of the species and species communities. Faunal groups were distinguished using cluster analysis using the concentration of species transformed to percentages (relative abundance) in each depth level, allowing a joint vertical and zonal analysis across the transect. The cluster analysis was carried out using Bray Curtis distance and unweighted pair-group
average (UPGMA) cluster method. The relationship between living planktonic foraminifera to environmental parameters was determined by a multivariate ordination analysis. We chose the canonical correspondence analysis (CCA), as the data presents a huge standard deviation, suggesting that methods for unimodal distribution are more appropriate (standard deviation > 4, Legendre and Gallagher, 2001). For that, the concentration of species was analyzed together with the CTD data. We performed CCAs with two different data
matrices: (1) grouped by station (no depth separation) and correlated with environmental parameters at the surface as well as data on thermocline and maximum chlorophyll level depth (Table 1), (2) separated by depth layers and correlated with the average value of CTD variables for each depth section. All multivariate analysis was carried out using Past 3.16 software (Hammer et al., 2001).

**3. Oceanographic Conditions**
The upper 700 m in our study area comprises three main oceanographic systems: the Subtropical Gyre (west and east), the Agulhas Leakage and the Benguela Current (Fig. 1). The Western Subtropical Gyre (36 – 16 °W) is characterized by high temperature and salinity and low nutrient concentration. Recirculation of warm waters from the Brazil and South Atlantic currents occur in the Western Subtropical Gyre favoring the accumulation of warm and saline waters resulting in a thick and warm mixed layer (Peterson and Stramma, 1991). In contrast, the
Eastern Subtropical Gyre (16 °W – 5 °E) mixed layer is colder. The Agulhas Leakage sector (5 – 15 °E) is characterized by the occurrence of large eddies formed by the interaction between waters leaked from the Agulhas system and waters of the Subtropical Gyre (Fig. 1b), which could be viewed by higher SSH anomaly variation between stations 227 and 202 (7 °E – 15 °E, Fig. 1c, 4). Along the African coast, tongues of nutrient-rich waters from Benguela upwelling system are entrained into the Agulhas leakage and associated westward-
moving eddies (Stramma and England, 1999), causing elevated productivity far offshore (Fig. 1b). Finally the Benguela Current (17 °E) differs from the others systems by having low temperature, salinity and high productivity due to Benguela coastal upwelling. It receives contribution of warm waters from Agulhas Leakage and cold waters from penetrations of the Subtropical Front (Peterson and Stramma, 1991; Fig. 1c).

The upper 700 m of the South Atlantic comprises three water masses. The first 100 to 200 m of the water column
in the Subtropical Gyre is composed of the Tropical Water (TW) (Stramma and England, 1999). The South Atlantic Central Water (SACW) is found between 200 and 600 m between the permanent thermocline (6 to 20 °C isotherm) and halocline (34.3 to 36.0) (Stramma and England, 1999) and its upwelling is responsible for high productivity off Africa. Finally, the Antarctic Intermediate Water (AAIW) can be observed below the thermocline, between 600 and 1500 m. It has low temperature and salinity, but high nutrient and oxygen
concentrations (Stramma and England, 1999). The higher oxygen content of AAIW distinguishes this water mass from the oxygen-depleted lower part of the SACW. The CTD profiles taken during M134 confirm the occurrence of these three water masses in the first 700 m of the cruise transect (Fig. 2). Salinity above 36 and high surface temperatures (Fig. 2c and 2d) indicated the presence of the Tropical Water (TW) that occupied the first 200 m between stations 394 and 344 (36 – 20 °W) and the first 50 m between the stations 332 and 252 (16
°W – 1 °E). Salinity below 36, slow decrease of temperature and lower oxygen concentration indicate the influence of SACW (Fig. 2c, 2d and 2f), which is particularly thick in the center of the transect (16 °W – 1 °E). Between stations 239 and 192 (3 – 17 ° E) low salinity and oxygen (Fig. 2c and 2f) as well as low temperature indicate a near surface SACW upwelling, mixing with warm surface waters from the Agulhas Leakage. The AAIW was identified by higher oxygen concentration (Fig. 2f) around 500 m at station 394 (36 °W) and about
600 m in the western and far eastern stations. Between stations 320 (11 °W) and 252 (1 °E), the SACW was present down to 700 m.

The surface thermally mixed layer (SML) occupied the first 30 - 40 m of the water column at all stations (Fig. 2c). The SML temperature was higher in the Western Subtropical Gyre, gradually decreasing towards the east.



Below the SML, a sharp temperature decrease is observed until about 100 m, followed by near to constant values until about 200 m, and a slow temperature decrease afterwards. Since the expedition took place during the late summer, the 40 – 100 m sharp decrease of temperature likely represents the seasonal thermocline and the boundary between the mixed layer and the permanent thermocline layers was defined where the slow

temperature decrease starts. At stations where the slow temperature decrease started just below the seasonal thermocline, the mixed layer and permanent thermocline boundary was placed below the seasonal thermocline. The delimitation of water mass boundaries across the studied transect is shown in Fig. S1.

The salinity anticorrelated with the temperature with high values in the mixed layer and a permanent halocline at the depth of the permanent thermocline. The salinity maximum was found at the surface, except in the Western

Subtropical Gyre, where it occurred between 50 and 100 m. The Dissolved Oxygen (DO) was relatively high along the studied transect with value ranging between 5 and 9 ml/l and lower concentrations found only at station 202 (14 °E). In a DO anomaly plot (Fig. 2f); the highest values were usually in the subsurface, especially in the Western Subtropical Gyre. The lowest anomaly values occurred between 200 and 600 m coinciding with the SACW domain. In the lowest part of the water column, higher DO anomalies were observed in the Western

Subtropical Gyre and Agulhas Leakage regions, demonstrating the presence of the AAIW. In the Eastern Subtropical Gyre, low DO anomalies were observed down to the deepest studied layers, indicating that the SACW/AAIW boundary was located deeper than 700 m.

The chlorophyll-*a* concentration was higher (63 to 77 µg.l$^{-1}$) between stations 214 and 192 (11 – 17 °E) with the Deep Chlorophyll Maximum (DCM) at about 50 m. Moderate chlorophyll-*a* concentrations (55 – 65 µg.l$^{-1}$) and a

DCM between 50 and 100 m was seen at stations 265, 229 and 227 (3 – 7 °E). The other stations had lower values ranging from 38 to 45 µg.l$^{-1}$ and a DCM below 100 m. High chlorophyll-*a* values in the east are associated with the Benguela upwelling system (station 192) and its tongues that mix with oligotrophic waters of Agulhas Leakage (stations 214 and 202). On the other hand, low chlorophyll-*a* concentrations in western and central South Atlantic are associated with strong stratification of the Subtropical Gyre. Surface water pH varied

in opposite direction to primary productivity with high values (up to 8.8) on the western side and lower (down to 8.3) on the eastern side. Lower pH values were observed in deeper layers, especially in the east where values ≤ 8.0 were observed below 500 m.

## 4. Results

We identified 38 species of planktonic foraminifera of which 22 species yielded five or more individuals per

station, which allowed the analysis of their habitat depth and the zonal variation (Table 2, Fig. 4). We observed high standing stocks of *Globigrinoides ruber*, *Globigerinoides elongatus*, *Trilobatus sacculifer*, *Globoturborotalita rubescens*, *Globigerinoides tenellus*, *Tenuitella iota* and *Globigerinita glutinata* in the mixed layer of the Subtropical Gyre and Agulhas Leakage region. In the Benguela Current and the subsurface sector of the Agulhas Leakage area, we observed high abundances of *Neogloboquadrina dutertrei*, *Neogloboquadrina*

*pachyderma*, *Neogloboquadrina incompta*, rounded specimens of *Globorotalia crassaformis* and *Globorotalia inflata*. In contrast, the water column below 100 m (permanent thermocline layer) in the whole transect had high abundances of *Tenuitella fleisheri*, *Globorotalia truncatulinoides* (left coiling) and *Globorotalia scitula* and *Hastigerina pelagica*. Apart from the Benguela and permanent thermocline dwelling species, some other species also had restricted distributions. The pink morphotype of *G. ruber* was found only in the Western Subtropical

Gyre, *Candeina nitida* occurred between the stations 356 (Western Subtropical Gyre) and 265 (Eastern Subtropical Gyre), *Globorotalia menardii* occurred in high abundances in the first 100 m at the three western stations (394, 382 and 370), and conical specimens of *G. crassaformis* were observed rarely in the thermocline of the Western Subtropical Gyre.

The total concentration of planktonic foraminifera was higher at stations 332, 344, 356 and 370 with values up to

12 shells.m$^{-3}$. However, if we consider concentration values in the upper 100 m, high concentrations occurred only in the western stations (Fig. 3a), with up to 75 shells.m$^{-3}$ in the surface layer of station 370 compared to 25 shells.m$^{-3}$ in the surface layer of the station 192. Concentrations of living specimens were always higher in the upper 100 m of the water column (Fig. S2), with a gradual increase in the proportion of dead (empty shells) specimens below 100 m and almost no living specimens below 500 m. The lowest concentrations (living and

dead specimens) were observed in the deepest sections of stations 394, 370 and 227 that were under influence of AAIW. Pre-adults specimens were abundant in central gyre stations (Fig. 3), where station 306 (11 °W) had almost 100 % neanic and juvenile specimens in the upper 40 m (Fig. S3). The number of adults tended to increase downward (> 40 m depths) in relation to pre-adults that were virtually absent below 100 m depth. The exception was *Globorotalia crassaformis* in the eastern stations. The concentrations of adults were higher than





those of pre-adults (near 100 % adult) at stations 214 and 192, where most of the fauna was composed of cold-water species.

The species *G. ruber* (pink and white), *T. sacculifer*, *Orbulina universa* and *Globigerinoides conglobatus* had an ALD between 20 and 30 m with a VD up to 16 m (Fig. 5) and these values showed little zonal variation (Fig. 4).

*G. ruber* white and *T. sacculifer* were present at all stations, the other two species were only found on the western side of the Subtropical Gyre, where temperature and salinity values were high. Most of the specimens of *G. conglobatus* were pre-adult. The ALDs of *C. nitida*, *G. rubescens*, *T. iota*, *G. menardii*, *G. calida*, *G. tenellus*, and *G. glutinata* were between 30 and 40 m and the VD between 20 and 30 m (Fig. 5). We observed more zonal variation in both ALD and VD for these species. Most of *G. calida* individuals found at stations 332 and 320

were pre-adult and they inhabited the upper 40 m, contrasting with others stations where the ALD varied between 40 and 70 m (Fig. 4). Specimens of *G. menardii* were identified in stations of the Western Subtropical Gyre and the Agulhas Leakage area, but the low abundance in the latter area did not allow reliable ALD and VD estimates. The species *C. nitida* was most abundant in the Eastern Subtropical Gyre stations (station 320 to 265), whereas *G. rubescens*, *G. tenellus*, *T. iota* and *G. glutinata* were found at similar abundances along the entire

transect.

The ALD of species *G. elongatus*, *G. crassaformis*, *N. dutertrei*, *N. incompta* and *N. pachyderma* was between 50 and 70 m with a VD between 25 and 50 m (Table 2, Fig. 4). This depth range corresponds to the seasonal thermocline. Specimens of *G. elongatus* occurred at higher abundances in the Western Subtropical Gyre and at stations 320 and 306 of the Eastern Subtropical Gyre, where their ALD was largest (between 80 and 90 m, Fig.

4). The remaining four species were more abundant in the Benguela Current and subsurface waters of the Agulhas Leakage, which are associated with upwelling-induced productivity. However, specimens of *G. crassaformis* found on the eastern side of the transect were usually pre-adult (small size, about 4½ to 5 chambers in the last whorl and arched aperture similar to *Globorotalia inflata*, see appendix 1 and plate 3). Typical adult specimens (conical equatorial view, four chambers in the last whorl and a low arched aperture with an

imperforate lip, see appendix 1 and plate 4) were found in the thermocline layer in the Western Subtropical Gyre at insufficient abundance to calculate a reliable ALD.

The ALD of *Tenuitella fleisheri*, *Hastigerina pelagica*, *Globorotalia inflata*, *Globorotalia truncatulinoides* and *Globorotalia scitula* ranged between 95 and 250 m with VDs between 36 and 110 m (Table 2, Fig. 4). Specimens of *G. inflata* were present in small numbers only in the thermocline layer of stations 192, 202 and

214 (Benguela and Agulhas Leakage). Specimens of *H. pelagica* were encountered below 200 m in the entire Subtropical Gyre and large living specimens were also present in the surface layer of station 278 (East Subtropical Gyre). However, the number of empty tests was usually high for this species, and only five stations had enough numbers of living specimens to calculate a reliable ALD. Specimens of *G. truncatulinoides* were most abundant inside the Subtropical Gyre (stations 394 to 265) and were usually sinistral. Dextral coiling *G.*

*truncatulinoides* were very rare, which did not allow distinguishing the ALDs of the two coiling variants. *T. fleisheri* and *G. scitula* were abundant at most stations, but the highest concentrations tended to differ zonally and vertically. Specimens of *T. fleisheri* were more abundant between 100 and 300 m in the Western Subtropical Gyre. Specimens of *G. scitula* were more abundant in the East Subtropical Gyre and Agulhas Leakage with ALD usually below 200 m. At stations under influence of Benguela upwelling (stations 202 and 192), *G. scitula* and

other thermocline dwelling species were encountered in shallower waters, but for *T. fleisheri* and *G. truncatulinoides*, the number of individuals was insufficient to estimate ALD.

The community variation across the Subtropical South Atlantic was analyzed by cluster and ordination multivariate analyzes. The cluster analysis of the community composition (relative abundance without cut level) for each depth and station revealed the presence of twelve clusters, composed of seven principal faunas,

consistently separated by region and depth (Fig. 7). These were the Subtropical Gyre fauna that was further divided in surface, western subsurface and eastern subsurface; Agulhas Leakage fauna; Thermocline fauna that was further divided in western and eastern; and Benguela fauna. The average relative abundances for the most important species in each cluster are summarized on Table 3. This reveals that the warm and oligotrophic areas are inhabited by the same species, but that their proportions vary. Thus, the subdivision of the Subtropical Gyre

fauna reflects increased abundance of *G. rubescens* and *G. tenellus* in the west and *G. glutinata* in the east and the Agulhas Leakage fauna is characterized by a higher contribution of *T. sacculifer*. In contrast, Thermocline and Benguela communities comprised distinctly different species assemblages. Thermocline fauna is characterized by *G. scitula*, *T. fleisheri* and *G. truncatulinoides*, whereas Benguela upwelling fauna is characterized by *G. crassaformis* and *N. pachyderma*. In contrast to the Subtropical Gyre stations, where surface

and subsurface faunas were recognized, the Agulhas Leakage lower mixed layer was occupied by the upwelling





fauna (Fig. 7b), which could be associated to high levels of chlorophyll-*a* (productivity) due to upwelling filaments (Fig. 2). The deepest samples at stations 394, 370 and 227 had too little or no living planktonic foraminifera and, therefore, could not be clustered.

To identify which variables or processes were associated with the observed planktonic foraminifera distribution
across the subtropical South Atlantic, we performed canonical correspondence analyses (CCA) in two ways: (1) a station separated analysis comprising species concentrations and environmental parameters (temperature, salinity, dissolved oxygen and pH at 30 and 200 m of depth, and total chlorophyll-*a* concentration) with collapsed depth sections (Fig. 8); and (2) station separated analysis of species concentrations and environmental parameters for each single depth interval (Fig. 9). The first, depth-integrated, CCA shows that most of the
variability in species composition can be explained with two CCA axes (total inertia explained 85 %). Most species are oriented along a productivity (chlorophyll-*a*) gradient, with the Benguela upwelling species on the high productivity end and the warm surface and gyre subsurface groups on the other; *G. scitula* and *G. ruber* pink, *T. iota* plot on opposite ends of the second axis (Fig. 8). Such an important role of productivity appears to contrast with global studies that have documented temperature as the most important predictor of foraminifera
assemblages (refs Moery etc). However, in our area SST and chlorophyll-*a* (productivity) are anticorrelated, since low temperatures mean less stratification, which causes enhanced nutrient availability at the photic zone. Moreover, temperature influences respiration and growth rates, rendering the influence of temperature and productivity difficult to separate. Due to the distinct foraminiferal community in the Benguela upwelling system, station 192 stands out in relation to the other stations and hence may be responsible for a large proportion of the
variability. If the station 192 is removed from the analysis (Fig. S4), the mentioned anti-correlation between SST and productivity is still evident, influenced by stations in the Agulhas Leakage domain that were also influenced by cold and productive waters below 40 m during the sampling time. However, the explained variance of the first axis decreases to 41%. Thus, the first principal component axis (Fig. 8) can be linked to productivity modulated by weak water stratification or upwelling.

The second axis, which explains 15% of the observed variance (Fig. 8), separates species with distinct depth habitats and temperature, salinity and pH at the thermocline layer. Most of explanation for the second factor is influenced by *G. scitula*, which had the deepest ALD (234 m). In this study, *G. scitula* was significantly abundant in the whole transect with its highest concentrations occurring at stations with a higher and shallower chlorophyll-*a* maximum (Fig. 5). On the opposite side of the axis, *G. ruber* pink was classified as upper mixed
layer dwelling (Fig. 5) and *T. iota* was classified as a whole mixed layer dwelling in this study with ALD down to 37 m. Both *G. ruber pink* and *T. iota* concentrations were higher in highly stratified areas with low chlorophyll-*a* concentration. The scatter suggests that living depth can be linked to this axis and that different vertical patterns of environmental parameters play a very important role in determining the foraminiferal community. That vertical variation of the community can be linked to organic matter budget from mixed layer to
permanent thermocline , whose amount depends to local productivity. High sinking organic matter budget increases the microbial respiration rate that has influence over pH at thermocline. These different vertical patterns of environmental parameters play a very important role in determining the foraminiferal community in both surface and thermocline layers.

Considering the importance of vertical gradients in environmental variables for the distribution of planktonic
foraminifera in the subtropical South Atlantic, CCA(s) for individual depth intervals were performed (Fig. 9 and Table S1). Those CCA showed four hierarchical changes of processes that define planktonic foraminifera community composition. For the first three depth sections (0 – 20, 20 – 40 and 40 – 60 m), chlorophyll-*a* was the most important variable explaining between 40 and 70 % of the variation, indicating that productivity is the most important factor in the subtropical South Atlantic. Physical-chemical parameters (temperature, salinity and pH)
explained most of the variation of the community in the 60 – 80 and 80 – 100 m depth intervals (60 and 40 % of inertia explained, respectively). For the 100 – 200 and 200 – 300 m depth sections, the community was predominantly explained by pH variations with a low, but downward increasing, contribution of dissolved oxygen (50 and 60 %, respectively).  This suggests that the species distribution was influenced by degradation of organic matter produced in the surface layer since microbial respiration contributes to pH and dissolved oxygen
variability in mesopelagic waters. It is interesting to note that the secondary contributor variables change from temperature and salinity at 100 – 200 m section to dissolved oxygen at 200 – 300 m section. This last environmental parameter reached the highest contribution at 300 – 500 m (50 % of the variation), which could be linked to the oxygen minimum layer. Even though chlorophyll-*a* appears as the most important variable at the deepest interval (500 – 700 m), species standing stocks were too low to define a significant pattern in the





community distribution. Besides, the chlorophyll-*a* concentration as a main controlling factor at this depth seems unrealistic, since photosynthetic activity is greatly reduced below 200 meters.

## 5. Discussion
### 5.1. Synchronized reproduction and ontogenetic vertical migration

Before interpreting the vertical distribution of planktonic foraminifera in the water column as a function of changing environmental properties across the studied transect, it is important to evaluate the possible effect of reproductive processes on species concentration and living depth. The M124 cruise lasted for 16 days with new moon occurring on the 10th day of the cruise and the preceding full moon occurring 7 days before the sampling at the first station. If some of the species reproduced consistently in phase with full moon (Spindler et al., 1979; Jonkers et al., 2015; Venancio et al, 2016) then the proportion of pre-adult specimens should have been higher during the first half of the cruise. Three species show elevated abundance of pre-adults consistent with such pattern: *G. ruber* white, *O. universa* and *G. calida* (Fig. S3). The remaining species show an even proportion of pre-adults throughout the cruise, which is not consistent with synchronized reproduction. For the three species that may have reproduced at full moon, we evaluated by periodic regression analysis whether the living depth of the populations could show signs of an ontogenetic vertical migration – a systematic change in depth habitat with progressive maturity of the reproductive cohort. We observe neither that their ALD is correlated with the proportion of pre-adults nor that there is any systematic change of ALD of these species during the lunar cycle. Taken together, the relatively constant proportions of pre-adult and adult specimens in the majority of the analyzed species speak against a strict synchronization of their reproduction, and even in the three species where the proportion of pre-adults was higher during one part of the lunar cycle, there was no evidence for ontogenetic vertical migration. In the absence of a strong evidence for the control of reproductive or ontogenetic processes on the vertical habitat, and ruling out an effect of daily vertical migration based on detailed observations elsewhere (Meilland et al., 2019), we proceed by interpreting the ALD of individual species and analyzing if the ALD varied predictably as a function of environmental parameters.

### 5.2. Species vertical distribution across the Subtropical South Atlantic

The ALD of planktonic foraminifera species shows a consistent depth ranking pattern and a mixture of stable and variable depth habitats (Fig. 5). Since the main feature of the water column relevant for the vertical distribution of non-motile plankton is the mixed-layer depth, we consider the observed ALD against this reference (Fig. 6). First, we observe a group of species whose ALD was consistently in the **Upper Mixed Layer (UML).** Species in this group had an ALD shallower than 40 m and a low VD that usually was below 40 m. This depth range corresponds to the extent of the warm thermally mixed surface layer (Fig 2). The most abundant species in this group were *G. ruber* (pink and white), *G. conglobatus*, *O. universa* and *T. sacculifer*. These species all bear algal endosymbionts (Takagi et al., 2019) and their observed habitat is consistent with the high light and low nutrient conditions in the SML. The shallow depth habitat of *G. ruber* was also observed by Berger (1969) and Rebotim et al(2017), but the observed ALD for *O. universa* and *G. conglobatus* and partly also for *T. sacculifer* is shallower than in many other studies/regions, indicating that globally this species has a more variable habitat than observed in the studied section. Clearly, the dominant habitats of these species are not reaching below the seasonal thermocline, indicating that they may be thermally constrained to the uppermost summer mixed layer, as light is not limited. A second group of species also inhabits preferentially the upper water layer (ALD usually above 50 m), but their VD (dispersion) indicates that their vertical habitat comprises the **Whole Mixed Layer (WML).** The most abundant species in this group are *C. nitida*, *G. glutinata*, *G. calida*, *T. iota*, *G. rubescens*, *G. tenellus* and *G. menardii*. With the exception of *T. iota*, these species also bear algal endosymbionts (Takagi et al., 2019), which is consistent with their habitat within the photic zone, but they appear less tightly linked to the surface layer, implying either a broader thermal tolerance or adaptation of their symbionts to lower light levels. Whereas for most species the observed habitat is comparable with previous work (Rebotim et al., 2017, Kemle-von-Mücke and Oberhänsli, 1999), the shallow habitat of *T. iota* is at odds with its concentration maximum around 300 m in the NE Atlantic reported by Rebotim et al. (2017). Clearly, the ecology of this species requires further investigation. For the remaining species of the UML and WML groups, it is clear that their habitat is above the deep chlorophyll maximum, suggesting that their vertical distribution is influenced by other parameters than the availability of fresh phytoplankton.

In contrast, species of the **Lower Mixed Layer (LML)** have a habitat that is still dominantly within the seasonal thermocline (above the permanent thermocline), but whose vertical distribution overlaps with the deep



chlorophyll maximum (Fig. 2). The species in this group had an ALD between 50 and 100 m and a VD similar to the UML group. The most abundant species in this group were *G. elongatus*, *N. dutertrei*, *N. pachyderma*, *N. incompta*, *G. crassaformis* and *G. inflata*. With exception of *G. elongatus*, these species occurred in the Eastern Subtropical South Atlantic, indicating that this group may respond to variations in productivity, since here the

chlorophyll concentration was higher, DCM shallower and the water column less stratified than on the western side of the gyre (Fig. 2). This is consistent with the majority of these species being non-symbiotic (Takagi et al., 2019). At station 192, which is influenced by Benguela upwelling, the planktonic foraminifera community was dominated by the LML group at all depths. The species *G. elongatus* was the only LML classified species in the Subtropical Gyre. This species was the most abundant form of the *Globigerinoides* plexus, but we caution

against a too strict interpretation of its apparently substantially deeper habitat than the remaining *G. ruber* morph types because these species can only be distinguished in their adult stages (Aurahs et al., 2011). Nevertheless, the observed depth stratification between *G. elongatus* and *G. ruber* (white) is consistent in sign with previous studies based on observations in the plankton (Kuroyanagi and Kawahata, 2004) and oxygen isotope and Mg/Ca ratios (Wang, 2000, Steinke et al., 2005). Thus, at least in summer, adult *G. elongatus* live below the warm and

stable SML in the South Atlantic.

A distinctly different ALD distribution, with the largest part of the population living below the mixed layer was shown by species of the **Thermocline** group. Species in this group have a variable ALD within the permanent thermocline (below 100 m) and do not show a clear relationship with the position of the DCM. In fact, most of their populations occur often below the DCM. The most abundant species belonging to this group are *T.*

*fleisheri*, *G. truncatulinoides*, *G. scitula* and *H. pelagica*. Thermocline dwelling species represent a clearly defined cluster distinct from other species assemblages (Fig. 6), suggesting that they constitute a distinct community that may not be shaped by surface processes. Since within their habitat photosynthesis is inhibited due to insufficient light, thermocline species are likely to feed on either zooplankton or sinking organic matter (Schiebel and Hemleben, 2017). Thus, species of the thermocline group may respond not only to productivity

that supplies food, but also to processes related to organic matter degradation below the DCM, as dissolved oxygen concentration and pH are influenced depending on the organic matter budget. Little is known about the ecological preferences of *T. fleisheri* because this species is small and hence often overlooked since most studies only investigate the specimens >150 μm. Rebotim et al (2017) classified *T. fleisheri* as a surface to subsurface dweller. However, the species was rare in their study, rendering this classification uncertain. In our, transect, *T.*

*fleisheri* was abundant between 100 and 300 m at all Subtropical Gyre stations with specimens up to 200 μm in size (see plate 5, 11-13). Within the Subtropical Gyre, the ALD of *T. fleisheri* was close to the DCM in ten out of 13 stations within the subtropical Gyre, suggesting a link between depth habitat and the depth of the chlorophyll maximum for this species (Fig. 5).

The ALD of *G. scitula* was the deepest (235 m) and most variable (VD 109 m) in our study. The mesopelagic

habitat of *G. scitula* has also been observed in others regions such as Atlantic and Indian Oceans (Bé and Tolderlund, 1971; Rebotim et al, 2017), NE Pacific off California (Field, 2004) and Western North Pacific off Japan (Itou et al., 2001). This demonstrates that the species is a clear thermocline dweller living below the deep chlorophyll maximum, where it must feed on sinking organic matter. The species *G. truncatulinoides* had an ALD near 170 m with a relatively low VD (55 m), which associates it to the DCM and permanent thermocline.

The cytoplasm color of *G. truncatulinoides* was similar to *G. scitula* (see plates 4 and 5 in the supplementary material) indicating a similar diet. The spatial distribution of the two species shows that *G. truncatulinoides* replaces *G. scitula* towards the west at stations in the Subtropical Gyre. Dissolved oxygen and pH were the most zonally variable environmental parameters within the permanent thermocline layer (Fig. 2). Consequently, the zonal distinction of the two species may be due to variations of processes expressed in these two environmental

parameters. The habitat of *H. pelagica* is dominantly subsurface but highly variable, likely reflecting the presence of both the surface and subsurface genetic types of this species (Weiner et al., 2012).

**5.3. Depth hierarchy of relationships between planktonic foraminifera and environmental parameters**

Plankton net samples represent a snapshot of the plankton state in an exact place and during an exact time. In this way they allow us to observe the direct response of the plankton community to environmental parameters. This

is fundamentally different from relationships extracted from sedimentary assemblages, which represent long-term (years to millennia) integrated fluxes of species (Jonkers et al., 2019). Since temperature consistently appears to be the single and dominant parameter explaining variation in community composition (Morey et al., 2015), our results can be interpreted as evidence for the effect of temperature on sedimentary assemblages being the result of seasonal superposition of assemblages driven by a more diverse set of abiotic and biotic parameters,

varying seasonally and, as our analysis shows, also with depth.





A consideration of habitat depth and their variability among species (Fig. 5,6) as well as the distribution of the species along the transect (Fig. 7) provides clear hints for the effect of multiple environmental parameters on the habitat and the abundance of the species. This observation is reinforced by the results of the CCA (Fig. 8). Since the foraminiferal communities showed a stronger vertical than horizontal pattern and many of the considered environmental parameters only varied at certain depths, we carried out a series of CCAs separately for each of the nine depth intervals (Fig. 9). These analyses reveal the presence of a vertical succession of environmental parameters driving community composition, indicating that the vertical changes in the community composition shown by the cluster analysis (Fig. 7) may be the consequence of vertically varying influence of different environmental processes. This is because the vertical succession of communities matches with changes of most important environmental parameters.

Because of its vertically stacked design, the analysis in Fig. 9 must not be interpreted as an explanation for the vertical succession of species habitats (Figs 5, 6). Instead, it explains which variable controls the species composition within each depth interval. For example, the UML fauna inhabits the surface likely because of its affinity to high light and high temperature, but variations in the composition of the communities in this layer across the studied transect seem to be driven by chlorophyll concentration at the surface. This may hint at differences in the adaptation of the UML species with respect to productivity. Remarkably, below 60 m, i.e. at the depth where the WML and LML fauna occur, temperature, salinity and pH explained most of the variation of the community within the layer. This likely reflects the effect of the thickness of the seasonal mixed layer, which is most expressed at those depths (Fig. 2). Below 100 m, the communities appear to be most influenced by the rate of degradation of organic matter produced in the mixed layer, which is reflected by pH and dissolved oxygen concentration. This is logical, since these communities are dominated by asymbiotic species that live below the permanent thermocline, where temperature variations are subdued. We note that the distribution of these species appeared not to be linked to the deep chlorophyll maximum and that variability in chlorophyll-*a* concentrations do not explain much of the variability in species composition at those depths. Dietary preferences of these species are poorly known, although indirect observations on *Neogloboquadrina* indicate an affinity to marine snow (Fehrenbacher et al, 2018). Our observations also suggest that, rather than feeding on fresh organic matter, the species at these depths feed on degraded organic matter. Below 300 m, oxygen concentration shows highest explanatory power (50 % of variation), which could be linked to direct effect of low oxygen on metabolic processes. Under such scenarios, the small and flat *G. scitula* should be better adapted to oxygen limitation than *G. truncatulinoides*, which is consistent with the observed distribution of these thermocline dwelling species (Fig. 4).

The existence of a vertical succession of parameters best explaining community composition of planktonic foraminifera implies that ecological models derived from sedimentary assemblages integrate seasonally and vertically separate assemblages, whose composition is driven by different processes. It means that SST reconstructions using census counts of planktonic foraminifera (e.g. transfer functions) may not directly reflect temperature, even though statistically temperature appears the most important explanatory variable of sedimentary species composition (Telford et al ., 2013). Theoretically, it should thus be possible to make use of the information hidden in the seasonally and vertically integrated assemblages to obtain information on the state of the vertically acting processes in the past. For example, the progressive replacement of *G. scitula* by *G. truncatulinoides* along the transect is clearly not driven by temperature and if this replacement is preserved across the seasonal cycle, it could represent a powerful proxy for organic matter degradation below the surface.

## 6. Conclusions

We investigated the zonal and vertical distribution of planktonic foraminifera at 14 stations across the subtropical South Atlantic region during the late austral summer 2014. Using environmental data from, CTD profiles and satellite observations, we accessed which factors drive the observed foraminifer species distribution:

– Species specific standing stocks varied regionally, with the most pronounced differences among communities observed between the Benguela (station 192) and Subtropical Gyre (stations from 394 to 239) and with an intermediate community in the Agulhas Leakage region (stations from 227 to 202).
– The highest standing stock was observed in the upper 60 m of water column, whereas the numbers of dead specimens (no cytoplasm) increased below 100 m. The highest concentrations of planktonic foraminifera occurred at stations in the oligotrophic western Subtropical Gyre, indicating that the total standing stock is not positively correlated with productivity during the summer.
– The species *G. ruber*, *G. calida* and *O. universa* had a high number of pre-adults consistent with reproductive cycle at full moon. However, the average living depth (ALS) of those species did not



show a significant lunar periodicity, suggesting that environmental factors are the prime drivers of their depth habitat variability.

- The permanent thermocline layer (200 – 700 m) has a planktonic foraminifera community distinct from the mixed layer. In the western South Atlantic, high abundances of *G. truncatulinoides*, *T. fleisheri* and *G. scitula* were observed, whereas *G. scitula* dominated the community in the eastern South Atlantic. Zonal differences in the mixed layer communities were less pronounced.

- The zonal distribution of species was primarily affected by the inverse relationship between chlorophyll-*a* and water temperature and secondarily by the amount of exported organic matter from the mixed layer to thermocline that reflects specially pH differences between western and eastern South Atlantic within the thermocline layer.

- The vertical distribution of planktonic foraminifera showed a clear depth-dependent hierarchy in the environmental parameters explaining abundance variability. The variability in the upper 40 m was mainly influenced by chlorophyll-*a* concentration (productivity), followed by physic-chemical variables (temperature, salinity and pH) between 40 and 100 m. Below 100 m, variables related to microbial respiration of sinking organic matter were the most important to determine species distribution with pH influencing between 100 and 300 m and dissolved oxygen between 200 and 500 m.

Overall planktonic foraminifera communities of subtropical South Atlantic seem to respond to a horizontally and vertically variable combination of environmental parameters. This should be taken in account when interpreting sedimentary assemblages for paleoceanography.

### 7. Data availability

The foraminifera concentration data and the accompanying environmental data from CTD casts have been deposited on PANGAEA (www.pangaea.de).

*Competing interests*: The authors declare that they have no conflict of interest.

### 8. Acknowledgements

We thank all crew members and scientists for their help in the collection of planktonic foraminifera during the M124 cruise onboard the FS Meteor. This study was financed in part by the Coordenação de Aperfeiçoamento de Pessoal de Nível Superior - CAPES/Brazil - Finance Code 001. Douglas V. O. Lessa acknowledges financial support from CAPES/Paleoceano project (23038.001417/2914-71), the CAPES Programas Estratégicos - DRI (PE 99999.000042/2017-00), and through the Cluster of Excellence "The Ocean Floor – Earth's Uncharted Interface".

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

**Figures**

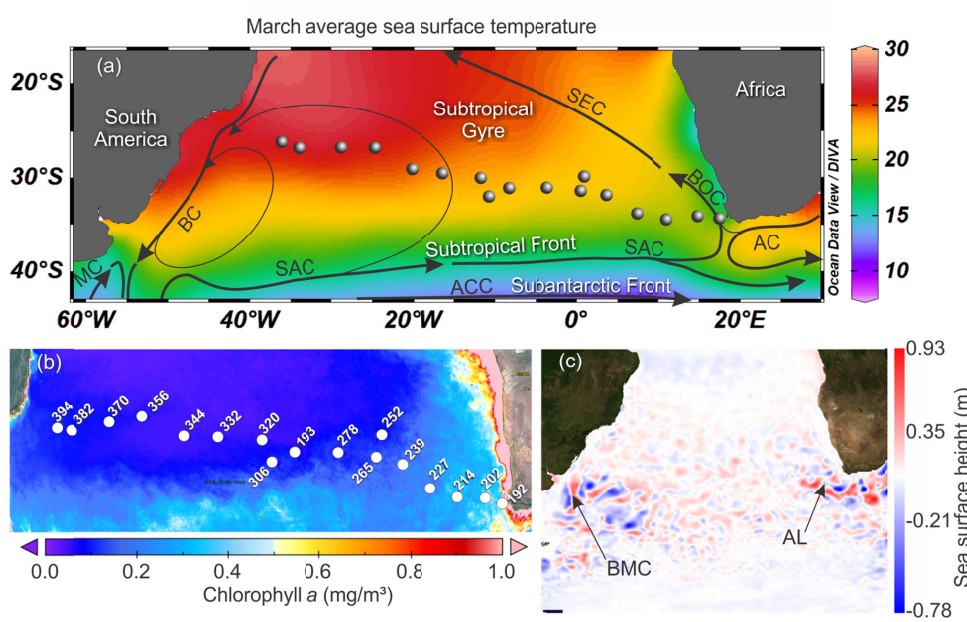

**Figure 1. Map of the South Atlantic Ocean showing the locations of the stations of the M124 cruise. (a) March sea surface temperature (SST) from World Ocean Atlas 2013 and the main surface current systems (modified from World Ocean Atlas 2013 and Stramma and England, 1999). (b) Average 2002 – 2018 of surface chlorophyll-*a* measured by satellite from MODIS – AQUA. (c) Sea surface height above the sea level on March 1st 2016 (sampling of station 202) from Ssalto/Duacs. Acronyms: SAC - South Atlantic Current ACC - Antarctic Circumpolar Current; SEC - South Equatorial Current; BOC - Benguela Oceanic Current; AC - Agulhas Current; BC - Brazil Current; MC –Malvinas Current; AL - Agulhas Leakage; BMC – Brazil – Malvinas Confluence zone.**

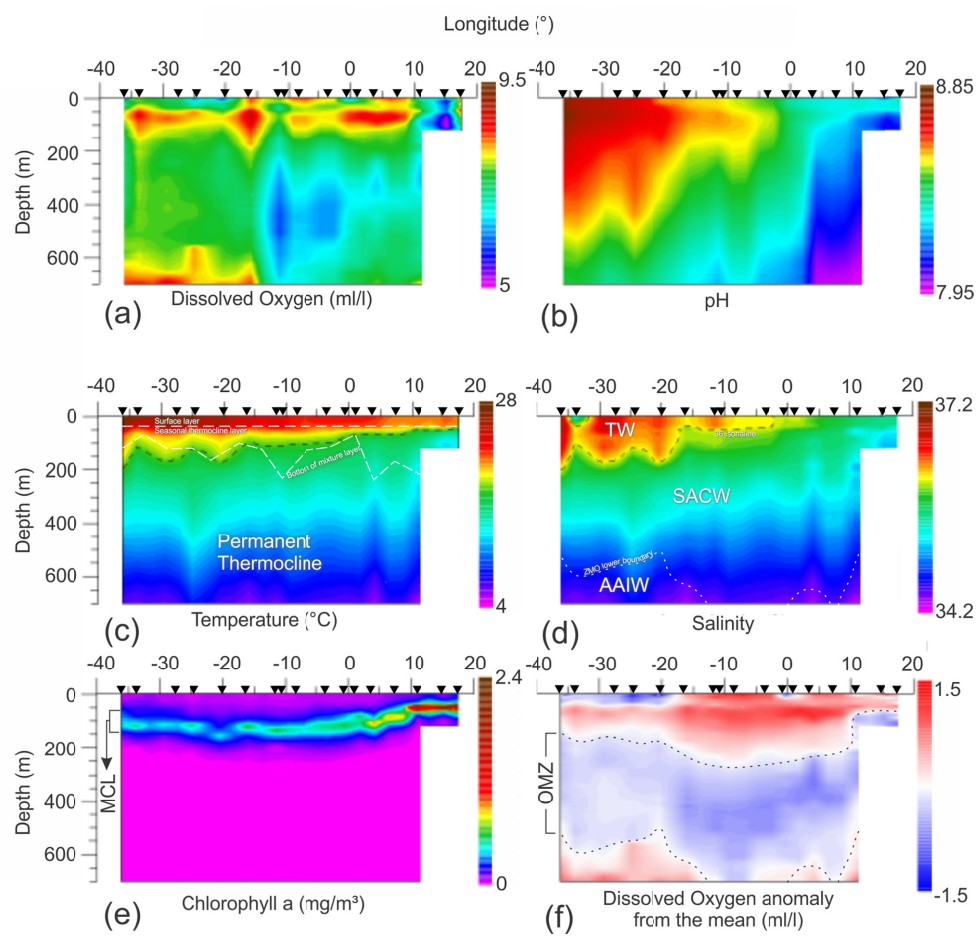

**Figure 2. Gridded variation measured in environmental parameters for the upper 700 m of water column along the transect. Triangles indicate the location of sampling sites. Panels (a) – (e) show raw values of all measured parameters and the panel (f) shows the dissolved oxygen anomaly related to each station mean. TW = Tropical Water, SACW = South Atlantic Central Water, AAIW = Antarctic intermediate water, OMZ = Oxygen Minimum Zone.**



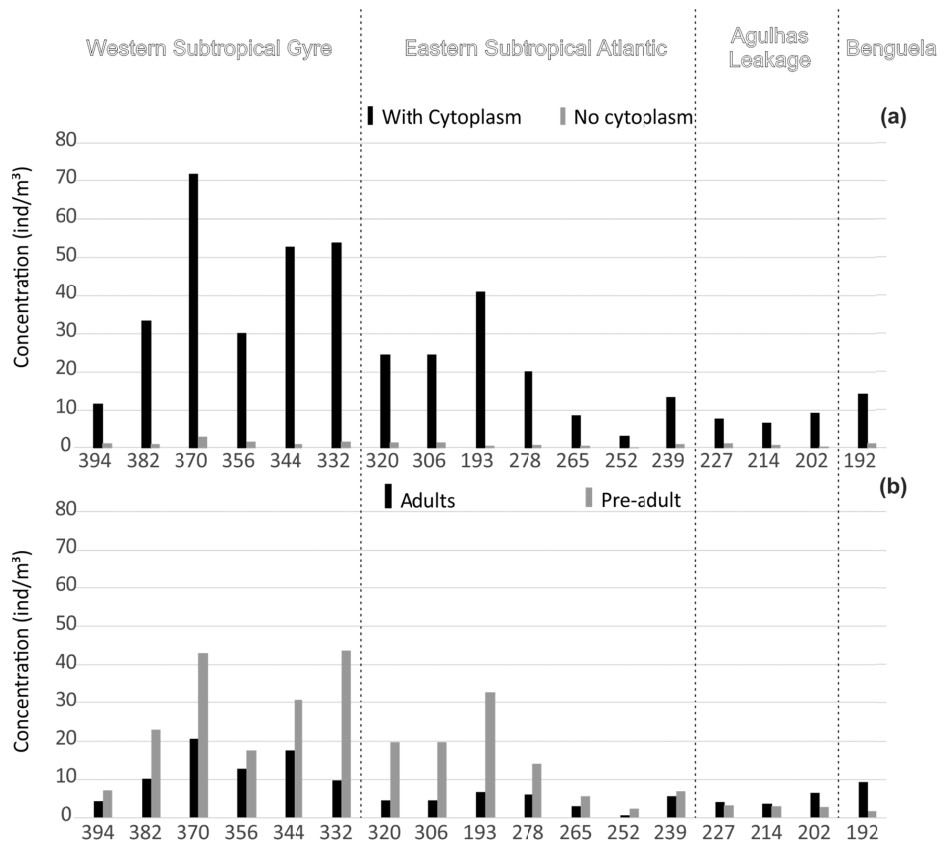

Figure 3. Concentration of planktonic foraminifera in the upper 100 m. (a) Total concentration divided by the presence of cytoplasm. (b) Total standing stock separated by ontogenetic stage. Please, check the supplement material (Figure S2 and S3) for the entire profile.





**Figure 4.** Variation of the depth habitat and standing stock (integrated over the upper 700 m) of main species across the subtropical South Atlantic. Species data is compared with temperature in °C (color filling), chlorophyll-*a* with thin lines delimiting layers with 0.2 mg.m-3 and thick line indicating the chlorophyll-*a* apex, sea surface height (SSH) and boundaries of surface mixed layer and mixed/permanent thermocline layers (white continuous lines). Note that panels have different depth scales.



**Figure 4. Continued.**



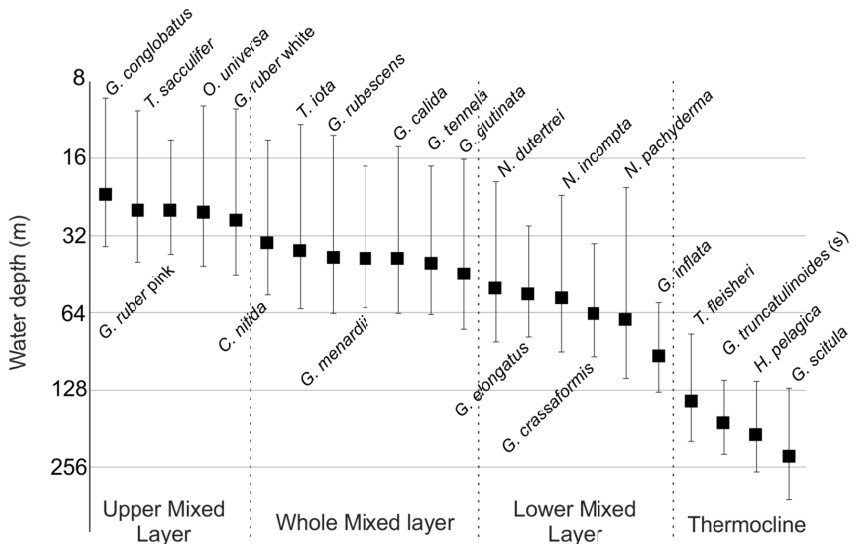

**Figure 5.** Average Living Depth (ALD) and Vertical dispersion (VD) of the most abundant planktonic foraminifera species sorted by depth (in base 2 logarithmic scale) with the defined depth groups.

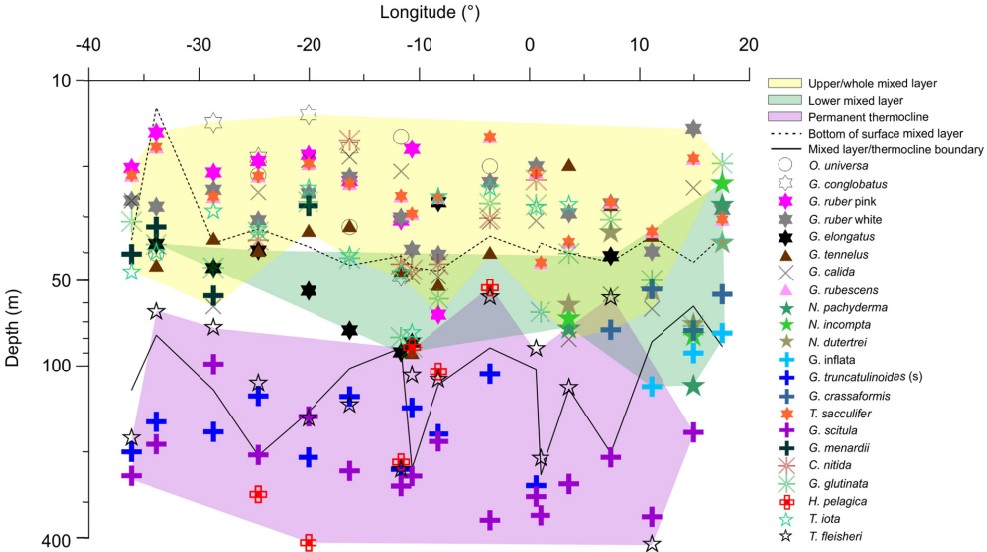

5      **Figure 6.** Average Living Depth (ALD) of the most abundant species along the transect in the South Atlantic. Depth groups are highlighted by different color filling. Note the ALD separation above and beneath 100 m and the lower mixed layer group that is more evident on the eastern side.

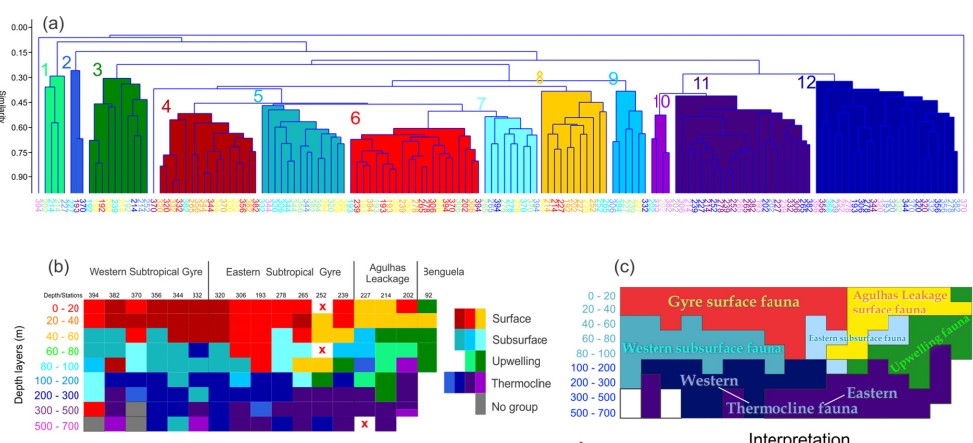

**Figure 7. (a) Cluster analysis of relative community composition in all stations and depths sections. (b) Graphical representation of clustered groups along the whole transect. Samples without living specimens are highlighted with an 'x'. (c) Zonally and vertically interpreted communities across the subtropical South Atlantic. Station names were placed with warm to cold colors in (a) and (b) in order to separate and to locate different depth layers. The main species contribution for each group is shown in table 3.**

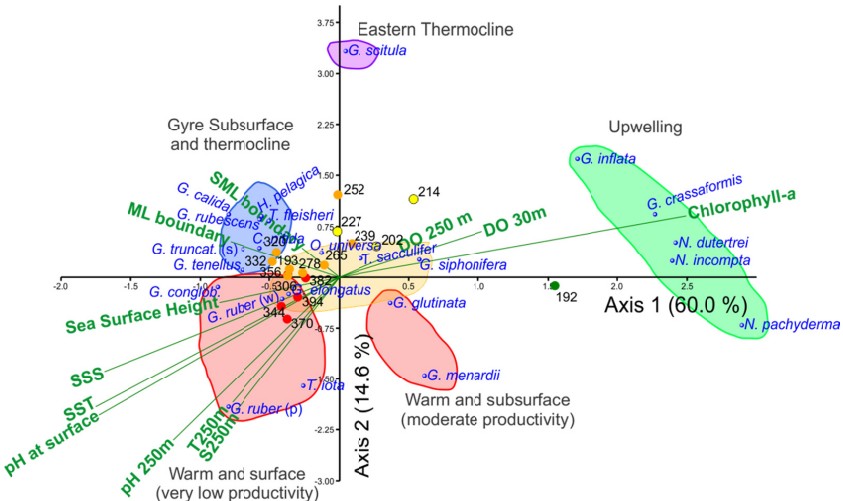

**Figure 8. Triplot of the CCA of the M124 stations with overlapped depth sections showing the two most important axis and highlighting species groups. The triplot is composed by stations, species with more than 2 % in abundance, and environmental parameters measured by CTD and remote sensing for surface and thermocline layers as well as their boundary depths.**


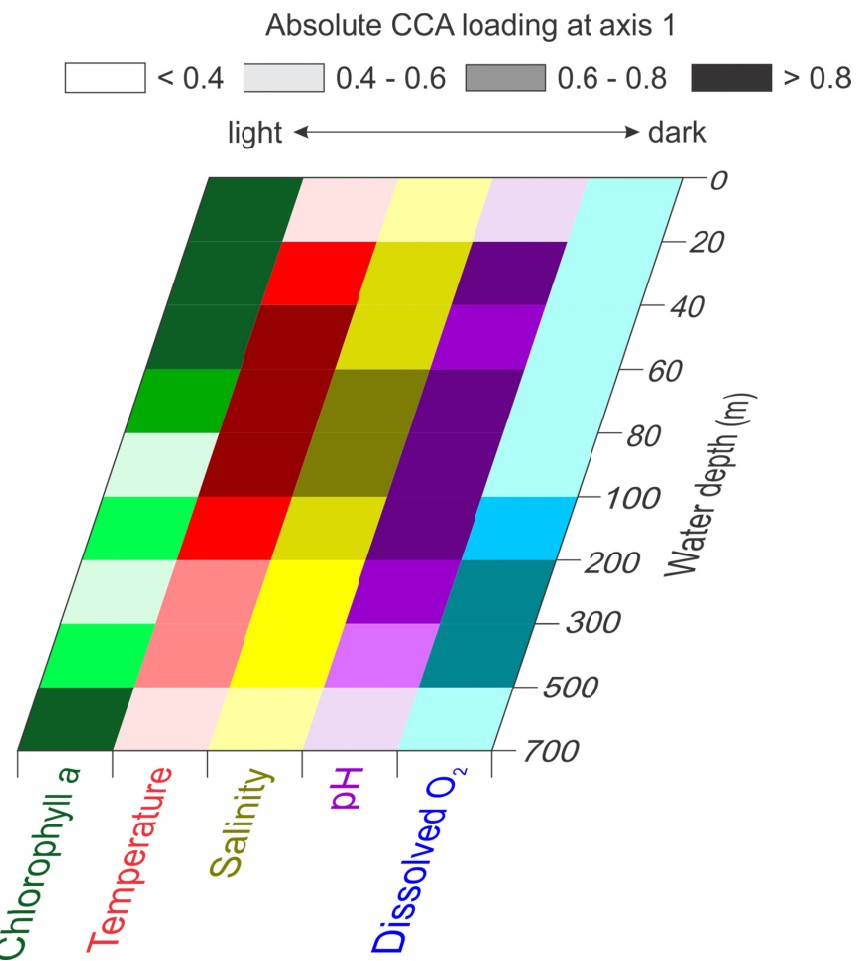

**Figure 9. CCA results for single depth resolved sections. color shading shows loadings values (absolute) for each environmental parameter. The CCA outputs are shown on table S1.**





**Table 1. Stations of the M124 cruise, location, time (day/month/year), environmental parameters for the mixed and thermocline layers, depth intervals, method used for preservation of the sample, counting size and filtered seawater volume.**

| Station | Latitude (degree decimal) | Longitude (degree decimal) | Date | DOY | Lunar day | MLD (m) | SST (°C) | Temperature at 250 m (°C) | SSS | Salinity at 250 m | Dissolved oxygen at 30 m (ml/l) | Dissolved oxygen at 250 m (ml/l) | pH at 30 m | pH at 250 m | Total chlorophyll-a | DCM (m) | SSH anomaly (cm) | Depth sections | Maximum depth (m) | Count size | Filtered volume (m³) |
|---|---|---|---|---|---|---|---|---|---|---|---|---|---|---|---|---|---|---|---|---|---|
| 394 | -26.25 | -36.12 | 16/03/2016 | 76 | 7 | 121.00 | 27.49 | 16.49 | 36.83 | 35.77 | 5.78 | 6.04 | 8.88 | 8.75 | 41.60 | 109.00 | 16.20 | 0-20, 20-40, 40-60, 60-80, 80-100, 100-200, 200-300, 300-500, 500-700 | 700 | > 100 μm | 175 |
| 382 | -26.89 | -33.89 | 15/03/2016 | 75 | 6 | 78.00 | 26.49 | 14.97 | 35.26 | 35.55 | 7.93 | 7.69 | 8.80 | 8.71 | 42.42 | 121.10 | 9.00 | 0-20, 20-40, 40-60, 60-80, 80-100, 100-200, 200-300, 300-500, 500-700 | 700 | > 100 μm | 175 |
| 370 | -26.88 | -28.69 | 14/03/2016 | 74 | 5 | 121.00 | 27.59 | 18.72 | 36.72 | 36.15 | 7.31 | 7.65 | 8.70 | 8.72 | 38.23 | 114.00 | 8.75 | 0-20, 20-40, 40-60, 60-80, 80-100, 100-200, 200-300, 300-500, 500-700 | 700 | > 100 μm | 175 |
| 356 | -26.87 | -24.63 | 13/03/2016 | 73 | 4 | 206.00 | 26.59 | 15.93 | 36.25 | 35.59 | 5.59 | 6.09 | 8.72 | 8.65 | 38.46 | 116.00 | 25.00 | 0-20, 20-40, 40-60, 60-80, 80-100, 100-200, 200-300, 300-500, 500-700 | 700 | > 100 μm | 175 |
| 344 | -29.18 | -20.05 | 12/03/2016 | 72 | 3 | 144.70 | 25.79 | 15.09 | 36.33 | 35.50 | 5.59 | 5.81 | 8.67 | 8.63 | 39.71 | 144.70 | 16.45 | 0-20, 20-40, 40-60, 60-80, 80-100, 100-200, 200-300, 300-500, 500-700 | 700 | > 100 μm | 175 |
| 332 | -29.56 | -16.28 | 11/03/2016 | 71 | 2 | 102.00 | 24.25 | 14.38 | 35.97 | 35.39 | 6.16 | 6.24 | 8.58 | 8.58 | 42.73 | 128.40 | 9.70 | 0-20, 20-40, 40-60, 60-80, 80-100, 100-200, 200-300, 300-500, 500-700 | 700 | > 100 μm | 175 |
| 320 | -30.03 | -11.66 | 10/03/2016 | 70 | 1 | 86.30 | 24.35 | 14.11 | 36.22 | 35.33 | 5.72 | 5.67 | 8.56 | 8.47 | 41.67 | 141.70 | 8.50 | 0-20, 20-40, 40-60, 60-80, 80-100, 100-200, 200-300, 300-500, 500-700 | 700 | > 100 μm | 175 |
| 306 | -32.05 | -10.67 | 09/03/2016 | 69 | 0 | 228.60 | 23.18 | 15.73 | 36.06 | 35.58 | 7.71 | 7.26 | 8.54 | 8.53 | 45.02 | 131.40 | 24.60 | 0-20, 20-40, 40-60, 60-80, 80-100, 100-200, 200-300, 300-500, 500-700 | 700 | > 100 μm | 175 |
| 193 | -31.17 | -8.22 | 08/03/2016 | 68 | 29 | 116.10 | 22.60 | 14.80 | 36.04 | 35.42 | 7.86 | 7.05 | 8.50 | 8.55 | 43.24 | 128.30 | 11.70 | 0-20, 20-40, 40-60, 60-80, 80-100, 100-200, 200-300, 300-500, 500-700 | 700 | > 100 μm | 175 |
| 278 | -31.10 | -3.57 | 07/03/2016 | 67 | 28 | 86.44 | 23.03 | 14.03 | 36.02 | 35.30 | 7.54 | 6.68 | 8.49 | 8.46 | 54.65 | 116.00 | 10.15 | 0-20, 20-40, 40-60, 60-80, 80-100, 100-200, 200-300, 300-500, 500-700 | 700 | > 100 μm | 175 |
| 265 | -31.41 | 0.58 | 06/03/2016 | 66 | 27 | 102.70 | 22.69 | 15.26 | 35.91 | 35.49 | 6.13 | 7.26 | 8.34 | 8.46 | 65.44 | 118.60 | 9.58 | 0-20, 20-40, 40-60, 60-80, 80-100, 100-200, 200-300, 300-500, 500-700 | 700 | > 100 μm | 175 |
| 252 | -29.92 | 1.01 | 05/03/2016 | 65 | 26 | 240.90 | 23.27 | 15.50 | 35.95 | 35.51 | 7.76 | 7.35 | 8.27 | 8.39 | 42.02 | 76.00 | 19.50 | 0-20, 20-40, 40-60, 60-80, 80-100, 100-200, 200-300, 300-500, 500-700 | 700 | > 100 μm | 175 |
| 239 | -31.89 | 3.63 | 04/03/2016 | 64 | 25 | 120.10 | 22.71 | 13.33 | 35.86 | 35.21 | 7.78 | 6.81 | 8.20 | 8.21 | 61.80 | 105.70 | 0.30 | 0-20, 20-40, 40-60, 60-80, 80-100, 100-200, 200-300, 300-500, 500-700 | 700 | > 100 μm | 175 |
| 227 | -33.89 | 7.39 | 03/03/2016 | 63 | 24 | 202.00 | 21.71 | 14.86 | 35.90 | 35.39 | 8.02 | 7.35 | 8.23 | 8.22 | 57.52 | 86.30 | 32.00 | 0-20, 20-40, 40-60, 60-80, 80-100, 100-200, 200-300, 300-500, 500-700 | 700 | > 100 μm | 175 |
| 214 | -34.45 | 11.09 | 02/03/2016 | 62 | 23 | 82.19 | 21.30 | 12.59 | 35.63 | 35.13 | 7.16 | 6.56 | 8.17 | 8.15 | 63.94 | 49.40 | 1.00 | 0-20, 20-40, 40-60, 60-80, 80-100, 100-200, 200-300, 300-500, 500-700 | 700 | > 100 μm | 175 |
| 202 | -34.13 | 14.94 | 01/03/2016 | 61 | 22 | 61.72 | 21.67 | - | 35.33 | - | 5.74 | - | 7.91 | - | 56.81 | 49.30 | -32.00 | 0-20, 20-40, 40-60, 60-80, 80-100, 100-200, 200-300, 300-400, 400-500 | 500 | > 100 μm | 125 |
| 192 | -34.39 | 17.56 | 29/02/2016 | 60 | 21 | 85.35 | 21.41 | - | 35.46 | - | 8.00 | - | 8.29 | - | 76.93 | 48.50 | 9.60 | 0-20, 20-40, 40-60, 60-80, 80-100 | 100 | > 100 μm | 25 |





Table 2. Average living depth and vertical dispersion of the 30 species with at least two living individuals counted in a station. The list was sorted by the maximum number of occurrences of living individuals within the samples. Interpretations of depth habitat and its corresponding variability or stability were performed for species with enough number of individuals. Species with * had an insufficient number of living individuals or the population was dominated by pre-adults, rendering interpretations about the depth habitat doubtful.

| Specie | Maximum N | ALD | SD ALD | VD | Depth habitat | Depth habitat variability |
|---|---|---|---|---|---|---|
| *Globigerinoides ruber* (white) | 375 | 28.04 | 6.95 | 16.7 | Surface mixed layer | stable |
| *Globigerinella calida* | 328 | 39 | 90.0 | 25.1 | Mixed Layer | variable |
| *Globigerinoides ruber* (pink) | 252 | 25.61 | 15.81 | 12.1 | Surface mixed layer | stable |
| *Trilobatus sacculifer* | 209 | 26 | 7.3 | 15.1 | Surface mixed layer | stable |
| *Tenuitella iota* | 123 | 37 | 16.3 | 25.0 | Mixed Layer | stable |
| *Neogloboquadrina pachyderma* | 80 | 68 | 123.6 | 47.5 | Subsurface | variable |
| *Globoturborotalita rubescens* | 76 | 39 | 11.6 | 25.9 | Surface mixed layer | stable |
| *Globigerinoides tenellus* | 67 | 41.11 | 17.51 | 24.0 | Mixed Layer | stable |
| *Globorotalia scitula* | 41 | 235 | 81.3 | 108.8 | Thermocline | variable |
| *Tenuitella fleisheri* | 39 | 141 | 92.5 | 63.7 | Thermocline | variable |
| *Candeina nitida* | 33 | 34 | 15.9 | 20.5 | Mixed Layer | stable |
| *Globorotalia truncatulinoides* (sinistral) | 31 | 173 | 57.1 | 55.7 | Thermocline | stable |
| *Globirerinita glutinata* | 28 | 45 | 20.0 | 29.2 | Mixed Layer | stable |
| *Globigerinoides conglobatus* | 27 | 27.50 | 16.56 | 15.3 | Surface mixed layer | * |
| *Globorotalia crassaformis* | 20 | 65 | 116.8 | 30.5 | Subsurface to thermocline | variable |
| *Globigerinella siphonifera* | 15 | 89 | 63.8 | 22.4 | Mixed Layer | variable |
| *Hastigerina pelagica* | 14 | 193 | 189.4 | 74.5 | Thermocline | variable |
| *Globigerinoides elongatus* | 13 | 54.59 | 47.72 | 25.3 | Subsurface | variable |
| *Neogloboquadrina incompta* | 11 | 57 | 28.7 | 34.2 | Subsurface | stable |
| *Neogloboquadrina dutertrei* | 9 | 52 | 18.9 | 32.0 | Subsurface | stable |
| *Turborotalita clarkei* | 8 | 77 | 102.4 | 65.8 | * | * |
| *Orbulina universa* | 7 | 30.35 | 15.72 | 11.1 | Surface mixed layer | * |
| *Turborotalita quinqueloba* | 6 | 152 | 120.5 | 7.4 | * | * |
| *Globorotalia menardii* | 6 | 39 | 53.2 | 22.2 | Mixed Layer | * |
| *Globorotalia inflata* | 5 | 95 | 166.3 | 36.1 | Subsurface to thermocline | * |
| *Dentigloborotalia anfracta* | 4 | 219 | 158.2 | 8.2 | * | * |
| *Globigerina bulloides* | 3 | 57 | 54.2 | 10.7 | * | * |
| *Berggrenia pumilio* | 3 | 155 | 196.1 | 10.4 | * | * |
| *Globorotalia truncatulinoides* (dextral) | 2 | 275 | 144.3 | | * | * |

Table 3. Contribution of most abundant planktonic foraminifera species (> 5 %) in each cluster (given by average relative abundance).



| Cluster | species | Contribution (%) | Cluster | specie | Contribution (%) |
|---|---|---|---|---|---|
| 1 | G. inflata | 28.8 | 7 | G. ruber white | 29.7 |
| | T. clarkei | 14.3 | | T. fleisheri | 10.1 |
| | G. calida | 13.8 | | G. glutinata | 9.7 |
| | N. incompta | 11.7 | | G. rubescens | 9.0 |
| | N. pachyderma | 8.6 | | T. iota | 8.8 |
| | G. siphonifera | 6.3 | | G. calida | 6.0 |
| | T. quinqueloba | 6.3 | | | |
| 2 | T. fleisheri | 51.4 | 8 | T. sacculifer | 37.9 |
| | T. iota | 11.2 | | G. rubescens | 9.4 |
| | G. inflata | 11.1 | | G. glutinata | 8.9 |
| | G. calida | 6.0 | | G. ruber white | 8.6 |
| | G. crassaformis | 6.0 | | | |
| 3 | G. crassaformis | 19.7 | 9 | G. rubescens | 40.4 |
| | N. pachyderma | 18.8 | | G. calida | 10.5 |
| | T. sacculifer | 10.2 | | T. sacculifer | 10.4 |
| | G. scitula | 8.2 | | T. fleisheri | 8.6 |
| | G. inflata | 5.8 | | G. tenellus | 7.3 |
| | G. ruber white | 5.2 | | | |
| 4 | T. sacculifer | 21.4 | 10 | G. scitula | 37.6 |
| | G. ruber white | 16.5 | | G. glutinata | 15.4 |
| | G. calida | 14.7 | | N. pachyderma | 11.3 |
| | G. ruber pink | 12.7 | | T. sacculifer | 10.4 |
| | G. rubescens | 8.6 | | G. rubescens | 6.2 |
| | G. tenellus | 7.4 | | G. calida | 5.8 |
| | T. iota | 6.1 | | | |
| 5 | G. ruber white | 21.8 | 11 | G. scitula | 54.9 |
| | G. tenellus | 20.1 | | T. fleisheri | 8.7 |
| | G. rubescens | 9.0 | | | |
| | G. glutinata | 7.5 | | | |
| | G. elongatus | 7.1 | | | |
| | T. iota | 6.5 | | | |
| | T. sacculifer | 5.7 | | | |
| 6 | G. ruber white | 42.8 | 12 | T. fleisheri | 17.7 |
| | T. sacculifer | 15.5 | | G. truncatulinoides (s) | 14.4 |
| | G. rubescens | 11.9 | | G. scitula | 12.1 |
| | T. iota | 5.5 | | G. tenellus | 8 |
| | G. glutinata | 5.3 | | G. calida | 6.1 |
| | | | | G. rubescens | 5.8 |
| | | | | H. pelagica | 5.3 |





Appendix A

Description and plates of main planktonic foraminifera species found during M124 Cruise. Plates are found in the supplementary material

Spinose species

*Globigerinella calida* Parker (1962) (Plate 1, 1 – 2)

Specimens with very low trocospiral coiling. The last whorl comprises from four to five chambers that rapidly increase in size. Chambers are elongated and much separated with a spined and canceled surface. Sutures are slightly curved and deep in both umbilical and spiral sides. The main aperture is extraumbilical with a long arch (1/4 circle) from the umbilicus to the periphery bordered by a lip. Pre-

adult specimens have the same morphological characteristics than adult specimens. Living specimens have a brownish stained cytoplasm.

*Globigerinoides conglobatus* Brady (1879) (Plate 1, pictures 3 – 5)

Only immature adult specimens of *G. conglobatus* were found during the cruise. Specimens were marked by low trocospiral coiling with four rough and densely spined chambers in the last whorl. All

chambers were spheric in pre-adult specimens (plate 1: picture 5), immature adult specimens have a slight flattened last chamber (Plate 1: pictures 3 and 4). The main aperture is umbilical, a low arch bordered by a rim similar to *Globigerina bulloides* in adult specimens. The primary aperture of pre-adult specimens is droplet shaped arch slightly displaced from the umbilicus. There are two small supplementary apertures by chamber in the spiral side, which are placed near to each other and they

are visible in both adult and pre-adult specimens. Living specimens have a brownish stained cytoplasm. Pre-adult *G. conglobatus* is distinguished of *G. calida* by the umbilical aperture and the presence of supplementary apertures in the spiral side. *G. conglobatus* can be distinguished of *G. bulloides* by the rough and densely spinned chamber's surface, and/or by the presence of supplementary apertures in the spiral side.

*Globigerinoides ruber* d'Orbigny (1839) (Plate 1, pictures 6 -13)

Adult specimens of *G. ruber* were low or moderate trocospiral tests, with large size and three spherical and symmetric placed chambers in the last whorl. Most of specimens presented a ruber-type wall structure (rougher than *G. bulloides*, but less than *G. calida* or *Globoturborotalita rubescens*, Schiebel and Helembem, 2017) with spines. Most of specimens found in the Agulhas Leakage and Benguela

realms presented sacculifer-type wall structure (honeycomb-like spines base, Schiebel and Helembem, 2017) wall surface. The primary aperture is an umbilical semicircular arch centralized over the suture of the two previous chambers. Two secondary apertures are found in the spiral side placed moderately far to each other. Pre-adults specimens diverge to adult specimens, their tests presented 3 ½ (neanic stage) to 4 ½  (juvenile stage) chambers in the last whorl. The surface is composed by very few spine

and pores and looks like microperforate in stereomicroscope view. The primary aperture is umbilical-extraumbilical reaching the periphery in juvenile specimens, migrating to the center in posterior ontogenetic states. No secondary apertures are visible in the spiral side. Living *G. ruber* specimens presents a brownish cytoplasm. Adult *G. ruber* pink specimens present bright pink chambers and pre-adult specimens presents pallid to bright pink chambers. It is difficult to separate pre-adult *G. ruber*

pink and pre-adult *G. rubescens*, the best way is to observe the equatorial periphery that tends to follow both adult *G. ruber* and G. rubescens. These same properties can be used to separate the neanic stage of *G. ruber* from late neanic stage of *Trilobatus sacculifer*. The neanic stage of *G. ruber* white can be separated from *Globigerinita glutinata* in stereomicroscope view by the presence of a semicircular primary aperture in *G. ruber* and the brownish cytoplasm.

*Globigerinoides elongatus* (Plate 1, pictures 14 and 15)





The test morphology is the same than *G. ruber*. *G. elongatus* is distinguished of *G. ruber* white by the flattening of chambers, especially the last one. The primary aperture is a reverse U-shaped arch in spite of a perfect semicircular G. ruber aperture. *G. elongatus* also presents a less deep umbilicus than *G. ruber* white. Living specimens presented a brownish cytoplasm. Pre-adult *G. ruber* and *G.*
*elongates* cannot be easily distinguished.

*Trilobatus sacculifer*  (Plate 2, pictures 1 – 6)

Test with large size, low trocospiral with sacculifer-type wall structure in adult specimens. Immature adult specimens (variants *T. trilobus*, *T. immaturus* and *T. quadrilobatus*) present spheric lobular chambers, 3 ½ in the last whorl. Mature adult specimens (variant *T. sacculifer*) present an elongated
sac-like last chamber and four chambers in the last whorl. Chambers are rather separated and increase size quickly. In immature adult specimens, the umbilicus is narrow and the primary aperture is interiomarginal umbilical, a long arch bordered by a rim. Mature adult specimens, the primary aperture is a more pronounced arch centralized over the third chamber, bordered by a rim. Adult specimens present one supplementary aperture by chamber in the spiral side. The morphology of pre-
adult *T. sacculifer* diverges from adults. Early neanic specimens present near to smooth wall structure with six or more compacted globoid chambers. Late neanic specimens present four chambers in the last whorl, the sacculifer-type wall structure is already present, but the equatorial view resembles more *Globorotalia inflata* than a typical *Globigerinoides*. The primary aperture is equatorial in juvenile specimens (Shiebel and Helemben, 2017), migrating to umbilicus in the next ontogenetic states. In
neanic specimens, the primary aperture is interiormarginal, a near to semicircle arch bordered by a rim and no supplementary aperture is visible. Late neanic T. sacculifer differs of neanic *G. ruber* white by the *Globorotalia inflata*-like chambers and more chambers visible in the spiral side.

*Globoturborotalita rubescens* Hofker (1956) (Plate 2, pictures 7 – 10)

Specimens are small or medium size, low to medium trocospiral with ruber-type wall structure. The
axial view is diamond-shaped (  ) with four chambers in the last whorl. Chambers are globoid and spheric with cancelate surface and small size increase by chamber. The primary aperture is umbilical, a semicircle ach bordered by a rim. No supplementary aperture visible. Pre-adult individuals present five to six near to smooth chambers in the last whorl and umbilical-extraumbilical aperture. Specimens present in general pink stained chambers, the living ones present brownish cytoplasm and a more
pronounced pink stained chambers. It is difficult to differ G. rubescens and neanic G. ruber pink/white. The difference is the diamond-shaped axial periphery. Pre-adult *G. rubescens* present more visible chambers in the juvenile whorl than *G. ruber*.

*Globigerinoides tenellus* Parker (1958) (Plate 2, pictures 10 and 11)

Tests are morphologically similar to *G. rubescens*. Specimens of *G. tenellus* have more cancelate
chambers and never pink stained. The primary aperture's arch goes more than a semicircle and there is a supplementary aperture in the spiral side, not always easily visible.

*Orbulina universa* d'Orbigny (1839) (Plate 2, pictures 12 and 13)

Mature adult specimens of *O. universa* have a big size with a single spheric chamber that covers the whole organism. The chamber is translucent and presents pores of diverse sizes and some spines.
Pores and the observation of previous Globigerina-like growing inside the spheric chamber are some features that allow differ *O. universa* from spheric Radiolarians. Immature adult specimens present a Globigerina-like growing with a low trocospiral test, four chambers in the last whorl and an umbilical aperture, an long arch sometimes bordered by a rim that covers almost all previous chambers  (similar to *Globigerina bulloides*). In some tests, a supplementary aperture in the spiral side was observed.
Chambers are smooth and very delicate and fragile, which can be broken with a moderate brush pressure. Living *O. universa* has a brownish cytoplasm, very dark brownish cytoplasm in immature adults. Pre-adult *O. universa* specimens were also present in the M124 collection, they differs from





immature adults by having five chambers in the last whorl a near to planispiral test, with can be differed from *G. calida* and *G. siphonifera* by smooth and very fragile chambers.

No spinose and macroperforate species

Genera *Neogloboquadrina* (Plate 3, pictures 1- 7)

Individuals of N. dutertrei and others Neogloboquadrinid species of the M124 collection were found in the east-most stations belonging to Benguela and Agulhas Leakage fauna. Neogloboquadrinid species tended to diverge morphologically from global *Neogloboquadrina* pattern.

*Neogloboquadrina dutertrei* d'Orbigny (1839) (Plate 3, pictures 1- 3)

Individuals with medium – bigsize and low trocospiral. Rounded axial view. Chambers with a ridge-
like wall structure (*Neogloboquadrina* type), 4 ½ or more unities in the last whorl. The morphology of chambers in M124 specimens diverged from the expected *Neogloboquadrina* pattern. They are more spheric and separated. Umbilicus opened and the aperture of two or three last chambers covers the upper side. The aperture is interiormarginal extraumbilical, sometimes with a tooth-like structure. Living individuals presented a hazel-greenish cytoplasm. Tests with four or four and a half chambers
in the last whorl were considered as *N. dutertrei* only if the tooth-like structure was present inside the aperture.

*Neogloboquadrina incompta* and *Neogloboquadrina pachyderma* (Plate 3, pictures 4 – 7)

*N. incompta* and *N. pachyderma* were classified according to specimen coiling direction. Right coiling specimens were classified as *N. incompta* and left coiled specimens were classified as *N. pachyderma*.
Tests with the size varying from small to medium and low trocospiral. The axial periphery is squared, with 4 or 4 ½ chambers in the last whorl. *N. incompta* and *N. pachyderma* specimens of M124 cruise collection present the Neogloboquadrina-type wall structure. However, this feature demanded more than 100 x magnifications in order to visualize in estereomicroscope. In < 100 x magnifications, tests looks smooth, similar to microperforate species. Similar to *N. dutertrei*, chambers of *N. incompta* and
*N. pachyderma* were much more spheric and separated by each other, diverging from typical Neogloboquadrina pattern. Umbilicus almost closed and the primary aperture is interiormarginal extraumbilical, an almost straight arch with a pronounced lip. Living individuals presented a transparent or pallid green cytoplasm color. Tests of *N. incompta* or *N. pachyderma* were differed from *N. dutertrei* with four or four and a half chambers in the last whorl if the aperture was bordered
by a lip and the test presented a squared axial periphery and a very smooth surface. The extraumbilical aperture with a lip and the moderate chambers's size increase allowed us differ these atypical smooth tests from *Globigerinita glutinata* and *Tenuitella iota* in low estereomicroscope magnification.

*Globorotalia crassaformis* (Plate 3, pictures 8 – 10; plate 4, pictures 1 – 3)

Individuals of *G. crassaformis* were encountered rarely in Subtropical Gyre (western side of the
transect) stations and in low abundances in Agulhas Leakage and Benguela (east side of transect) stations. However, test morphology differed strongly between west (Subtropical Gyre) and east (Agulhas Leakage and Benguela) stations. Specimens classified as *G. crassaformis* have a test very low trocospiral with a flat spiral side and a conic umbilical side. The outline is angular with a square appearance. The equatorial border is sharp giving a plan-convex aspect if side viewed. Specimens
present 4 – 4 ½ chambers in the final whorl. Chambers are smooth with visible pores in the spiral side and pustules in the umbilical side forming a strong calcite crust. Sutures strongly curved in the spiral side and almost straight in the umbilical side.

In specimens of Subtropical Gyre (Plate 4, pictures 1 – 3), the chambers border is strongly sharpen with a keel appearance, with strictly four unities in the last whorl. The umbilical view resembles
*Globorotalia truncatulinoides*, with the aperture inside the umbilicus, which is interiormarginal extraumbilical, a slit with a lip. Living individuals presented a strong dark hazel stained cytoplasm.



Those specimens can be differed from *G. truncatulinoides* by having four chambers in the last whorl and a squared outiline.

In specimens of Agulhas Leakage and Benguela (Plate 3, pictures 8 – 10), the chambers have a slight globular shape, but they still maintain a concave side view. Most of them were small individuals with 4 ½ chambers in the last whorl. The umbilical view resembles *Globorotalia inflata*, but the aperture is a short semicircle arch. Living specimens presented a pallid green stained cytoplasm. Those individuals can be differed from *G. inflata* by having more than four chambers in the last whorl, a plan-convex side view, curved sutures in the spiral side and a short arch of the aperture.

*Globorotalia menardii* (Plate 4, pictures 4 – 6)

Tests very low trocospiral, rounded outline with a peripheral keel and a very bi-flat side view. Chambers are smooth with pores, flat and sharp with five or six unities in the last whorl. Some carbonate pustules are common near to the aperture. Sutures curved and keeled in the spiral side, slight curved and deepen in the umbilical side. The aperture is interiormarginal extraumbilical, a slit arch bordered by a lip. Some specimens presented a flap going out the aperture. Living specimens presented a greenish cytoplasm with a strong red spot in early chambers.

*Globorotalia inflata* (Plate 4, pictures 7 – 9).

Tests very low trocospiral, in general medium sized. The outline is very slighted angulated and the side view is a not sharp plan-convex. Chambers are globoid, smooth with some tooth-like calcite pustules, bean-shaped in the spiral side and triangle-shaped in the umbilical view, three to four unities in the last whorl. Sutures very slight curved deepen in both spiral and umbilical side. The aperture is a long arch interiormarginal extraumbilical, sometimes bordered by a rim. Living specimens presents a pallid green or light hazel cytoplasm. G. inflata was found only in Agulhas Leakage and Benguela stations occupying the subsurface and upper thermocline layers. Despite the not easy identification, individuals of *G. inflata* can be differed from eastern *G. crassaformis* by presenting three chambers in the last whorl, globoid chambers, smoother surface, very slight curved sutures and a big aperture in long arch.

*Globorotalia scitula* (Plate 4, pictures 10 – 12)

Tests very low trocospiral; rounded outline without keel and a biconvex side view. Chambers are smooth with big pores and sharp border, five unitiea in the last whorl. Sutures are very curved and almost flat in the spiral side and slightly curved and deepened in the umbilical side. The aperture is an slit bordered by a lip interiormarginal extraumbilical. Living individuals had a dark hazel cytoplasm. G. scitula differed from G. crassaformis by having more chambers in the last whorl, a near to flat biconvex side view and a rounded outline.

*Globorotalia truncatulinoides* (Plate 5, pictures 1 – 3)

Tests very low trocospiral and conic, adults reach more than 400 μm. The spiral side is flat and the umbilical side is triangular, giving a plan-convex (cone-shaped) side view. The outline is very round with a thick keel. Chambers are pyramidal, keeled border calcite encrusted umbilical side by high number of pustules, five unities in the last whorl. The umbilicus is deep, similar to a volcano crater in some specimens. The aperture is located in the bottom of the umbilicus, a slit bordered by a lip, interiormarginal extraumbilical. Living specimens presented a brow to dark hazel cytoplasm. G. truncatulinoides is differed from G. crassaformis of the Subtropical Gyre by having a round outline, five chambers in the last whorl and a thick peripheral keel.

Microperforate non-spinose species

*Globigerinita glutinata* (Plate 5, pictures 4 and 5)





Test low trocospiral, small or medium sized. The wall structure is, smooth with very small calcite granules, pores is not visible in estereomicroscope, 3 ½ or four chambers in the last whorl. Chambers are globoid and spheric, compressed on each other and increase size moderately. Mature adults of *G. glutinata* develop a "bulla" last chamber over the umbilicus, but it was very rare in M124 collection.

Umbilicus near to closed and the primary aperture is umbilical, a very narrow arch bordered by a lip or rim. A secondary aperture in the spiral side may be present in the last chamber of some specimens. Living specimens presented a green or orange stained cytoplasm. Pre-adult specimens differ morphologically from adults, but they cannot be differed easily from pre-adults of *Tenuitella* species and immature *Turborotalita clarkei* in stereomicroscope.

*Candeina nitida* (Plate 5, pictures 6 – 8)

Test low trocospiral, medium to big sized. The wall structure is very smooth, with very few calcite granules, giving a transparent and reflective surface. Micropores are not visible in stereomicroscope. There are three chambers in the last whorl, which are globoid and spheric and compressed. Adults specimens does not present an unique primary aperture. Instead this, several sutural small apertures are
present. Pre-adult specimens present a small primary aperture and resembles *G. glutinata*, which can be differed by the much smoother surface and the persistence of three chambers in the juvenile whorl (visible in the spiral side). Living adult specimens presented between green and orange cytoplasm color.

*Tenuitella iota* (Plate 5, pictures 9 and 10)

Tests low trocospiral, tiny to small sized. The wall structure is smooth with many thick spike-shaped calcite granules. Micropores are not visible in stereomicroscope. Chambers are globoid spheric, separated among each other, four unities in the last whorl. Mature adults of *T. iota* develop a "bulla" last chamber over the umbilicus, but no specimens with this feature was observed in the M124 collection. The primary aperture is umbilical-extraumbilical, a short arch bordered by a rim. Living
specimens presents a green to hazel cytoplasm, with orange or red stained early chambers. Adults *T. iota* differ from *G. glutinata* by presenting thick and spike-shaped calcite granules and more separated chambers. Pre-adults specimens have five chambers in the last whorl, globoid chambers with a flattened side view, and extraumbilical aperture, but other pre-adults *Tenuitella* species and *G. glutinata* present a similar morphology, turning difficult the separation.

*Tenuitella fleisheri* (Plate 5, pictures 11 – 13)

Tests low trocospiral, tiny to medium sized (up to 200 µm). The wall structure is smooth, some specimens present few calcite granules, and others specimens have many. Chambers are flattened globoid, elongated, becoming ampulated in mature adults. The growing morphology shows four to five separated chambers in the last whorl with a moderate increase rate. The aperture is
interiormarginal extra umbilical with a flap-like lip. There is no difference between pre-adults and adults specimens, but other pre-adults *Tenuitella* species and *G. glutinata* present a similar morphology, turning difficult the separation. Living individuals presented greenish hazel or hazel cytoplasm. Adult *T. fleisheri* differ from *T. iota* by having a flattened side view. Adult *T. fleisheri* diffe from *T. pakerae* by not having pronounced elongated chambers and more calcite granules on the
surface.