# Peer review of "Vertical distribution of planktonic foraminifera in the Subtropical South Atlantic: depth hierarchy of controlling factors"

_Biogeosciences, 2019_

## Referee Comment (RC1) · Antje Voelker (Referee) · 30 Oct 2019

Antje Voelker (Referee)

antje.voelker@ipma.pt

Lessa and co-authors analyzed the planktonic foraminifera fauna in vertical plankton tows along a transect crossing the subtropical South Atlantic where a dearth of such data exists. They correlate the faunal observations to physical and chemical water column data to unearth water mass specific assemblages and environmental properties controlling species presence/abundance. In addition, they infer average living depths for the foraminifera species encountered. This study is an important contribution to our understanding of planktonic foraminifera diversity and environmental conditions controlling their presence and abundance in different regions of the world oceans.

The paper is well written and well-structured and I do not have any major comments. My comments -listed below- just point out small improvements. There are a few grammatical issues in the listed that I marked in the uploaded pdf file. Many of them occur in the Appendix with the species description. One important correction is in the first paragraph of the Conclusions: it needs to say "assess" instead of "access".

Specific comments: 1) Figure call-outs/ order of Figures: a) Figure 4 (p. 4, line 28; p. 5, line 30) before Figure 3 (p. 5, line 51). b) Figure 6 is only referred to in the discussion (p. 9, line 21) and thus after all the others.

2) For the reader it would be helpful if the Results would have sub-headers, i.e. the lengthily text gets subdivided.

3) relationship between pH values and planktonic foraminifera: this relationship or better said the indirect relationship that you infer between organic carbon degradation, microbial respiration and pH needs more explanation and references supporting this. In general, I am missing a text "justifying" why including pH in the CCA makes scientific sense (and is not just done because the parameter was measured).

4) p. 2 lines 26-27: it would be helpful (for future studies), if you could specify which are the relevant environmental parameters and depth ranges a study should cover.

5) p. 3 1st paragraph: please provide details on the CTD manufacturer (Seabird??) and the sensors used to measure oxygen, chlorophyll a and pH. Identifying the particular sensor is also relevant information for the data uploaded to Pangaea, so that other uses (or oceanographic databases like GLODAP) can judge the quality of the data.

6) p. 3 lines 28 and following: include here references to the appendix and the plates. The same could be done in the first paragraph of the Results.

7) p. 5 line 7: you are not delimitating water mass boundaries, but the mixed layer and the permanent thermocline.

8) p. 5 line 10: DO anomaly: mention here why you calculated it and how (just mention

the "how" in the figure caption is not enough).

9) p. 5 line 12: I would point out that station 202 is in the Benguela upwelling, i.e. make it easier for the reader.

10) p. 6 line 1: List the names of the cold water species and specify if you mean the adult or total fauna. I have trouble following your argument/ seeing this in Fig. 3b.

11) p. 6 line 15: refer to Fig. 4 at the end of the paragraph.

12) p. 6 last paragraph: please refer to Cluster numbers for the different oceanographic provinces/faunas. Relating Table 3 to Figure 7 is not easy. I would also recommend to provide the province/fauna information together with the cluster number in Table 3, especially as subsequent cluster numbers do not necessary refer to the same province/fauna.

13) p. 6 line 52: it would be good if you sometimes specified again that you mean the permanent thermocline when you write thermocline.

14) p. 7 line 15: correct/complete the Morey references. In line 17, can you provide a reference for the influence of temperature on respiration and growth rates?

15) p. 7 line 49: specify here (or earlier; see comment 3) how respiration contributes to pH and in which direction the change is – does higher respiration cause a lower pH?

16) I would have liked to see in the manuscript short comments on/references to: a) how does the Agulhas leakage fauna you identify compare to the one defined by Peeters et al. (2004; Science) or seen in the Loncaric (2006) paper you are citing? b) may be highlighting that your Benguela upwelling fauna includes few G. bulloides, the species most often associated with upwelling. c) can you specify/distinguish if the shape of (some of) your G. truncatulinoides specimens agrees with the shape of sp. 3 (or others) defined by de Vargas et al., 2001. Pleistocene adaptive radiation in Globorotalia truncatulinoides: genetic, morphologic, and environmental evidence. Paleobiology 27, 104-125. It looks to me as if the specimens depicted in the plate could belong to sp. 3. However, the presence of right coiling specimens in your samples would point to the presence of sp. 2 as well. You could mention that in your species description in the appendix.

Please also note the supplement to this comment:
https://www.biogeosciences-discuss.net/bg-2019-355/bg-2019-355-RC1-supplement.pdf

―――――――――――――――――

**Supplement:**

[revised manuscript text omitted]

---

## Referee Comment (RC2) · Ralf Schiebel (Referee) · 13 Nov 2019

The paper of Lessa and coauthors on the "Vertical distribution of planktonic foraminifera in the Subtropical South Atlantic: depth hierarchy of controlling factors" examines the so far neglected ecology of planktic foraminifers of the southern Atlantic Ocean, and adds important data to the understanding of planktic foraminifers in general. Therefore, the papers merits publication. However, the manuscript needs substantial improvement of data base, syntax, organization, and chain of argumentation (incl. use of references) before publication. I would suggest rejection of the current manuscript and resubmission of an improved paper.

[Figure]

Long passages of the text read nicely narrative, but not scientific (e.g., the second paragraph of section 2. Material and Methods). For example, mention that a device needs to be switched on is trivial and may want to be skipped in a scientific paper. The narrative writing may result from the fact that, for example, the first two paragraphs of the chapter 2. Material and Methods read very much like the Meteor M124 cruise report (Karstensen et al. 2016). Considering this, Lessa and coauthors may want to be careful to avoid unintended plagiarism.

In general, the paper needs some reorganizing. The Results and Discussion chapters include information from the other chapters. For example: Page 5, lines 18-27: In this section, methodology, results, and discussion are mixed, and should be disentangled. Page 5, last paragraph: results and interpretation are mixed up: please disentangle. Page 7, in lines 34-38, and lines 48-49, the "vertical variation of the community" is discussed in the Results chapter. Page 8, lines 14-15, results on ontogenetic effects are presented in the Discussion chapter. Page 10, lines 5-6, is Methods, not Discussion; here the results should be discussed I don't understand page 10, lines 18-19; please rephrase. From page 10, line 19, the discussion reads rambling and not to the point. Lines 24-26: The observation of Fehrenbacher et al has been on N. dutertrei, not Neogloboquadrina in general; please be specific. Line 27, degraded organic matter; ref. to Schiebel and Hemleben (2017). Lines 32-34: syntax?! Line 38: please change "hidden in" to "from". Lines 39-41: any proof? Please refer to data or figures or literature references!

From Figure 2, I have the impression that some of the environmental data are wrong. This might result from the fact that "raw data" are presented instead of "final data", i.e. calibrated data. In a publication (not preliminary report), calibrated values should be presented, which have undergone quality control. In particular, the high pH values (near 8.8) are possibly not realistic in open marine waters, and the data should not be used. I would guess that the pH probe was broken or not correctly calibrated. In addition, DO values are very high, and may be revisited / calibrated. Having said that,

I would suggest to revisit all data to guarantee correct values.

The use of the term "permanent thermocline" (e.g., page 4, line 36), given in the manuscript, is wrong. Actually, multiple seasonal thermoclines are observed (Figure S1), out of which even the deepest seasonal thermoclines are not the permanent thermocline. In some profiles, even deeper seasonal thermoclines can be seen (e.g., Profile 370 near 200 m). The permanent thermocline is much deeper at possibly all of the case shown in Figure S1. Unfortunately, there is not much literature available on this topic for the very stations discussed here (for a start, Chiessi 2008, and Gordon 1981 may be consulted).

Classification: Given the rough surface texture, closed umbilicius, and shape and number of chambers in the final whorl (6) of the specimens depicted in Plate 3 images 1-3, this is possibly T. humilis, and certainly not N. dutertrei. I do also have a different idea about T. iota, shown in Plate 5, images 9-10, and the rather unusual distribution pattern of T. iota (page 8, lines 47-49, "...the shallow habitat of T. iota is at odds with its concentration maximum around 300 m in the NE Atlantic reported by Rebotim et al. (2017). Clearly, the ecology of this species requires further investigation") may result from misidentification.

Another misunderstanding concerns the classification of adult versus pre-adult individuals (lines 31-33): "...were classified as "pre-adult" when their identification was performed at a magnification higher than 100x and surface features typically found in adults (e.g., spines, pustules, large pores) were lacking." This is not a valid method. To distinguish adult from pre-adult individuals, GAM calcification should be looked at to get an idea about the average size of adult vs. pre-adult individuals. If this is not possible, the terms small (i.e., smaller than ...) and large (i.e., larger than...) tests may be used.

The use of statistics is this paper is the wrong way round, or presented in the wrong way. In general, statistics may be used to confirm and explain observations, and may

not be an end in itself in paleoceanography (in mathematics, this may be the other way round).

In general, referencing in the manuscript is selective, and much important information has not been included in the paper. This is particularly inadequate, because little has so far been known on the planktic foraminifers from the region sampled here, and the results would need to be discussed in comparison to existing studies in a similar setting, as, for example, the northern limit of the North Atlantic subtropical gyre. Referring only to Rebotim et al. (2017) is not sufficient. Most importantly, the paper of Kemle-von-Mücke and Hemleben (1999), in "South Atlantic Zooplankton" needs to be discussed. Page 2, line 2: Temperature is possibly an indicator, not "control"; see, e.g., Jentzen et al. 2018 Page 2, lines 8-9: please see also Schiebel et al. 2001 (among others) Page 2, line 18: please see also Schiebel 2002 (among many others) Page 8, line 12: please refer to Bijma et al 1990 Page 8, line 37: "in many other studies/regions": please be specific; which studies/regions? Refer to the papers of Bé, Bijma, Jentzen, Salmon, Schmuker, Schiebel, etc Page 8, line 39: why only "thermally constrained"; please discussion with reference to the existing literature (e.g., Jentzen et al. 2018, etc) Page 9, line 6: what is meant by majority? Please be specific, and discuss the different species. Page 9, lines 41-42: "G. truncatulinoides replaces G. scitula towards. . ."; please compare to the distribution of G. truncatulinoides and G. scitula in the Azores Front Current System, which is a similar hydrological and ecological setting as studied here. Page 9, line 54: ". . .the result of seasonal superposition of. . ."; please discuss in comparison to earlier papers. You may start from Schiebel and Hemleben, 2000, and Schiebel et al. 2001

Finally, the chapter 6. Conclusions may be rewritten follwing the changes in the manuscript.

Some details: Title: I wonder why a rather self-limiting title has been chosen for the much broader topic presented in the paper. I would suggest to skip "Vertical" and make the title "Distribution...". Page 2, line 39: not "cod-end" (which are soft) have

been used, but "sampling cups" Page 2, line 51: I wonder how the nets were changed "manually" at grate depth: I guess that the right expression is "changed by remote control" Page 3, line 3: pH, not PH Page 3, line 9: skip "planktonic" Page 3, line 39: change "concentrations" to "standing stocks" Page 4, line 5: change "trace" to "confirm" Page 4, lines 42-44: change "first" to "upper" Page 5, lines 18-19: unfinished relative clause: higher than what? Page 5, lines 44-45: unfinished relative clause: higher than what? Page 6, line 44: change "revealed" to "confirmed" Page 7, line 15: (refs Moery etc) is not the correct way of referencing Page 8, lines 29-30: better change "this reference" to "this depth level" Page 11, line 9: How should pH affect species distribution? Any data that may support the statement? Page 13, lines 17-18: not eds. but authors Figure 4: The upper 260 m max are displayed, not 700 m as stated in the caption. Figure 9: What shall "light" to "dark" mean in this context? The different parameters from Chl-a to O2 may not be easily put into relation. Plates: I congratulate the authors on the quality of the light micrographs. However, using a ring light produces light rings on the reflecting surface of chambers. The authors may want to play with more diffused light to produce even better images in the future. Plate 2: change second (12) to (13) Appendix A: The species descriptions read good in general, but some typos (e.g., page 25, line 6, change "trocospiral" to "trochospiral"; page 29, line 39, change pakerae to parkerae; etc etc) may be corrected. I have no clue what is meant by granules (in T. iota, and T. fleisheri), and pustules may be meant.

---

## Author Comment (AC1) · 13 Nov 2019

We thank Dr. Antje Voelker for her review, which will help to improve the manuscript. We have carefully read the comments and we tried to answer all queries clearly and concisely. We also checked carefully the supplement of the comment and all found small issues will be corrected in both main text and appendix.

Lessa and co-authors analyzed the planktonic foraminifera fauna in vertical plankton tows along a transect crossing the subtropical South Atlantic where a dearth of such data exists. They correlate the faunal observations to physical and chemical water column data to unearth water mass specific assemblages and environmental properties

controlling species presence/abundance. In addition, they infer average living depths for the foraminifera species encountered. This study is an important contribution to our understanding of planktonic foraminifera diversity and environmental conditions controlling their presence and abundance in different regions of the world oceans. The paper is well written and well-structured and I do not have any major comments. My comments -listed below- just point out small improvements. There are a few grammatical issues in the listed that I marked in the uploaded pdf file. Many of them occur in the Appendix with the species description.

One important correction is in the first paragraph of the Conclusions: it needs to say "assess" instead of "access".

R: The requested correction will be done.

Specific comments: 1) Figure call-outs/ order of Figures: a) Figure 4 (p. 4, line 28; p. 5, line 30) before Figure 3 (p. 5, line 51).

R: In order to organize the order of figures, we will insert the sea surface height variation on figure 2 as a new panel and we will update each panel call-outs of the figure 2 along the text.

b) Figure 6 is only referred to in the discussion (p. 9, line 21) and thus after all the others.

R: We will carry out some modifications in this part of the text in order to insert Fig. 6.

2) For the reader it would be helpful if the Results would have sub-headers, i.e. the lengthily text gets subdivided.

R: Sub-headers will be inserted in the results section.

3) relationship between pH values and planktonic foraminifera: this relationship or better said the indirect relationship that you infer between organic carbon degradation, microbial respiration and pH needs more explanation and references supporting this.

In general, I am missing a text "justifying" why including pH in the CCA makes scientific sense (and is not just done because the parameter was measured).

R: We thank Dr. Voelker for her comment. We consider pH as an important parameter to be evaluated since this parameter is closely linked to CO2 chemistry (Clayton et al., 1995), whose one of main natural factors is the biologic respiration (Hofmann et al, 2011). The pH has also been evaluated in studies focusing not only the current ocean acidification, but also reconstruction of CO2 levels by pH proxies (Manno et al, 2012, Rae, 2012). Our data showed a zonal variation of the pH opposed to the total chlorophyll-a, with effect over planktonic foraminifer's community composition at the permanent thermocline (Fig. 2a, 2e, 7). Then, the pH can help to identify habitats and environmental processes that could determine the distribution of species at the permanent thermocline layer. We will add more explanation about the importance of pH in the Material and Methods section.

4) p. 2 lines 26-27: it would be helpful (for future studies), if you could specify which are the relevant environmental parameters and depth ranges a study should cover.

R: The requested statements will be done

5) p. 3 1st paragraph: please provide details on the CTD manufacturer (Seabird??) and the sensors used to measure oxygen, chlorophyll a and pH. Identifying the particular sensor is also relevant information for the data uploaded to Pangaea, so that other uses (or oceanographic databases like GLODAP) can judge the quality of the data.

R: Information about the CTD's manufacturer and sensor types will be inserted at Material and Methods section.

6) p. 3 lines 28 and following: include here references to the appendix and the plates. The same could be done in the first paragraph of the Results.

R: References for the appendix and plates will be added

7) p. 5 line 7: you are not delimitating water mass boundaries, but the mixed layer and

the permanent thermocline. R: "water masses" will be replaced by "mixed layer and permanent thermocline" at this part of the text. 8) p. 5 line 10: DO anomaly: mention here why you calculated it and how (just mention the "how" in the figure caption is not enough).

R: Information about the calculation of the DO anomaly plot will be inserted at this part of the text.

9) p. 5 line 12: I would point out that station 202 is in the Benguela upwelling, i.e. make it easier for the reader.

R: We will insert "Benguela region" inside brackets at this part of the text.

10) p. 6 line 1: List the names of the cold water species and specify if you mean the adult or total fauna. I have trouble following your argument/ seeing this in Fig. 3b.

R: The requested corrections will be carried out.

11) p. 6 line 15: refer to Fig. 4 at the end of the paragraph.

R: The Fig. 4 will be cited at the end of the paragraph.

12) p. 6 last paragraph: please refer to Cluster numbers for the different oceano-graphic provinces/faunas. Relating Table 3 to Figure 7 is not easy. I would also recommend to provide the province/fauna information together with the cluster number in Table 3, especially as subsequent cluster numbers do not necessary refer to the same province/fauna.

R: The respective clusters will be inserted in brackets after the fauna entries. The final faunal groups will be inserted in the first column of the table 3.

13) p. 6 line 52: it would be good if you sometimes specified again that you mean the permanent thermocline when you write thermocline.

R: For better clarification, we will check all thermocline entries and we will specify them

as "seasonal" or "permanent" thermocline. We will also highlight that the "thermocline group" refers to "communities below the seasonal thermocline" for better clarification.

14) p. 7 line 15: correct/complete the Morey references.

R: The reference will be updated at this part of the text.

In line 17, can you provide a reference for the influence of temperature on respiration and growth rates?

R: The reference Sandnes et al (2005) will be inserted at this part of the text.

15) p. 7 line 49: specify here (or earlier; see comment 3) how respiration contributes to pH and in which direction the change is – does higher respiration cause a lower pH?

R: More information about the relationship between microbial activity and pH will be inserted at this part of the text.

16) I would have liked to see in the manuscript short comments on/references to: a) how does the Agulhas leakage fauna you identify compare to the one defined by Peeters et al. (2004; Science) or seen in the Loncaric (2006) paper you are citing?

R: We will add a short comparison between our Agulhas Leakage fauna and the ones from Peeters et al (2004) and Loncaric (2006) at the cluster analysis paragraph.

b) may be highlighting that your Benguela upwelling fauna includes few G. bulloides, the species most often associated with upwelling.

R: We will insert information about the small contribution of others cold water species linked to Benguela fauna at page 7.

c) can you specify/distinguish if the shape of (some of) your G. truncatulinoides specimens agrees with the shape of sp. 3 (or others) defined by de Vargas et al., 2001. Pleistocene adaptive radiation in Globorotalia truncatulinoides: genetic, morphologic, and environmental evidence. Paleobiology 27, 104-125. It looks to me as if the specimens depicted in the plate could belong to sp. 3. However, the presence of right coiling specimens in your samples would point to the presence of sp. 2 as well. You could mention that in your species description in the appendix.

R: A short comment about the genetic type (cryptospecies) will be inserted at G. truncatulinoides description in the appendix A.

Supplement pdf: RC1 comment: Figure 1b: station 193 or 293:

R: This station is defined as 193 in the cruise report (Karstensen et al., 2016). Based on this, we will keep the current station numbering.

References Clayton, T. D., Byrne, R. H., Breland, J. A., Feely, R. A., Millero, F. J., Campbell, D. M., murphy, P. P., and Lamb, M. F. (1995). The role of pH measurements in modern oceanic $CO_2$-system characterizations: Precision and thermodynamic consistency. Deep Sea Research Part II: Topical Studies in Oceanography, 42(2-3), 411-429. Hofmann GE, Smith JE, Johnson KS, Send U, Levin LA, Micheli F, et al. (2011) High-Frequency Dynamics of Ocean pH: A Multi-Ecosystem Comparison. PLoS ONE 6(12): e28983. https://doi.org/10.1371/journal.pone.0028983 Karstensen, J., Speich, S., Morard, R., Bumke, K., Clarke, J., Giorgetta, M., Fu, Y., Köhn, E., Pinck, A., Manzini, E., Lübben, B., Baumeister, A., Reuter, R., Scherhag, A., de Groot, T., Louropoulou, E., Geißler, F., and Raetke, A. (2016). Oceanic & atmospheric variability in the South Atlantic Cruise No. M124. DFG-MARUM, Bremen, 59 p. Lončarić, N.(2006). Planktic Foraminiferal Content in a Mature Agulhas Eddy from the SE Atlantic: Any Influence on Foraminiferal Export Fluxes?, Geologia Croatica, 59(1), 41–50. Manno, C., Morata, N., and Bellerby, R. (2012). Effect of ocean acidification and temperature increase on the planktonic foraminifer Neogloboquadrina pachyderma (sinistral). Polar Biology, 35(9), 1311-1319. Peeters, F. J., Acheson, R., Brummer, G. J. A., De Ruijter, W. P., Schneider, R. R., Ganssen, G. M., Ufkes, E., and Kroon, D. (2004). Vigorous exchange between the Indian and Atlantic oceans at the end of the past five glacial periods. Nature, 430(7000), 661-665. Rae, J. W. (2018). Boron isotopes in foraminifera: Systematics,

biomineralisation, and co 2 reconstruction. In Boron Isotopes (pp. 107-143). Springer, Cham., 107-143.

---

## Author Comment (AC2) · 13 Dec 2019

We appreciated the comments of Prof. Dr. Ralf Schiebel (Referee 2 – R2), which will make important improvements to the manuscript, especially with literature suggestions. Based on his comments, we have rechecked our environmental parameters obtained by CTD and we decided to change pH and dissolved oxygen data from CTD for total alkalinity from GLODAP and dissolved oxygen from World Ocean Atlas (WOA) 2013. We observed that total alkalinity and WOA dissolved oxygen was in part similar to variation to pH and dissolved oxygen from CTD, respectively, along the M124 transect. Thus, we believe that our discussion and conclusions will not suffer major changes.

Regarding the taxonomy concerns, our specimens exhibited in the plates were checked by all our co-author team and everyone agreed with the current taxonomy. We have carefully analyzed and responded each comment. Referee 2's comments are stated as "R2C" and our responses are stated as "R".

R2C: Long passages of the text read nicely narrative, but not scientific (e.g., the second paragraph of section 2. Material and Methods). For example, mention that a device needs to be switched on is trivial and may want to be skipped in a scientific paper. The narrative writing may result from the fact that, for example, the first two paragraphs of the chapter 2. Material and Methods read very much like the Meteor M124 cruise report (Karstensen et al. 2016). Considering this, Lessa and coauthors may want to be careful to avoid unintended plagiarism. R: We will work on the second paragraphs of Material and Methods rephrasing parts that can be narrative or similar to M124 cruise report. R2C: In general, the paper needs some reorganizing. The Results and Discussion chapters include information from the other chapters. For example: Page 5, lines 18-27: In this section, methodology, results, and discussion are mixed, and should be disentangled. R: The highlighted text interval will have some changes in order to disentangle sections. R2C: Page 5, last paragraph: results and interpretation are mixed up: please disentangle. R: This paragraph will be rephrased for better clarification. R2C: Page 7, in lines 34-38, and lines 48-49, the "vertical variation of the community" is discussed in the Results chapter. R: This part will be moved and incorporated to the penultimate paragraph of the section 5.3. R2C: Page 8, lines 14-15, results on ontogenetic effects are presented in the Discussion chapter. R: This part will be rewritten in order to remove the "result" apparent reading. R2C: Page 10, lines 5-6, is Methods, not Discussion; here the results should be discussed R: We will rephrase this text segment so that the discussion aspect is maintained. R2C: I don't understand page 10, lines 18-19; please rephrase. R: This segment will be rephrased for better clarification. R2C: From page 10, line 19, the discussion reads rambling and not to the point. Lines 24-26: The observation of Fehrenbacher et al has been on N. dutertrei, not Neogloboquadrina in general; please be specific. R: "Neogloboquadrina"

at line 25 will be replaced by N. dutertrei. R2C: Line 27, degraded organic matter; ref. to Schiebel and Hemleben (2017). R: The citation will be inserted at line 27 R2C: Lines 32-34: syntax?! R: This part will be rephrased for better clarification. R2C: Line 38: please change "hidden in" to "from". R: The requested change will be done R2C: Lines 39-41: any proof? Please refer to data or figures or literature references! R: We will insert references to figures 2 and 7, and we will change "... along the transect ..." for " ... observed between eastern and western thermocline faunas ..." at line 40 for better clarification.. R2C: From Figure 2, I have the impression that some of the environmental data are wrong. This might result from the fact that "raw data" are presented instead of "final data", i.e. calibrated data. In a publication (not preliminary report), calibrated values should be presented, which have undergone quality control. In particular, the high pH values (near 8.8) are possibly not realistic in open marine waters, and the data should not be used. I would guess that the pH probe was broken or not correctly calibrated. In addition, DO values are very high, and may be revisited / calibrated. Having said that, I would suggest to revisit all data to guarantee correct values. R: We thank to Referee for spotting this issue with pH and dissolved oxygen from CTD. Considering this, we rechecked CTD data and we observed that there was a problem with pH and dissolved oxygen data. The others parameters are ok and available on Pangaea (https://doi.pangaea.de/10.1594/PANGAEA.895426). Based on this, we decided to use total alkalinity data from GLODAP and March's dissolved oxygen data from World Ocean Atlas (WOA) 2013. We compared the variation between our probe and GLODAP/WOA databank and we observed that variation was in part similar along the transect. Because of this, we believe that our discussion and conclusions will not have major changes.

R2C: The use of the term "permanent thermocline" (e.g., page 4, line 36), given in the manuscript, is wrong. Actually, multiple seasonal thermoclines are observed (Figure S1), out of which even the deepest seasonal thermoclines are not the permanent thermocline. In some profiles, even deeper seasonal thermoclines can be seen (e.g., Profile 370 near 200 m). The permanent thermocline is much deeper at possibly all of the

case shown in Figure S1. Unfortunately, there is not much literature available on this topic for the very stations discussed here (for a start, Chiessi 2008, and Gordon 1981 may be consulted). R: Based on Gordon (1981), our 700 m profile reached near to the bottom of the permanent thermocline. Thus, we will keep the layer below the seasonal thermocline as permanent thermocline. However, Referee 1 stated issues with thermocline entries (seasonal or permanent), so we will check all thermocline entries and we will specify them as "seasonal" or "permanent" thermocline. We will also highlight that the "thermocline group" refers to "communities below the seasonal thermocline" for better clarification. R2C: Classification: Given the rough surface texture, closed umbilicius, and shape and number of chambers in the final whorl (6) of the specimens depicted in Plate 3 images 1-3, this is possibly T. humilis, and certainly not N. dutertrei. R: The specimens displayed in plates were carefully checked by our co-author team and everyone agree that this specimen is N. dutertrei. The shape of tests 1 to 3 on plate 3 is quite normal for small N. dutertrei specimen. There are no spines, the surface is ridge-shaped and N. dutertrei can have six or more chambers at the last whorl, and a small "tooth" is seen inside the penultimate chamber's aperture. We did not understand why this specimen cannot be classified as N. dutertrei. R2C: I do also have a different idea about T. iota, shown in Plate 5, images 9-10 R: Specimens 9 and 10 from the plate were also carefully checked by our co-authors team. The specimen 9 had all morphologic requisites to be classified as T. iota with four chambers at the last whorl, microperforate surface, and plenty of spiked pustules in the umbilical side. We are confident about the quality of our taxonomy.. R2C: and the rather unusual distribution pattern of T. iota (page 8, lines 47-49, "...the shallow habitat of T. iota is at odds with its concentration maximum around 300 m in the NE Atlantic reported by Rebotim et al. (2017). Clearly, the ecology of this species requires further investigation") may result from misidentification. R: The ecology of T. iota is little known, so it is expected that the depth habitat can vary depending to oceanic realm and the season. For example, G. crassaformis, was observed with very shallow distribution in the Eastern Azores current (Rebotim et al, 2017) whereas it distributed below 100 m in other parts

of the tropical Atlantic (this study, Jentzen et al, 2018, Meilland et al, 2019). Thus, it is more probable that the environmental conditions of the South Atlantic Subtropical Gyre during the late summer controlled the distribution of T. iota instead of taxonomic misidentification. R2C: Another misunderstanding concerns the classification of adult versus pre-adult individuals (lines 31-33): "...were classified as "pre-adult" when their identification was performed at a magnification higher than 100x and surface features typically found in adults (e.g., spines, pustules, large pores) were lacking." This is not a valid method. To distinguish adult from pre-adult individuals, GAM calcification should be looked at to get an idea about the average size of adult vs. pre-adult individuals. If this is not possible, the terms small (i.e., smaller than ...) and large (i.e., larger than...) tests may be used. R: We thank to Referee 2 for his worry about adult and pre-adult terms. Based on it, we will carry out a rephrasing of this part in order to clarify the used method for adult and pre-adult specimens separation. The classification of adults and pre-adult specimens was based on morphological aspects described by Brummer et al (1986). The "pre-adult" term was used to separating specimens in juvenile and neanic stages from adults, since these two initial stages shared morphological similarities that could not be correctly separated on M124 collection. GAM calcification is a feature observed in specimens that reached the terminal adult ontogenetic stage. This feature is difficultly recognizable on light stereomicroscope and it may not correctly separate immature from mature adults. GAM calcification is also difficult to recognize in Brummer et al (1986) images of adult specimens. R2C: The use of statistics is this paper is the wrong way round, or presented in the wrong way. In general, statistics may be used to confirm and explain observations, and may not be an end in itself in paleoceanography (in mathematics, this may be the other way round). R: The statistical analyses used in our study had the purpose to give a numerical support to our observations and providing valuable information about our described faunas and their relationship with environmental processes that occur in the upper ocean. Indeed we have the purpose that our findings may also be applied on paleoceanography. Based on this, we believe that providing statistics between our faunas and environmental parameters is an effec-

tive way to characterize the relationships between the species relative abundances and environmental variables.. R2C: In general, referencing in the manuscript is selective, and much important information has not been included in the paper. This is particularly inadequate, because little has so far been known on the planktic foraminifers from the region sampled here, and the results would need to be discussed in comparison to existing studies in a similar setting, as, for example, the northern limit of the North Atlantic subtropical gyre. Referring only to Rebotim et al. (2017) is not sufficient. Most importantly, the paper of Kemle-von- Mücke and Hemleben (1999), in "South Atlantic Zooplankton" needs to be discussed. R: We thank Referee 2 for his references suggestion, which will give a high improvement to the manuscript. Indeed searching for references was a tricky part of the manuscript's development, since there is very little information for the tropical and subtropical South Atlantic. We agree with the close-up to North Atlantic Subtropical Gyre to compare our results, which will be incorporated in the main text on Discussion section. R2C: Page 2, line 2: Temperature is possibly an indicator, not "control"; see, e.g., Jentzen et al. 2018 R: We will change the segment "...dominant temperature control on..." for "...high relationship of the temperature with..." and citing Jentzen et al (2018) R2C: Page 2, lines 8-9: please see also Schiebel et al. 2001 (among others) R: The citation will be inserted. R2C: Page 2, line 18: please see also Schiebel 2002 (among many others) R: The reference will be inserted together with Bé, 1960. R2C: Page 8, line 12: please refer to Bijma et al 1990 R: The reference will be inserted prior to figure S3 call. R2C: Page 8, line 37: "in many other studies/regions": please be specific; which studies/regions? Refer to the papers of Bé, Bijma, Jentzen, Salmon, Schmuker, Schiebel, etc R: The sentence will be rephrased to better clarification. R2C: Page 8, line 39: why only "thermally constrained"; please discussion with reference to the existing literature (e.g., Jentzen et al. 2018, etc) R: The sentence will be rephrased to better clarification R2C: Page 9, line 6: what is meant by majority? Please be specific, and discuss the different species. R: We revised that sentence and we concluded that it is irrelevant and it will be deleted. The cited reference will be placed at the end of the previous sentence. R2C: Page 9, lines

41-42: "G. truncatulinoides replaces G. scitula towards. . ."; please compare to the distribution of G. truncatulinoides and G. scitula in the Azores Front Current System, which is a similar hydrological and ecological setting as studied here. R: A comparison with the Azores System will be inserted in this paragraph. R2C: Page 9, line 54: ". . .the result of seasonal superposition of. . ."; please discuss in comparison to earlier papers. You may start from Schiebel and Hemleben, 2000, and Schiebel et al. 2001 R: This part will be rewritten comparing the requested references. R2C: Finally, the chapter 6. Conclusions may be rewritten following the changes in the manuscript. R: Based on our response given about CTD quality query, our conclusions may not have major changes since the discussed processes linked to pH also have effects over the dissolved oxygen. R2C: Some details: Title: I wonder why a rather self-limiting title has been chosen for the much broader topic presented in the paper. I would suggest to skip "Vertical" and make the title "Distribution. . .". R: The change will be implemented. R2C: Page 2, line 39: not "cod-end" (which are soft) have been used, but "sampling cups" R: The issue will be corrected at this point and elsewhere R2C: Page 2, line 51: I wonder how the nets were changed "manually" at grate depth: I guess that the right expression is "changed by remote control" R: "Manually will be changed by "offline" R2C: Page 3, line 3: pH, not PH Page 3, line 9: skip "planktonic" R: This issue was also signaled by Referee 1 and it will be corrected R2C: Page 3, line 39: change "concentrations" to "standing stocks" R: In this segment, we refer to the general concentration calculation, which includes both living and dead shells. Since "standing stock" is only referred to the living shells, we decided to maintain the word "concentration". R2C: Page 4, line 5: change "trace" to "confirm" R: The change will be done R2C: Page 4, lines 42-44: change "first" to "upper" R: The changes will be done R2C: Page 5, lines 18-19: unfinished relative clause: higher than what? R: The sentence will be rephrased for a superlative clause R2C: Page 5, lines 44-45: unfinished relative clause: higher than what? R: The sentence will be rephrased for a superlative clause R2C: Page 6, line 44: change "revealed" to "confirmed" R: The change will be done R2C: Page 7, line 15: (refs Moery etc) is not the correct way of referencing R: This issue was also signaled

by Referee 1 and it will be corrected R2C: Page 8, lines 29-30: better change "this reference" to "this depth level" R: The change will be done R2C: Page 11, line 9: How should pH affect species distribution? Any data that may support the statement? R: The sentence will be rephrased to better clarification R2C: Page 13, lines 17-18: not eds. but authors R: The correction will be done R2C: Figure 4: The upper 260 m max are displayed, not 700 m as stated in the caption. R: "700" will be changed by "550" in order to include the second panel of the figure 4. R2C: Figure 9: What shall "light" to "dark" mean in this context? The different parameters from Chl-a to O2 may not be easily put into relation. R: The figure 9 will have modifications since pH and dissolved oxygen data will be changed by GLODAP and WOA 2013 March data. R2C: Plates: I congratulate the authors on the quality of the light micrographs. However, using a ring light produces light rings on the reflecting surface of chambers. The authors may want to play with more diffused light to produce even better images in the future. R: We thank Referee 2 for your comment about our light micrographs. We acquired this microscope little time prior to the manuscript production. However, many light combinations were tried, but specimens with flat and smooth tests produced reflected light on our photos. This light combination produced the smallest impacts. We keep working on light combinations in order to remove the reflectance effect over smooth tests. On the other hand, this reflected light may give to readers, the idea that these tests have a glass-like appearance on light stereomicroscope. R2C: Plate 2: change second (12) to (13) R: The correction will be carried out R2C: Appendix A: The species descriptions read good in general, but some typos (e.g., page 25, line 6, change "trocospiral" to "trochospiral"; R: The text will be checked for issues typos R2C: page 29, line 39, change pakerae to parkerae; etc etc) may be corrected. R: The correction will be carried out R2C: I have no clue what is meant by granules (in T. iota, and T. fleisheri), and pustules may be meant. R: We will change "granules" for "pustules" along the appendix.

References Brummer, G. J. A., Hemleben, C., and Spindler, M.: Planktonic foraminiferal ontogeny and new perspectives for micropalaeontology, Nature, 319(6048), 50, 1986. Gordon, A. L.: South Atlantic thermocline ventilation. Deep

Sea Research Part A. Oceanographic Research Papers, 28(11), 1239-1264, 1981. Jentzen, A., Schönfeld, J., and Schiebel, R. : Assessment of the effect of increasing temperature on the ecology and assemblage structure of modern planktic foraminifers in the Caribbean and surrounding seas. Journal of Foraminiferal Research, 48(3), 251-272, 2018. Karstensen, J., Speich, S., Morard, R., Bumke, K., Clarke, J., Giorgetta, M., Fu, Y., Köhn, E., Pinck, A., Manzini, E., Lübben, B., Baumeister, A., Reuter, R., Scherhag, A., de Groot, T., Louropoulou, E., Geißler, F., and Raetke, A. Oceanic & atmospheric variability in the South Atlantic Cruise No. M124. DFG-MARUM, Bremen, 59 p., 2016. Meilland, J., Siccha, M., Weinkauf, M. F., Jonkers, L., Morard, R., Baranowski, U., Bertlich, J., Brummer, G.-J., Debray, P., Fritz-Endres, T., Groeneveld, G., Magrel, L., Munz, P., Rillo, M. C., Schmidt, C., Takagi, H., Theara, G., and Kucera, M.: Highly replicated sampling reveals no diurnal vertical migration but stable species-specific vertical habitats in planktonic foraminifera. Journal of Plankton Research. Journal of Plankton Research, 00(00), 1-15, doi:10.1093/plankt/fbz002, 2019. Rebotim, A., Voelker, A. H. L., Jonkers, L., Waniek, J. J., Meggers, H., Schiebel, R., Fraile, I., Schulz, M., and Kucera, M.: Factors controlling the depth habitat of planktonic foraminifera in the subtropical eastern North Atlantic, Biogeosciences, 14, 827-859, 2017.

---

## Author Response (AR1)

Dear Editor,

We are pleased to submit the revised version of the manuscript entitled "Distribution of planktonic foraminifera in the Subtropical South Atlantic: depth hierarchy of controlling factors" for consideration for publication in Biogeosciences. We apologize for the long time needed for the revision, but this was necessary since, considering the reviews, we decided to make substantial changes to the design of the analyses, which also required substantial changes to the discussion section of the paper.

We are grateful to the reviewers for requesting details on the oxygen and pH parameters and on questioning the interpretation of these parameters. This was very much warranted as some of the data proved to be unreliable. We have hence removed pH and we now use oxygen concentration data from CTD casts made in parallel with our sampling. Unfortunately, dissolved oxygen data is not available for two shallow stations (202 and 192), but this is more than compensated by the fact the new data are calibrated by on-board titration and thus entirely reliable. We have also considered the merit of using alkalinity, instead of pH, to explain the faunal variation. Initially, we considered that alkalinity at depth would be, like oxygen, to a certain degree related to respiration. However, a further examination of reliable alkalinity profiles in the South Atlantic convinced us that this parameter is too strongly reflecting the water masses rather than local respiration and that there is thus no merit in including it in our analyses as a parameter reflecting ecologically relevant processes.

As a result, we have re-calculated all of the ordinations including only reliable parameters that could conceivably explain the faunal variability. Whilst the general pattern and the main conclusion of our paper remain robust, there are differences in details, such as the depth-levels where the controlling parameters change, and, considering the comments by Ralf Schiebel, we are now more careful in the interpretation of the controlling factors at depth. Previously we were attributing the variation explicitly to respiration and thus POM concentration, but now we are more general and only state that at depth, the assemblages are likely affected by factors that are not analyzed, and that the existing data suggest a role of aggregates and biotic interactions.

Together with the extensive comments by both referees, we believe that these changes have allowed us to produce a much more robust case, supporting the conjecture of different factors affecting the species distribution at different depth layers. Our response to the reviews is appended below. We believe that the readers of this journal will find the presented dataset and interpretation not only exciting but perhaps also meaningful to their studies.

Kind regards,

Douglas Villela de Oliveira Lessa On Behalf of all authors

**Referee 1: Antje Voelker**

We thank Dr. Antje Voelker for her review, which helped to improve the manuscript. We have carefully read the comments and we tried to answer all queries clearly and concisely. We also checked carefully the supplement of the comment and all small issues will be corrected in both main text and appendix. The reviewer's comments are shown below in black and our response and changes to the manuscript are shown in blue. Referee 1's comments are stated as "R1C" and our responses are stated as "R". Page and line numbers in our responses refers to the manuscript's clean version.

R1C: Lessa and co-authors analyzed the planktonic foraminifera fauna in vertical plankton

tows along a transect crossing the subtropical South Atlantic where a dearth of such

data exists. They correlate the faunal observations to physical and chemical water column

data to unearth water mass specific assemblages and environmental properties

controlling species presence/abundance. In addition, they infer average living depths

for the foraminifera species encountered. This study is an important contribution to our

understanding of planktonic foraminifera diversity and environmental conditions controlling their presence and abundance in different regions of the world oceans.

The paper is well written and well-structured and I do not have any major comments.

My comments -listed below- just point out small improvements. There are a few grammatical issues in the listed that I marked in the uploaded pdf file. Many of them occur in the Appendix with the species description.

R1C: One important correction is in the first paragraph of the Conclusions: it needs to say "assess" instead of "access".

**R: The requested correction was done (page 9, lines 43).**

R1C: Specific comments: 1) Figure call-outs/ order of Figures: a) Figure 4 (p. 4, line 28; p. 5, line 30) before Figure 3 (p. 5, line 51).

R: We have updated the order in which the figures are mentioned in the text.

R1C: b) Figure 6 is only referred to in the discussion

(p. 9, line 21) and thus after all the others.

R: We carried out some modifications in this part of the text (now section 4.2) in order to insert Fig. 6 (page 6, line 5).

R1C: 2) For the reader it would be helpful if the Results would have sub-headers, i.e. the lengthily text gets subdivided.

R: Sub-headers were inserted in the results section.

R1C: 3) relationship between pH values and planktonic foraminifera: this relationship or better said the indirect relationship that you infer between organic carbon degradation, microbial respiration and pH needs more explanation and references supporting this. In general, I am missing a text "justifying" why including pH in the CCA makes scientific sense (and is not just done because the parameter was measured).

R: We thank Dr. Voelker for her comment. Our initial reason to include pH in the analysis was indeed based on the inferred indirect relationship between degradation and pH as a characterisation of the state or organic matter available to the foraminifera community. However, based on the critical comments by Referee 2, we discovered that the pH probe of the CTD did not function properly. So, we removed pH from the manuscript. Please also refer to our cover letter above.

R1C: 4) p. 2 lines 26-27: it would be helpful (for future studies), if you could specify which are the relevant environmental parameters and depth ranges a study should cover.

R: The requested statements was done (page 2, lines 29-31)

R1C: 5) p. 31st paragraph: please provide details on the CTD manufacturer (Seabird??) and the sensors used to measure oxygen, chlorophyll a and pH. Identifying the particular sensor is also relevant information for the data uploaded to Pangaea, so that other uses (or oceanographic databases like GLODAP) can judge the quality of the data.

R: Information about the CTD's manufacturer and sensor types was inserted at Material and Methods section (page 2, line 48).

R1C: 6) p. 3 lines 28 and following: include here references to the appendix and the plates.

The same could be done in the first paragraph of the Results.

R: References for the appendix and plates were added (page 3, line 10 and page 4, line 48)

R1C: 7) p. 5 line 7: you are not delimitating water mass boundaries, but the mixed layer and the permanent thermocline.

R: "water masses" was replaced by "mixed layer and permanent thermocline" at page 4, line 28.

R1C: 8) p. 5 line 10: DO anomaly: mention here why you calculated it and how (just mention the "how" in the figure caption is not enough).

R: As with pH, we re-evaluated the DO data and discovered that the DO probe also did not measure properly. Instead we have used calibrated DO data from the shipboard CTD rosette. Since we use DO and not DO anomaly in our analyses we decided to remove the DO anomaly from the text and figure 2.

R1C: 9) p. 5 line 12: I would point out that station 202 is in the Benguela upwelling, i.e. make it easier for the reader.

**R: This paragraph (page 4, lines 30 - 37) was rephrased**

R1C: 10) p. 6 line 1: List the names of the cold water species and specify if you mean the adult or total fauna. I have trouble following your argument/ seeing this in Fig. 3b.

R: The requested corrections were carried out (page 4, line 53 – page 5, line 1).

R1C: 11) p. 6 line 15: refer to Fig. 4 at the end of the paragraph.

R: The Fig. 4 was cited at the end of the paragraph (page 5, line 32).

R1C: 12) p. 6 last paragraph: please refer to Cluster numbers for the different oceanographic provinces/faunas. Relating Table 3 to Figure 7 is not easy. I would also recommend to provide the province/fauna information together with the cluster number in R1C: Table 3, especially as subsequent cluster numbers do not necessary refer to the same province/fauna.

R: The respective clusters were inserted in brackets after the fauna entries (page 6, lines 9 - 12). The final faunal groups were also inserted in the first column of the table 3.

R1C: 13) p. 6 line 52: it would be good if you sometimes specified again that you mean the permanent thermocline when you write thermocline.

R: For better clarification, we checked all thermocline entries and we specified them as "seasonal" or "permanent" thermocline. We also highlighted that the "thermocline group" refers to "communities below the seasonal thermocline" for better clarification (page 8, line 10).

R1C: 14) p. 7 line 15: correct/complete the Morey references.

R: The reference was updated, as well as this part was replaced to the section 5.3 (Page 8, line 41), as requested by the Referee 2.

R1C: In line 17, can you provide a reference for the influence of temperature on respiration and growth rates?

R: This paragraph (now section 4.3) was rewritten entirely.

R1C: 15) p. 7 line 49: specify here (or earlier; see comment 3) how respiration contributes to pH and in which direction the change is – does higher respiration cause a lower pH?

R: This part was excluded from the text, as new CCA plots were carried out due to the removal of pH and the replacement of DO data.

R1C: 16) I would have liked to see in the manuscript short comments on/references to:

a) how does the Agulhas leakage fauna you identify compare to the one defined by

Peeters et al. (2004; Science) or seen in the Loncaric (2006) paper you are citing?

R: We added a short comparison between our Agulhas Leakage fauna and the ones from previous studies (page 8, line 54).

R1C: b) may be highlighting that your Benguela upwelling fauna includes few G. bulloides, the species most often associated with upwelling.

R: We inserted *G. bulloides* in depth integrated CCA plots (figures 8 and S4), since no changes occurred in the data dispersion.

R1C: c) can you specify/distinguish if the shape of (some of) your G. truncatulinoides specimens agrees with the shape of sp. 3 (or others) defined by de Vargas et al., 2001. Pleistocene adaptive radiation in Globorotalia truncatulinoides: genetic, morphologic,

and environmental evidence. Paleobiology 27, 104-125. It looks to me as if the specimens depicted in the plate could belong to sp. 3. However, the presence of right coiling specimens in your samples would point to the presence of sp. 2 as well. You could mention that in your species description in the appendix.

R: A short comment about the genetic type (cryptospecies) was inserted at *G*. *truncatulinoides* description in the appendix A (page 31, lines 29 - 31).

R1C: Supplement pdf:

RC comment: Figure 1b: station 193 or 293:

R: This station is defined as 193 in the cruise report (Karstensen et al., 2016). Based on this, we kept the current station numbering.

References

Clayton, T. D., Byrne, R. H., Breland, J. A., Feely, R. A., Millero, F. J., Campbell, D. M., murphy, P. P., and Lamb, M. F. (1995). The role of pH measurements in modern oceanic CO2-system characterizations: Precision and thermodynamic consistency. Deep Sea Research Part II: Topical Studies in Oceanography, 42(2-3), 411-429.

Hofmann GE, Smith JE, Johnson KS, Send U, Levin LA, Micheli F, et al. (2011) High-Frequency Dynamics of Ocean pH: A Multi-Ecosystem Comparison. PLoS ONE 6(12): e28983. https://doi.org/10.1371/journal.pone.0028983

Karstensen, J., Speich, S., Morard, R., Bumke, K., Clarke, J., Giorgetta, M., Fu, Y., Köhn, E., Pinck, A., Manzini, E., Lübben, B., Baumeister, A., Reuter, R., Scherhag, A., de Groot, T., Louropoulou, E., Geißler, F., and Raetke, A. (2016). Oceanic & atmospheric variability in the South Atlantic Cruise No. M124. DFG-MARUM, Bremen, 59 p.

Lončarić, N.(2006). Planktic Foraminiferal Content in a Mature Agulhas Eddy from the SE Atlantic: Any Influence on Foraminiferal Export Fluxes?, Geologia Croatica, 59(1), 41–50.

Manno, C., Morata, N., and Bellerby, R. (2012). Effect of ocean acidification and temperature increase on the planktonic foraminifer *Neogloboquadrina pachyderma* (sinistral). Polar Biology, 35(9), 1311-1319.

Peeters, F. J., Acheson, R., Brummer, G. J. A., De Ruijter, W. P., Schneider, R. R., Ganssen, G. M., Ufkes, E., and Kroon, D. (2004). Vigorous exchange between the Indian and Atlantic oceans at the end of the past five glacial periods. Nature, 430(7000), 661-665.

Rae, J. W. (2018). Boron isotopes in foraminifera: Systematics, biomineralisation, and co 2 reconstruction. In *Boron Isotopes* (pp. 107-143). Springer, Cham., 107-143.

We appreciated the comments of Prof. Dr. Ralf Schiebel (Referee 2 - R2), which helped us to make important improvements to the manuscript, especially with literature suggestions. As indicated above, thanks to the comments we discovered that the pH and dissolved oxygen sensors of our plankton sampler did not work properly. Hence we now use dissolved oxygen concentration data from the O2 sensor mounted on the CTD rosette, these data were calibrated using conventional titration and passed quality control. Unfortunately, no reliable pH data are available. We have re-calculated all of the ordinations including only reliable parameters. Whilst the general pattern and the main conclusion of our paper remain robust, there are differences in details, such as the depth-levels where the controlling parameters change and, considering the comments, we are now more careful in the interpretation of the controlling factors at depth. Previously we were attributing the variation explicitly to respiration and thus POM concentration, but now we are more general and only state that at depth, the assemblages are likely affected by factors that are not analyzed, and that the existing data suggest a role of aggregates and biotic interactions.

Regarding the taxonomy concerns, our specimens exhibited in the plates were checked by all our co-author team and everyone agreed with the current taxonomy.

We have copied each comment below and provide our response in blue text. Referee 2's comments are stated as "R2C" and our responses are stated as "R". Page and line numbers in our responses refers to the manuscript's clean version.

R2C: Long passages of the text read nicely narrative, but not scientific (e.g., the second paragraph of section 2. Material and Methods). For example, mention that a device needs to be switched on is trivial and may want to be skipped in a scientific paper. The narrative writing may result from the fact that, for example, the first two paragraphs of the chapter 2. Material and Methods read very much like the Meteor M124 cruise report (Karstensen et al. 2016). Considering this, Lessa and coauthors may want to be careful to avoid unintended plagiarism.

**R: The second paragraph of the Material and methods was partially rephrased to better description of on board methods (page 2, lines 48-54 – page 3, lines 1-3)**

R2C: In general, the paper needs some reorganizing. The Results and Discussion chapters include information from the other chapters. For example: Page 5, lines 18-27: In this section, methodology, results, and discussion are mixed, and should be disentangled.

R: Some rephrasing was made in this paragraph (page 5, lines 29 - 35). However, this highlighted part corresponds to the section 3 (Oceanographic Conditions), where CTD and satellite data are used to describe the environment along the transect and we consider that the data presented here are not part of the main results of our study.

R2C: Page 5, last paragraph: results and interpretation are mixed up: please disentangle.

R: Corrections were done along the second paragraph of results (page 5, lines 9 - 19).

R2C: Page 7, in lines 34-38, and lines 48-49, the "vertical variation of the community" is discussed in the Results chapter.

R: This part was removed as pH data were removed from the manuscript.

R2C: Page 8, lines 14-15, results on ontogenetic effects are presented in the Discussion chapter.

R: This part was rewritten (Page 7, lines 12 - 15).

R2C: Page 10, lines 5-6, is Methods, not Discussion; here the results should be discussed

R: This text segment was rephrased so that the discussion aspect is maintained (page 8, lines 41 - 45).

R2C: I don't understand page 10, lines 18-19; please rephrase.

R: This paragraph was rephrased as changes in the environmental variable that we considered made reinterpretation necessary (page 9, lines 15 - 28).

R2C: From page 10, line 19, the discussion reads rambling and not to the point. Lines 24-26: The observation of Fehrenbacher et al has been on N. dutertrei, not Neogloboquadrina in general; please be specific.

R: This paragraph was rephrased as changes on CTD environmental parameters made reinterpretation necessary (Page 9, lines 15 - 39). Regarding the misobservation of Fehrenbacher et al, we replaced "*Neogloboquadrina*" for *N. dutertrei* (page 9, line 28).

R2C: Line 27, degraded organic matter; ref. to Schiebel and Hemleben (2017).

R: This paragraph was rewritten completely and the reference was included (page 9, lines 15 - 28).

R2C: Lines 32-34: syntax?!

R: The last paragraph of the section 5.3 was rewritten and syntax errors were corrected (page 9, lines 36 - 39).

R2C: Line 38: please change "hidden in" to "from".

R: This issue was excluded as the paragraph was reworded completely.

R2C: Lines 39-41: any proof? Please refer to data or figures or literature references!

R: We inserted references to figures 2 and 7, and we changed "... along the transect ..." for "... observed between eastern and western thermocline faunas ..." at line page 9, lines 37 - 38 for better clarification.

R2C: From Figure 2, I have the impression that some of the environmental data are wrong. This might result from the fact that "raw data" are presented instead of "final data", i.e. calibrated data. In a publication (not preliminary report), calibrated values should be presented, which have undergone quality control. In particular, the high pH values (near 8.8) are possibly not realistic in open marine waters, and the data should not be used. I would guess that the pH probe was broken or not correctly calibrated. In addition, DO values are very high, and may be revisited / calibrated. Having said that, I would suggest to revisit all data to guarantee correct values.

R: We thank the Referee for spotting these issues with pH and dissolved oxygen data from the sensors mounted on the plankton sampler. We checked environmental data again and found that there was indeed a problem with pH and dissolved oxygen sensors. No alternative pH or alkalinity measurements were made, but dissolved O2 (DO) data were measured using a different sensor mounted on the ship CTD rosette. However, at the time of writing calibrated DO data were not yet available, but we have been able to make use of quality controlled and calibrated DO data for this revision (Karstensen et al., 2016; data available at: https://doi.pangaea.de/10.1594/PANGAEA.895426).

We have repeated the multivariate analyses and changed the results and discussion where necessary (sections 3, 4.3 and 5.3). The depth integrated CCAs were recalculated with six variables (chlorophyll-*a*, temperature, salinity, SSH, UML depth and mixed layer depth) and depth separated CCAs were plotted with four CTD variables (chlorophyll-*a*, temperature, salinity and DO) and stations 202 and 192 excluded because i) for these stations we do not have all environmental data at our disposal and ii) the fauna at these stations was not characterised down to 700 m water depth.

We would like to stress that these new analyses with reliable data did not fundamentally affect the main conclusion that the planktonic foraminifera community is shaped by different environmental variables at different levels in the water column. There are however some differences in details, such as the depth-levels where the controlling

parameters change. Considering the comments, we are now more careful in the interpretation of the controlling factors at depth. Previously we were attributing the variation explicitly to respiration and thus POM concentration, but now we are more general and only state that at depth, the assemblages are likely affected by factors that are not analyzed, and that the existing data suggest a role of aggregates and biotic interactions.

R2C: The use of the term "permanent thermocline" (e.g., page 4, line 36), given in the manuscript, is wrong. Actually, multiple seasonal thermoclines are observed (Figure S1), out of which even the deepest seasonal thermoclines are not the permanent thermocline. In some profiles, even deeper seasonal thermoclines can be seen (e.g., Pro-file 370 near 200 m). The permanent thermocline is much deeper at possibly all of the case shown in Figure S1. Unfortunately, there is not much literature available on this topic for the very stations discussed here (for a start, Chiessi 2008, and Gordon 1981 may be consulted).

R: Based on Gordon (1981), our 700 m profile reached near to the bottom of the permanent thermocline. Thus, we kept the layer below the seasonal thermocline as permanent thermocline. However, Referee 1 stated issues with thermocline entries (seasonal or permanent), so we checked all thermocline entries and we specified them as "seasonal" or "permanent" thermocline for better clarification. We also highlighted that the "thermocline group" refers to "communities below the seasonal thermocline" (page 8, line 10).

R2C: Classification: Given the rough surface texture, closed umbilicius, and shape and number of chambers in the final whorl (6) of the specimens depicted in Plate 3 images 1-3, this is possibly *T. humilis*, and certainly not *N. dutertrei*.

R: The specimens displayed in plates were carefully rechecked by the author team and we remain convinced that the depicted specimen is *N. dutertrei*. The shape of tests 1 to 3 on plate 3 is quite normal for small *N. dutertrei* specimens. There are no spines, the surface is ornamented with ridges, small *N. dutertrei* can have six or more chambers at the last whorl, and a small "tooth" is often seen inside the penultimate chamber's aperture, see e.g. Brummer and Kroon (1988): page 12 figure 2 and page 259 Plate 3, figures 11-13.

R2C: I do also have a different idea about T. iota, shown in Plate 5, images 9-10

R: Specimens 9 and 10 from the plate were also carefully checked by the authors. We removed specimen 10 from plate 5 due to a not much didactic shape. However, both specimens follow all morphologic requisites to be classified as *T. iota*. The specimen 9

had four chambers in the last whorl, microperforate surface, and sparse but distinct pustules on the umbilical side. Specimen 10 also has a microperforated surface and the pustules can be seen near to the periphery of the spiral side. The identification of the specimens in our samples is consistent with the work by Parker (1962), cf specimen 29 on plate 10. We are therefore confident about the accuracy of our taxonomy.

R2C: and the rather unusual distribution pattern of T. iota (page 8, lines 47-49, "...the shallow habitat of T. iota is at odds with its concentration maximum around 300 m in the NE Atlantic reported by Rebotim et al. (2017). Clearly, the ecology of this species requires further investigation") may result from misidentification.

R: The ecology of *T. iota* is very poorly constrained, so it is expected that the depth habitat range of this species may be wide and variable depending on oceanic realm and season. Other species of planktonic foraminifera also show a wide and variable vertical distribution. For example, *G. crassaformis*, was observed with very shallow distribution in the Eastern Azores current (Rebotim et al, 2017), whereas it is distributed below 100 m in other parts of the tropical Atlantic (this study, Jentzen et al, 2018, Meilland et al, 2019). Given the lack of evidence that *T. iota* is a species strictly confined to the subsurface and our rigorous taxonomic control (see the point above), we interpret the shallow habitat of the species in South Atlantic Subtropical Gyre during the late summer as evidence of a wider vertical dispersion globally than observed near the Azores.

R2C: Another misunderstanding concerns the classification of adult versus pre-adult individuals (lines 31-33): "...were classified as "pre-adult" when their identification was performed at a magnification higher than 100x and surface features typically found in adults (e.g., spines, pustules, large pores) were lacking." This is not a valid method. To distinguish adult from pre-adult individuals, GAM calcification should be looked at to get an idea about the average size of adult vs. pre-adult individuals. If this is not possible, the terms small (i.e., smaller than ...) and large (i.e., larger than...) tests may be used.

R: We thank the Referee for pointing this out. Based on the comment, we rephrased this part of the manuscript in order to clarify the method for separation of adult and preadult specimens (page 3, lines 16 - 20). The classification of adults and pre-adult specimens was based on morphological aspects described by Brummer et al (1986). The "pre-adult" term was used to separate specimens in juvenile and neanic stages from specimens showing characters typical for adult and terminal stages. GAM calcification is only present during the terminal stage and thus rarely seen among living specimens in the plankton and we therefore abstained from distinguishing the ontogenetic stages with any greater precision (i.e. not separating adult and terminal stages). R2C: The use of statistics is this paper is the wrong way round, or presented in the wrong way. In general, statistics may be used to confirm and explain observations, and may not be an end in itself in paleoceanography (in mathematics, this may be the other way round).

R: The statistical analyses used in our study have the purpose to give numerical support to our observations and to provide objective information about the foraminifera fauna and their relationship with environmental variables. We have made effort to express more clearly the purpose of each analysis in the methods and results, which we hope addresses the concerns by the reviewer.

R2C: In general, referencing in the manuscript is selective, and much important information has not been included in the paper. This is particularly inadequate, because little has so far been known on the planktic foraminifers from the region sampled here, and the results would need to be discussed in comparison to existing studies in a similar setting, as, for example, the northern limit of the North Atlantic subtropical gyre. Referring only to Rebotim et al. (2017) is not sufficient. Most importantly, the paper of Kemle-von- Mücke and Hemleben (1999), in "South Atlantic Zooplankton" needs to be discussed.

R: We thank Referee 2 for his suggestions, which we have now included in the discussion section.

R2C: Page 2, line 2: Temperature is possibly an indicator, not "control"; see, e.g., Jentzen et al. 2018

R: We changed the segment "...dominant temperature control on..." for "...strong relationship of the temperature with..." and citing Jentzen et al (2018) (page 2, lines 1 - 3).

R2C: Page 2, lines 8-9: please see also Schiebel et al. 2001 (among others)

R: The citation was inserted (page 2, line 8).

R2C: Page 2, line 18: please see also Schiebel 2002 (among many others)

R: The reference was inserted together with Bé, 1960 (page 2, line 18).

R2C: Page 8, line 12: please refer to Bijma et al 1990

R: The reference was inserted prior to figure S3 call (page 7, line 10).

R2C: Page 8, line 37: "in many other studies/regions": please be specific; which studies/regions? Refer to the papers of Bé, Bijma, Jentzen, Salmon, Schmuker, Schiebel, etc

R: The sentence was rephrased to improve clarity (page 7, lines 29 - 31).

R2C: Page 8, line 39: why only "thermally constrained"; please discussion with reference to the existing literature (e.g., Jentzen et al. 2018, etc)

R: The sentence was rephrased to make it clearer (page 7, line 31)

R2C: Page 9, line 6: what is meant by majority? Please be specific, and discuss the different species.

R: We have rewritten this section and hope that it is now clearer and more specific (page 7, line 52).

R2C: Page 9, lines 41-42: "G. truncatulinoides replaces G. scitula towards. . ."; please compare to the distribution of G. truncatulinoides and G. scitula in the Azores Front Current System, which is a similar hydrological and ecological setting as studied here.

R: A comparison with the Azores System was inserted in this paragraph (page 8, lines 32 - 34).

R2C: Page 9, line 54: ". . .the result of seasonal superposition of. . ."; please discuss in comparison to earlier papers. You may start from Schiebel and Hemleben, 2000, and Schiebel et al. 2001

R: Citations were inserted in this part (page 8, lines 41 - 45).

R2C: Finally, the chapter 6. Conclusions may be rewritten following the changes in the manuscript.

R: The conclusion were reworded in order to reflect the new analyses with calibrated environmental variables (page 10, lines 8 - 12).

R2C: Some details: Title: I wonder why a rather self-limiting title has been chosen for the much broader topic presented in the paper. I would suggest to skip "Vertical" and make the title "Distribution...".

R: The change was implemented.

R2C: Page 2, line 39: not "cod-end" (which are soft) have been used, but "sampling cups"

**R: The correction was done (page 2, line 43 and elsewhere)**

R2C: Page 2, line 51: I wonder how the nets were changed "manually" at grate depth: I guess that the right expression is "changed by remote control"

R: This paragraph (page 2, lines 48-54 – page 3, lines 1-3) was rephrased.

R2C: Page 3, line 3: pH, not PH Page 3, line 9: skip "planktonic"

R: Since pH was removed from the manuscript, this issue was excluded. Regarding the issue at page 3, line 9, this part of the text is completely rewritten.

R2C: Page 3, line 39: change "concentrations" to "standing stocks"

R: In this segment, we refer to the general concentration calculation, which includes both living and dead shells. Since "standing stock" is only referring to the living shells, we decided to maintain the word "concentration" (page 3, line 27).

R2C: Page 4, line 5: change "trace" to "confirm"

R: This sentence was rephrased (page 3, lines 29 - 31)

R2C: Page 4, lines 42-44: change "first" to "upper"

R: The change was done (page 4, line 13)

R2C: Page 5, lines 18-19: unfinished relative clause: higher than what?

R: The sentence was rephrased for a superlative clause (page 5, lines 38 – 39)

R2C: Page 5, lines 44-45: unfinished relative clause: higher than what?

R: The sentence was rephrased (page 5, lines 9 - 10)

R2C: Page 6, line 44: change "revealed" to "confirmed"

R: The change was done (page 6, line 8)

R2C: Page 7, line 15: (refs Moery etc) is not the correct way of referencing

R: Corrected (page 8, line 41)

R2C: Page 8, lines 29-30: better change "this reference" to "this depth level"

R: The change was done (page 7, lines 22 - 23)

R2C: Page 11, line 9: How should pH affect species distribution? Any data that may support the statement?

R: Since pH was removed from the manuscript, this sentence was excluded.

R2C: Page 13, lines 17-18: not eds. but authors

R: The correction was done (page 13, line 20)

R2C: Figure 4: The upper 260 m max are displayed, not 700 m as stated in the caption.

R: "700" was changed by "550" in order to include the second panel of the figure 4 (Page 17, line 1).

R2C: Figure 9: What shall "light" to "dark" mean in this context? The different parameters from Chl-a to O2 may not be easily put into relation.

R: The figure 9 was modified; they grey scaling represents CCA loading  $\geq 0.4$ .

R2C: Plates: I congratulate the authors on the quality of the light micrographs. However, using a ring light produces light rings on the reflecting surface of chambers. The authors may want to play with more diffused light to produce even better images in the future.

R: We thank the Referee for their comment about our light micrographs. We have done our best to minimize the effect using the current light settings, but because many of the specimens were photographed wet (the specimens were kept deep frozen on the slides) to preserve the color of the cytoplasm, the effect could not be completely avoided. We keep working on other light combinations in order to remove the reflectance effect over smooth tests and hope to find a solution in the future.

R2C: Plate 2: change second (12) to (13)

R: The correction was carried out

R2C: Appendix A: The species descriptions read good in general, but some typos (e.g., page 25, line 6, change "trocospiral" to "trochospiral";

**R: We have done our best to avoid any typos**

R2C: page 29, line 39, change pakerae to parkerae; etc etc) may be corrected.

R: The correction was carried out

R2C: I have no clue what is meant by granules (in T. iota, and T. fleisheri), and pustules may be meant.

R: We changed "granules" for "pustules" along the appendix.

**References**

Brummer, G. J. A., Hemleben, C., and Spindler, M.: Planktonic foraminiferal ontogeny and new perspectives for micropalaeontology, Nature, 319(6048), 50, 1986.

Brummer, G. J. A., & Kroon, D. (1988). Planktonic foraminifers as tracers of oceanclimate history: Ontogeny, relationships and preservation of modern species and stable isotopes, phenotypes and assemblage distribution in different water masses. Free University Press.

Gordon, A. L.: South Atlantic thermocline ventilation. Deep Sea Research Part A. Oceanographic Research Papers, 28(11), 1239-1264, 1981.

Jentzen, A., Schönfeld, J., and Schiebel, R. : Assessment of the effect of increasing temperature on the ecology and assemblage structure of modern planktic foraminifers in the Caribbean and surrounding seas. Journal of Foraminiferal Research, 48(3), 251-272, 2018.

Karstensen, J., Speich, S., Morard, R., Bumke, K., Clarke, J., Giorgetta, M., Fu, Y., Köhn, E., Pinck, A., Manzini, E., Lübben, B., Baumeister, A., Reuter, R., Scherhag, A., de Groot, T., Louropoulou, E., Geißler, F., and Raetke, A. Oceanic & atmospheric variability in the South Atlantic Cruise No. M124. DFG-MARUM, Bremen, 59 p., 2016.

Meilland, J., Siccha, M., Weinkauf, M. F., Jonkers, L., Morard, R., Baranowski, U., Bertlich, J., Brummer, G.-J., Debray, P., Fritz-Endres, T., Groeneveld, G., Magrel, L., Munz, P., Rillo, M. C., Schmidt, C., Takagi, H., Theara, G., and Kucera, M.: Highly replicated sampling reveals no diurnal vertical migration but stable species-specific vertical habitats in planktonic foraminifera. Journal of Plankton Research. Journal of Plankton Research, 00(00), 1-15, doi:10.1093/plankt/fbz002, 2019.

Parker, F. L.: Planktonic foraminiferal species in Pacific sediments, Micropaleontology, 8, 219-254, 1962.

Rebotim, A., Voelker, A. H. L., Jonkers, L., Waniek, J. J., Meggers, H., Schiebel, R., Fraile, I., Schulz, M., and Kucera, M.: Factors controlling the depth habitat of

planktonic foraminifera in the subtropical eastern North Atlantic, Biogeosciences, 14, 827-859, 2017.

**Vertical distributionDistribution** of planktonic foraminifera in the **SubtropicalSouth Atlantic: depth hierarchy of** controlling factors**

5 Douglas Lessa1,2, Raphaël Morard1, Lukas Jonkers1, Igor M. Venancio3, Runa Reuter1, Adrian Baumeister1, Ana Luiza Albuquerque2, Michal Kucera1

1 MARUM - Center for Marine Environmental Sciences, University of Bremen, D-28359 Bremen, Germany

2 Programa de Pós-Graduação em Geoquímica Ambiental, Universidade Federal Fluminense, Niterói, Brazil, 24.020-141

10 3 Center for Weather Forecasting and Climate Studies (CPTEC), National Institute for Space Research (INPE), Rodovia Pres. Dutra, km 39, 12.630-000 Cachoeira Paulista, SP, Brazil

Correspondence to: Douglas Villela de Oliveira Lessa (dvolessa@id.uff.br)

Abstract. Temperature appears to be the best predictor of species composition of planktonic foraminifera communities, making it possible to use their fossil assemblages to reconstruct sea surface temperature (SST)
 variation in the past. However, the role of other environmental factors potentially modulating the spatial and vertical distribution of planktonic foraminifera species is poorly understood. This is especially relevant for environmental factors affecting the subsurface habitat. If such factors play a role, changes in the abundance of deeper-subsurface\_dwelling species may not solely reflect SST variation. In order to constrain the effect of subsurface parameters on species composition, we here characterize the vertical distribution of living planktonic

- foraminifera community across an E-W transect through the subtropical South Atlantic Ocean, where SST variability iswas small but the subsurface water mass structure ehangeschanged dramatically. Four planktonic foraminifera communities could be identified across the top 700 m of the E-W-transect. Gyre and Agulhas Leakage surface faunas were predominantly composed of *Globigerinoides ruber*, *Globigerinoides tenellus*, *Trilobatus sacculifer*, *Globoturborotalita rubescens*, *Globigerinella calida*, *Tenuitella iota* and *Globigerinita*
- glutinata, and only differed in terms of relative abundances (community composition). Upwelling fauna was dominated by *Neogloboquadrina pachyderma*, *Neogloboquadrina incompta*, *Globorotalia crassaformis* and *Globorotalia inflata*. Thermocline fauna was dominated by *Tenuitella fleisheri*, *Globorotalia truncatulinoides* and *Globorotalia scitula* in the western sidewest, and by *G. scitula* in the eastern side of the basincast. The largest part of the standing stock was consistently found in the surface layer, but SST was not the main predictor
- 30 of species composition, neither for the totaldepth-integrated fauna at each stationacross the stations nor in analyses considering eachat individual depth layer separatelylayers. Instead, we identified a consistentpattern of vertical pattern in-stacking of different parameters controlling species composition at different depths, in which the parameters appear to reflect, reflecting different aspects of the pelagic habitat. Whereas productivity appears to dominate in the mixed layer (0-\_\_60 m), physical-chemical parameters are properties (temperature, salinity)
- 35 become important at depth immediately below (60–100 m), followed by parameters related to the degradation of organic matter (100-300 m), intermediate depths and parameters describing the dissolvedin the subsurface, a complex combination of factors including oxygen availability (>300 m), concentration is required to explain the assemblage composition. These results indicate that the seemingly straightforward relationship between assemblage composition and SST in sedimentary assemblages reflects vertically and seasonally integrated
- 40 processes that are only indirectly linked to SST. ThisIt also implies that fossil assemblages of planktonic foraminifera should also contain a signature of subsurface processes, which could be used for paleoceanographic reconstructions.

Key words: plankton net, planktonic foraminifera, species composition, South Atlantic, vertical distribution,
 ecology, micropaleontology, plankton multinets

**1. Introduction The composition of planktonic foraminifera communities in the water column changes in response to key**

properties of their habitat such as water temperature, salinity and food availability (Bé, 1977; Ottens, 1992; Ufkes et al., 1998; Schiebel and Hemleben, 2017). In contrast, the composition of sedimentary assemblages, integrating the seasonal shell flux across years to centuries consistently appears to be best predicted by 5 temperature alone (Morey et al, 2005) and this relationship has been exploited by paleoceanographers to reconstruct past temperature (e.g. Kucera et al., 2005). However, the reason for the dominantstrong relationship of temperature control on with sedimentary assemblage composition remains poorly understood-, especially in the light of species composition data derived from the plankton (Jentzen et al., 2018). If species composition 1( primarily reflects other factors, and the covariance of these factors with temperature differed in the past then D variability of fossil assemblages may not reflect solely temperature variation, violating a key assumption of the transfer function methods used to convert past assemblage composition to temperature (Telford and Birks, 2005; Juggins et al, 2015). Indeed, data from plankton tows and sediment traps show that the living community likely responds to multiple processes in the water column (Ortiz et al, 1995; Schiebel et al, 2001; Field et al., 2006; 1! Storz et al, 2009; Jonkers and Kucera, 2015). This does not mean that the planktonic foraminifera community 5 does not respond to temperature, but it implies that this relationship may be indirect, mediated by temperature control on productivity or other aspects of the foraminifera habitat. The lack of understanding on how exactly the environment shapes planktonic foraminifera communities is a common issue for the choice of variables used to construct forward models (Žarić et al, 2005; Lombard et al, 2009; Fraile et al, 2009; Kretschmer et al, 2016) as 2( well as transfer function models (e.g. Morey et al, 2005; Siccha et al, 2009; Telford et al., 2013). Attempts to disentangle the processes that affect planktonic foraminifera community composition must consider the vertical dimension of species habitat. Observations from vertically resolved plankton nets indicate a vertical 5 spanrange of the living community up to 1000 m with the majority of the standing stock concentrated in the upper 300 m (Bé, 1960; Schiebel, 2002). Because their habitat stretches across such a large depth range of 2! depths, communities at different depths likely should reflect a vertical hierarchy of vertically differentiated controlling processes: light intensity and photosynthesis are only relevant in the photic zone, whereas sinking at depth, processes regulating the export of organic matter budget acts at depth, where degradation activity become 0 important factors leadingfrom the surface layer may control the food that is available to oxygen limitation and

lower water pH. In such a scenario, the community. It is therefore likely that there will not be a universal
 predictor of community composition- across the depth range of their habitat. Instead, we hypothesize that the community will reflect an integrated signal of depth-stratified processes affecting composition in different depth layers.

Planktonic foraminifera counts from stratified plankton net samples coupled with observations of physical and chemical parameters in the water column provide the most direct means to tackle this issue resolve whether or not such a depth-hierarchy of controlling factors exists. Unfortunately, most previous studies based on

not such a depth-hierarchy of controlling factors exists. Unfortunately, most previous studies based on
 foraminifera census counts from plankton nets do not present direct measurements of relevant environmental parameters or such as temperature, salinity, chlorophyll-a and dissolved oxygen (DO), lack an appropriate depth ranges resolution that comprise the all or most of species found in sediment assemblages, or originate from regions where surface and subsurface properties are highly correlated (Bé, 1960; Ottens, 1992; Kemle-von

Mücke and Oberhänsli et al., 1999; Lončarić, 2006; Sousa et al., 2014; Rebotim et al., 2017). In this study, we
 analyze stratified plankton net samples collected in the subtropical South Atlantic, along a transect covering the gradient from productive coastal waters off South Africa to the oligotrophic waters of the subtropical gyre off
 Brazil. The transect has been specifically chosen because it samples a region where SST variability is small but the subsurface water mass structure changes dramatically. differs drastically. We make use of the availability of in-situ vertically resolved measurements of environmental parameters and aim to determine which processes
 explain compositional variability at different depth layers across the transect.

**2. Material and Methods**

The M124 cruise (Karstensen et al., 2016) took place between February 29th to March 16th 2016 on board the RV
 Meteor, sampling across the subtropical South Atlantic (26 to 34.°S and 18.°E to 36.°W) inalong an east – west
 (zonal) transect (Fig. 1a). Planktonic foraminifera were collected at 17 stations using a Multiple Plankton
 Sampler (MPS, HydroBios, Kiel) with 50 x 50 cm opening, 100 µm mesh size and 5 cod-ends, which allowed
 sampling five different depths per haul-nets. Two MPS casts were performed at eachper station, a shallow cast
 with to resolve the planktonic foraminifera standing stock in nine depth intervals-levels (700-500 m, 400-300 m, 300-200 m, 200-100-m, 100-80 m, 80-60 m, 60-40 m, 40-20 m, 20-0 m-0 m) with the exception of the two
 easternmost stations (192 and a deep cast with-202), where the maximum sampling depth intervals 700-500 m,

500 300 m, 300 200 m, 200 100 m, was 100 0 and 500 m. This sampling scheme provides a nine level resolution of the water column. At the shallow station 192, only a shallow MPS cast was done. respectively (Table 1).

For all deployments, the MPS was slacked with all the nets closed to avoid contamination. The slacking was
 done at a speed of 0.5 m/s and stopped when the rope length equaled the lowest depth plus 10 to 20 meters to account for the angle of the rope. The MPS was hoisted at a speed of 0.5 m/s and the successive closing/opening depth level were automatically triggered by an in-house software running under MATLAB 2011b based on the absolute depth determined by the pressure sensor of the MPS. Rough sea was encountered at station 265 and the hoist speed was lowered at 0.3 m/s to reduce the tension on the nets. The triggering was activated 2.5 meters
 before the MPS reached a given depth to account for the time needed for the net to open/close. For 2 MPS

deployments at stations 202 and 306, the opening and closing of the nets was done manually because of connectivity problems between the software and the controlling unit of the MPS. After each haul, the nets were carefully rinsed using seawater. The collected plankton was recovered in the cod ends and brought to the lab and empty cod ends were mounted on the MPS for the next cast. At the end of each station the MPS was carefully rinsed with soft water and the nets were inspected to ensure that they were not damaged during the deployment.

In addition to the MPS sensors that measure the pressure and activate the opening/elosing of the nets, a CTD was mounted on the MPS to measure physical and chemical characteristics of the water column (Temperature, Salinity, Oxygen, Chlorophyll-a, PH) during the cast. The CTD was set on a recording mode to make measurement every second. The CTD was switched on before starting the operation and was running during the whole station. We also obtained sea surface height (SSH) data from SSALTO/DUACS for each station during the sampling day. SSH data was used to recognize the presence of eddies related to Agulhas Leakage in our stations and possible relationship between planktonic foraminifera species and eddies environment.

20

The recovered plankton samples were transferred from the cod ends to glass cups. The cod ends were carefully rinsed with filtered sea water several times to ensure that all planktonic particles were recovered. After having recovered each sample, the cod ends were rinsed thoroughly using freshwater and cleaned in an ultrasonic bath to remove the finest planktonic particles that may have clogged the mesh. The samples were swirled to concentrate the planktonic foraminifera in the middle of the dish and separate them from other zooplankton and organic particles. The planktonic foraminifera were pipetted out on a filter and transferred onto miero paleontological slides with a brush. All small patches of organic matter were also checked to pick exhaustively all foraminifera. When each cardboard was fully covered with foraminifera, the cardboard slides were air dried for at least one hour and stored at -80°C. For all stations, except for the station 192, all the samples have been processed during one working day. The samples of the station 192 were only partially processed on board and the plankton residues were placed into sampling bag and frozen. All samples were kept frozen during their transport back to the Bremen University in Germany.

The picked planktonie The MPS was equipped with a Sea & Sun CTD providing *in-situ* measurements of temperature, salinity and chlorophyll-a concentration. Dissolved oxygen (DO) content was measured during separate CTD casts using a Seabird Electronics sensor mounted on the rosette. The sensor was calibrated on board through comparison with oxygen concentrations determined by titration (Karstensen et al., 2016). Concentration data were averaged for the plankton net intervals and in cases where a plankton haul was not accompanied by a CTD cast (Table 1); values were linearly interpolated along longitude. For the westernmost multinet we used DO data from the nearest CTD cast. Due to a sensor failure DO data are not available for stations 192 and 202. To assess the influence of eddies (a prominent feature of South Atlantic oceanography, Stramma and England, 1999) on the planktonic foraminifera distribution, sea surface height data were obtained

from SSALTO/DUACS for each station at the day of sampling.

All planktonic foraminifera specimens were picked onboard and dried on micropaleontological slides and frozen, with the exception of the station 192. At this station, the samples were partly picked, and the remaining plankton samples were frozen in sampling bag and further processed on shore. All samples were shipped frozen to Bremen University (Germany). A detailed description of the planktonic foraminifera sampling process is given in Karstensen et al. (2016). Planktonic foraminifera were counted and identified to species level following Schiebel and Hemleben (2017). Living Descriptions and dead specimens were recognized by their cytoplasm contentimages of the most abundant species can be found in Appendix A and considered separately and specimens with cytoplasm in the last whorl were considered as living, whereas tests with noneno (or almost none)no) visible cytoplasm in the last whorl, given bydisplaying a distinctive white coloration of the test-wall,

were considered dead. This distinction is likely overestimating the number of living specimens, because the presence of cytoplasm itself does not guarantee that a specimen was alive during collection. This simplification should resultIt results in a slightly deeper estimate of the living depth, caused by dead specimens with cytoplasm being found beneath their original habitat (see Rebotim et al, 2017). In addition, specimens that showed adult

5 morphology were separated from pre-adult ontogenetic stages. The ontogenetic classification of specimens followed the morphological differences between pre-adult and adult stages as summarized by Brummer et al (1986). Species with relatively small morphological differences among the ontogenetic stages (e.g. *Globigerinella calida, Globigerina bulloides, Globorotalia scitula* and *Globorotalia truncatulinoides*) were classified as "pre-adult" when their identification was performed at a magnification higher than 100x and surface

- 10 features typically found in adults (e.g. spines, pustules, large pores) were lacking they lacked morphological features that define the neanic to adult stage transition as proposed by Brummer et al (1986). In many small sized species (e.g. *Tenuitella fleisheri*, *Turborotalita clarkei*, *Turborotalita quinqueloba*, *Globigerinita glutinata* and *Globigerinoides tenellus*), pre-adults could not be identified due to the high morphologiemorphological similarity among them and pre-adults of big species and such specimens were grouped together in the category
- 15 "unidentified juveniles". Initial ontogenetic stages of *Globigerinoides ruber* white and *Globigerinoides elongatus* were lumped together as "*G. ruber* juveniles", because the diagnostic trait of the two species is only observed among adult specimens. Species counts were converted to concentrations using the volume estimated from the thickness of the collection interval. The concentrations of living specimens were subsequently used to calculate average living depth (ALD) and vertical dispersion (VD) following Rebotim et al. (2017).
- 20 The countsMultivariate analyses were used to ealculate concentrations (shells per cubic meter) for each station through the formula

$$\frac{Ci \, (shells/m^3) = Ni/V}{(1)}$$

Where " $C_i$ " is the concentration of the identify co-varying species "i" " $N_i$ " is the number of eounted specimens for the specie "i"(faunas) and "V" is the filtered volume by the plankton net (in m3), to assess which was obtained by multiplying the haul depth and the multiplying area.

25 Concentration values of the "living" category (standing stocks) were used to calculate the average living depth (ALD) and vertical dispersion (VD) using the following equations proposed by Rebotim et al. (2017) in order to determine the preferential depth habitat and the estimated potential vertical range of species.

$$ALD = \frac{\sum (Ci * Di)}{\sum Ci}$$
(2)

 $\langle \alpha \rangle$

$$\frac{VD}{\Sigma Ci} = \frac{\sum((ALD - Di) * Ci)}{\sum Ci}$$

30 Where *Ci* is the concentration of a specie or the total number of foraminifera (shells.m3) and *Di* is the middle value of the depth interval *i*. ALD and VD were calculated only for species with at least five counted shells at a station.

In addition to depth habitat calculations, we performed multivariate analysis in order to trace faunal groups and of the tested environmental variables, which determined determine the spatial and vertical distribution of 35 theindividual species and species communities. Faunal groupsCo-varying species (faunas) were distinguished usingidentified by cluster analysis using the concentration of species transformed to percentages (relative abundance) of species in each depth level, allowing a joint vertical and zonal analysis across the transect. The cluster analysis was carried out using Bray\_ Curtis distance and the unweighted pair-group average (UPGMA) eluster-method-with arithmetic mean algorithm. The relationship between living planktonic foraminifera 40 tospecies concentrations and environmental parameters was determined by a multivariate ordination analysis. We chose the canonical correspondence analysis (CCA);) as the data presents a huge standard deviation, suggestinggradient length indicated that methods for unimodal distribution are more appropriate (standard deviation > 4, Legendre and Gallagher, 2001). For that, the concentration of species was analyzed together with the CTD data. We performed CCAs with two different data matrices: (1) grouped by station (no depth 45 separation) and correlated with environmental parameters The spatial distribution of the planktonic foraminifera

communities across stations was tested against SST and SSS at the surface as well as data on thermocline and maximum (accounting for conditions affecting most of the fauna at each station) and subsurface (250 m, accounting for conditions affecting the subsurface component of the fauna), SSH and mixed-layer depth

(accounting for the vertical structure of the water column) and chlorophyll-level depth (Table 1), (2) separated by-a (accounting for the overall productivity regime at each station). Then, the species concentrations at each

D station were tested separately for each of the nine depth layers and correlated with the average value of CTD variables for each depth section. All multivariate against in-situ SST, SSS, chlorophyll a and DO, always taking parameter values at the center of each depth interval. This analysis excluded the two shallower easternmost stations. Cluster analysis was carried out using the Past 3.16 software (versions 3.16) (Hammer et al., 2001) and CCA analysis were carried out using in R Core team (2019) using the vegan package (Oksanen et al., 2019).

**3. Oceanographic Conditions**

ŗ

- The upper 700 m in our study areastudied region comprises three main oceanographic systems: the Subtropical 5 Gyre (west and east), the Agulhas Leakage and the Benguela Current (Fig. 1). The Western Subtropical Gyre (36 1( - 16-°W) is characterized by high temperature and salinity and low nutrient concentration.concentrations. Recirculation of warm waters from the Brazil and South Atlantic currents occurs in the Western Subtropical Gyre favoring and leads to the accumulation of warm and saline waters resulting in a thick and warm 0 mixed layer (Peterson and Stramma, 1991). In contrast, the mixed layer in the Eastern Subtropical Gyre (16-°W 1!  $-5^{\circ}E$ ) mixed layer is colder. The Agulhas Leakage sector ( $5 - 15^{\circ}E$ ) is characterized by the occurrence of large eddies formed by the interaction between waters leaked from the Agulhas system and waters of the Subtropical Gyre (Fig. 1b), sector (Stramma and England, 1999), which could be viewed is indicated by higher larger SSH anomaly variation anomalies between stations 227 and 202 (7-°E – 15-°E, Fig. 1c, 4). Along the African coast, 5 tongues2a). Tongues of nutrient-rich waters from the Benguela upwelling system are entrained into the Agulhas 2( leakage and associated in westward-moving eddies-(Stramma and England, 1999), \_\_causing elevated productivity
- far offshore in the Agulhas Leakage (Fig. 1b). Finally, the Benguela Current (17 °E) differs from the othersother systems by havingits low temperature, salinity and high productivity due to the Benguela coastal upwelling. It receives contribution of warm waters from the Agulhas Leakage and cold waters from penetrations of the D Subtropical Front (Peterson and Stramma, 1991; Fig. 1c).
- 2! The upper 700 m of the South Atlantic comprises three water masses. The first 100 to 200 m of the water column 5 in the Subtropical Gyre is composed of the Tropical Water (TW) (Stramma and England, 1999). The-South Atlantic Central Water (SACW) is found between 200 and 600 m between in the permanent thermocline (6 to 20 °C isotherm) and halocline (34.3 to 36.0) (Stramma and England, 1999) and its upwelling is responsible for high productivity off Southwest Africa. Finally, the Antarctic Intermediate Water (AAIW) can be observed below the permanent thermocline, between 600 and 1500 m. It has low temperature and salinity, but high nutrient and 3(
- 0 oxygen concentrations (Stramma and England, 1999). The higher oxygen content of AAIW distinguishes this water mass from the oxygen-depleted lower part of the SACW. The CTD profiles taken during M134 confirm the occurrence of these three water masses in the first 700 m of the eruise transect (Fig. 2). Salinity above 36 and high surface temperatures (Fig. 2e2b and 2d2c) indicated the presence of the Tropical Water (TW) that occupied 3! the firstupper 200 m between stations 394 and 344 (36 – 20 °W) and the firstupper 50 m between the stations
- 332 and 252 (16 °W 1 °E). Salinity below 36<del>, slow decrease of temperature</del> and lowera low oxygen 5 entration concentrations indicate the influence of SACW (Fig. 2c, 2d and 242e), which is particularly thick in the center of the transect (16 °W – 1 °E). Between stations 239 and 192 (3 – 17 ° E) low salinity and oxygen (Fig. 2c and 2f) as well as low temperature indicate(Fig. 2b, 2c) indicates a near surface presence of SACW
- upwelling, which is mixing with warm surface waters from the Agulhas Leakage. The AAIW was identified by 4( higherhigh oxygen concentration (Fig. 2f) around 500 m at station 394 (36 °W) and about 600 m2e) and salinity 0 values up to 34.3 that are viewed only in the western and far eastern deepest layers (near to 700 m) at stations-Between 214 and, 239. In the other stations 320 (11 °W) and 252 (1 °E), the SACW was present down to 700 m.
- 5 The surface thermally stable upper mixed layer (SMLUML) occupied the first 30 - 40 m of the water column at all stations (Fig. 2e2b). The SMLUML temperature was higher in the Western Subtropical Gyre, gradually 4١ decreasing towards the east. Below the SML2UML, referred here as lower mixed layer (LML), a sharp temperature decrease is observed until about 100 m, followed by near to constant values until about 200 m, and a slow temperature decrease afterwards. Since the expedition took place during the late summer, the 40 - 100 m
- D sharp decrease of temperature likely represents the seasonal thermocline and the boundary between the mixed 5( layer and beginning of the permanent thermocline layerslayer was therefore defined where the slowlow gradual temperature decreasedecline starts. At stations where the slow-temperature decrease started just below the seasonal thermocline, the mixed layerboundary between the LML and permanent thermocline boundary was placed below the seasonal thermocline. The delimitation of water mass-mixed layer and permanent thermocline
- 5 boundaries across the studied transect is shown in Fig. S1.

The salinity anticorrelated with the temperature with high values in the mixed layer and a permanent halocline at the depth of the permanent thermocline. The salinity maximum was found at the surface, except in the Western Subtropical Gyre, where it occurred between 50 and 100 m. The Dissolved Oxygen (DO) was relatively high along the studied transect with value ranging, and permanent halocline coincided with the permanent

- 5 thermocline at around 200 m. The DO content varied between 4.5 and 96 ml/l and lower with high concentrations found only at station 202 (14 °E). In a DO anomaly plot (Fig. 2f); the highest values were usually in the subsurface, especially in the Western Subtropical Gyre. The lowest anomaly values occurred between 200 and 600 m coinciding with the SACW domain. In the lowest part of the water column, LML and low values below 300 m. The DO content in the surface mixed layer was higher DO anomalies were observed at the Eastern
- Subtropical Gyre than in the Western Subtropical Gyre and Agulhas Leakage regions, demonstrating the presence of the AAIW. In the Eastern Subtropical Gyre, low DO anomalies were observed down to the deepest studied layers, indicating that the SACW/AAIW boundary(Fig. 1e). At the permanent thermocline, DO content was located deeper than 700 m.generally low with lowest values occurring in a broad area of the Western Subtropical Gyre where values below 5.0 ml/l can be seen up to 150 m. The central Subtropical Gyre around 10°W is marked by oxygen concentrations above 5.7 ml/l. Towards the eastern Subtropical Gyre, oxygen concentrations decrease, but remain higher and more variable than in the west.

The highest chlorophyll-a concentration was higher concentrations (63 to 77 µg-/l+)) occurred between stations 214 and 192 (11 – 17 °E) with the Deep Chlorophyll Maximum (DCM) at about 50 m-(Fig. 2d). Moderate chlorophyll-a concentrations (55 – 65 µg-/l+) and a DCM between 50 and 100 m was seen at stations 265, 229239 and 227 (3 – 7 °E). The other stations had lower values ranging from 38 to 45 µg-/l+ and a DCM below 100 m. High chlorophyll-a values in the east are associated with the Benguela upwelling system (station 192) and its tongues that mix with oligotrophic waters of Agulhas Leakage (stations 214 and 202). On the other hand, lowLow chlorophyll-a concentrations in western and central South Atlantic are associated with strong stratification of the Subtropical Gyre. Surface water pH varied in opposite direction to primary productivity with high values (up to 8.8) on the western side and lower (down to 8.3) on the eastern side. Lower pH values were observed in deeper layers, especially in the east where values ≤ 8.0 were observed below 500 m.

**4. Results**

4.1. Concentration and distribution of living and dead planktonic foraminifera

We identified 38 species of planktonic foraminifera of which 22 species yielded five or more individuals per 30 station, which allowed allowing an the analysis of their habitat depth and the zonal variation (Table 2, Fig. 4): Appendix A, Plates 1 – 5 in the supplementary material). We observed high standing stocks of Globigrinoides Globigerinoides ruber, (pink and white), Globigerinoides elongatus, Trilobatus sacculifer, Globoturborotalita rubescens, Globigerinoides tenellus, Tenuitella iota and Globigerinita glutinata in the mixed layer of the Subtropical Gyre and Agulhas Leakage region. In the Benguela Current and the subsurface sector of 35 the Agulhas Leakage area, we observed high abundances of Neogloboquadrina dutertrei, Neogloboquadrina pachyderma, Neogloboquadrina incompta, rounded specimens of Globorotalia crassaformis and Globorotalia inflata. In contrast, the The water column below 100 m (permanent thermocline layer) in across the whole transect had high abundances of Tenuitella fleisheri, Globorotalia truncatulinoides (left coiling) and), Globorotalia scitula and Hastigerina pelagica. Apart from the Benguela and permanent thermocline dwelling species, some 40 other species also had restricted distributions. The pink morphotype of G. ruber pink was found only in the Western Subtropical Gyre, Candeina nitida occurred between the stations from station 356 (Western Subtropical Gyre) and to 265 (Eastern Subtropical Gyre), Globorotalia menardii occurred in high abundances in the first 100 m at the three western stations (394, 382 and 370), and conical specimens of G. crassaformis were observed rarely in the permanent thermocline of the Western Subtropical Gyre.

45 The total concentrationConcentrations of planktonic foraminifera was higher at stations 332, 344, 356 and 370 with values up to 12increased from around 10 shells.m-3. However, if we consider concentration values/m3 in the upper 100 m, high concentrations occurred only in the western stations (Fig. 3a), with up to east to a maximum of 75 shells.m-3 in the surface layer of station 370 compared to 25 shells.m-3 in the surface layer/m3 in the central part of the station 192. Western Subtropical Gyre (Fig. 3a). Concentrations of living specimens were always

- higher in the upper 100 m of the water column (Fig. S2), with a gradual increase in the proportion of dead (empty shells) specimens below 100 m and almost no living specimens occurring below 500 m. The lowest concentrations (living and dead specimens) were observed in the deepest sections of stations 394, 370 and 227 that were under influence of AAIW. Pre-adults. Pre-adult specimens were abundant in central gyre stations (Fig. 3), where station 306 (11 °W) had almost 100 % neanic and juvenilepre-adult specimens in the upper 40 m (Fig. 3).
- 55 S3). The number of adults tended to increase  $\frac{\text{downward}\text{with depth}}{\text{depth}}$  (> 40 m depths) in relation to with pre-adults

that werebeing virtually absent below 100 m-depth. The exception was Globorotalia crassaformis in the eastern stations. The concentrations of living adults were higher than those of living pre-adults (near 100 % adult) at stations 214 and 192, where most of the fauna was composed consisted of cold-water species N. pachyderma, N. incompta, G. inflata and G. crassaformis.

[revised manuscript text omitted]

45 The community variation across the 4.2. Subtropical South Atlantic was analyzedplanktonic foraminifera **communities** The distribution of species ALDs across the stations indicates the presence of three groups of species occupying different portions of the water column (Fig. 6). The upper 100 m layer is inhabited by elusterupper and ordination multivariate analyzes. The lower surface-dwelling species, separated from each other more distinctly 50 in the center of Subtropical Gyre than in the Benguela region. This pattern was confirmed by the cluster analysis of the community composition (relative abundance without cut level) for each depth and station-revealed,

confirming the presence of twelve clusters, composed of sevenfive principal faunas, consistently separated by region and depth (Fig. 7). These were the Subtropical Gyre fauna that was further divided in surface, (clusters 4 and 6), western subsurface (cluster 5) and eastern subsurface; (clusters 7 and 9); the Agulhas Leakage fauna; 55 Thermoeline (cluster 8); the permanent thermoeline fauna (clusters 2, 10, 11 and 12) that was further divided in

western (clusters 2 and 12) and eastern; (clusters 10 and 11); and finally the Benguela fauna; (clusters 1 and 3). The average relative abundances for the most important species in each cluster are summarized onin Table 3. This reveals that the warm and oligotrophic areas are inhabited by the same species, but that their proportions vary. Thus, the The subdivision of the Subtropical Gyre fauna reflects increased abundance of *G. rubescens* and

5 G. tenellus in the west and G. glutinata in the east and the. The Agulhas Leakage fauna iswas characterized by a higherhigh contribution of T. sacculifer. In contrast, Thermoeline, and the presence of G. rubescens and G. tenellus. The Agulhas Leakage fauna was only present in the UML. At the LML, species belonging to the Benguela faunal cluster dominated the assemblages. Permanent thermocline and Benguela communities comprised distinctly different species assemblages. Thermoelinegroups. Permanent thermocline fauna is

characterized by *G. scitula*, *T. fleisheri* and *G. truncatulinoides*, whereas Benguela upwelling fauna is
 characterized\_defined by *G. crassaformis* and *N. pachyderma*. In contrast to the Subtropical Gyre stations, where
 surface and subsurface faunas were recognized, the Agulhas Leakage lower mixed layer was occupied by the The
 Benguela fauna also contained small contributions of *G. inflata*, *N. incompta* and *G. bulloides* that are besides
 upwelling fauna (Fig. 7b), which could beconditions also associated to high levels of chlorophyll *a* (productivity) due to unwelling filaments (Fig. 2) with the Subtropical Front. The deepest samples intervals at

15 (productivity) due to upwelling filaments (Fig. 2).with the Subtropical Front. The deepest samplesintervals at stations 394, 370 and 227 had too littlefew or no living planktonic foraminifera and, therefore, could not be clustered.

To identify which variables or 4.3. Variables and processes were associated with controlling the observed
 planktonic foraminifera distribution across the subtropical South Atlantic, we performed canonical
 correspondence analyses (CCA) in two ways: (1) a station separated analysis comprising species concentrations and environmental parameters (temperature, salinity, dissolved oxygen and pH at 30 and 200 m of depth, and total chlorophyll-*a* concentration) with collapsed depth sections (Fig. 8); and (2) station separated analysis of species concentrations and environmental parameters for each single depth interval (Fig. 9). The first, The depth-integrated3 CCA shows that most68 % of the variability in species composition along the transect can

- 25 be constrained by the tested environmental factors, with more than half of the total inertia being explained with two CCA axes (Fig. 8). The first axis (43 % of the total inertia explained 85 %). Most) orients most species are oriented along a productivity (chlorophyll a)-temperature gradient, with the Benguela upwelling species on the high productivity end and the warm surface and gyre subsurface groupsspecies on the other; G. scitula and G. ruber pink, T. iota plot on opposite ends of the warm oligotrophic end. The second axis (Fig. 8). Such an
- 30 important role of productivity11% of the total inertia) appears to contrast with global studies that have documented temperature as the most important predictor of foraminifera assemblages (refs Moery etc). However, in our area SST and chlorophyll *a* (productivity) are anticorrelated, since low temperatures mean less stratification, which causes enhanced nutrient availability at the photic zone. Moreover, temperature influences respiration and growth rates, rendering be related to the influence of temperature and productivity difficult to
- separatedepth of the mixed layer and is responsible for the remaining variation in the non-upwelling faunas. It orients *G. scitula* at the extreme positive end opposite to *G. ruber* pink, *G. calida* and *G. conglobatus*. Due to the distinct foraminiferal community in the Benguela upwelling system, station 192 stands out in relation to the other stations and hence may be responsible for a large proportion of the variability. If the station 192 is removed from the Therefore, we carried out an additional analysis (Fig. S4), the mentioned anti-without this station, revealing that the negative correlation between SST and productivity is still evident, influenced by stations in remains at the ordination along the tested environmental variables still explains more than half of the variability in species composition (Fig. S4), reflecting the fact that the Agulhas Leakage domain that wereis also influenced by cold and productive waters below 40 m during the sampling time. However, the explained variance of the
- by cold and productive waters below 40 m during the sampling time. However, the explained variance of the first axis decreases to 41%. Thus, the first principal component axis (Fig. 8) can be linked to productivity modulated by weak water stratification or upwelling.

The second axis, which explains 15% of the observed variance (Fig. 8), separates species with distinct depth habitats and vs temperature, salinity and pH at the thermocline layer. Most of explanation for the second factor is influenced by *G. scitula*, which had the deepest ALD (234 m). In this study, *G. scitula* was significantly abundant in the whole transect with its highest concentrations occurring at stations with a higher and shallower ehlorophyll-a maximum (Fig. 5). On the opposite side of the axis, *G. ruber* pink was classified as upper mixed layer dwelling (Fig. 5) and *T. iota* was classified as a whole mixed layer dwelling in this study with ALD down to 37 m. Both *G. ruber pink* and *T. iota* concentrations were higher in highly stratified areas with low ehlorophyll a concentration. The scatter suggests that living depth can be linked to this axis and that different vertical patterns of environmental parameters play a very important role in determining the foraminiferal community. That vertical gradient describes the faunal variation of the community can be linked to organic

matter budget from mixed layer to permanent thermocline , whose amount depends to local productivity. High

sinking organic matter budget increases the microbial respiration rate that has influence over pH at thermocline. These different vertical patterns of environmental parameters play a very important role in determining the foraminiferal community in both surface and thermocline layers.in the Subtropical South Atlantic even outside of the Benguela upwelling.

- 5 Considering the importance of vertical gradients in environmental variables for the distribution of planktonic foraminifera in the subtropical South Atlantic, CCA(s) for individual depth intervals were performed (Fig. 9 and Table S1). Those CCA showed four hierarchical changes of processes that define planktonic foraminifera community composition. For the first three depth sections (0 20, 20 40 and 40 60 m), chlorophyll a was the most important variable explaining between 40 and 70 % of the variation, indicating that productivity is the most 10 important factor in the subtropical South Atlantic. Physical-chemical parameters (temperature, salinity and pH) explained most of the variation of the community in the 60-80 and 80-100 m depth intervals (60 and 40 % of inertia explained, respectively). For the 100 200 and 200 300 m depth sections, the community was
- predominantly explained by pH variations with a low, but downward increasing, contribution of dissolved oxygen (50 and 60 %, respectively). This suggests that the species distribution was influenced by degradation of 15 organic matter produced in the surface layer since microbial respiration contributes to pH and dissolved oxygen variability in mesopelagic waters. It is interesting to note that the secondary contributor variables change from temperature and salinity at 100 - 200 m section to dissolved oxygen at 200 - 300 m section. This last
- environmental parameter reached the highest contribution at 300 500 m (50 % of the variation), which could be linked to the oxygen minimum layer. Even though chlorophyll a appears as the most important variable at the 20 deepest interval (500 - 700 m), species standing stocks were too low to define a significant pattern in the community distribution. Besides, the chlorophyll-a concentration as a main controlling factor at this depth seems unrealistic, since photosynthetic activity is greatly reduced below 200 meters. separate CCA(s) were performed
- on faunas from the nine depth layers (Fig. 9, Table 4 and Fig S5). These analyses revealed a pattern of vertical stacking in the relative contribution of the tested environmental factors on the planktonic foraminifera 25 community composition. In the surface layer (0-60 m), chlorophyll-a shows the highest absolute loading on the
- first CCA axis, and the four tested environmental variables together explain up to 57 % of the inertia in the fauna. Below 60 m, the tested variables explain less than 45 % of the total inertia, chlorophyll-a concentrations cease to be the most important variable and temperature becomes more important. The temperature-chlorophyll gradient, identified as the dominant control of the total faunal composition across the stations (Fig. 8), is
- 30 manifested in the depth-resolved CCAs only between 20 and 80 m. It is not present at the surface (where chlorophyll a dominates) and below 100 m. Instead, from 80 m, dissolved oxygen concentration start explaining a considerable amount of variance in the fauna and between 80-100 m and 300-500 m it appears most important. Temperature and salinity are the most important variables between 100 and 300 m, where the species are ordered along a temperature-oxygen gradient. Counter-intuitively, the faunal variation at the deepest level (500-700 m) 35 appears to be best explained by chlorophyll-a concentration, but we note that the analysis at this depth level is strongly affected by the small number of living specimens and the resulting pattern should be interpreted with

**5. Discussion**

caution.

**5.1. Synchronized reproduction and ontogenetic vertical migration**

- 40 Before interpreting the vertical distribution of planktonic foraminifera in the water column as a function of changing environmental properties across the studied transect, it is important to evaluate the possible effect of reproductive processes on species concentration and living depth. The M124 cruise lasted for 16 days with new moon occurring on the 10th day of the cruise and the preceding full moon occurring 7 days before the sampling at the first station. If some of the species reproduced consistently in phase with full moon (Spindler et al., 1979;
- 45 Jonkers et al., 2015; Venancio et al, 2016) then the proportion of pre-adult specimens should have been higher during the first half of the cruise. Three species show elevated abundance of pre-adults consistent with such a pattern: G. ruber white, O. universa and G. calida (Bijma et al, 1990; Fig. S3). The remaining species show an even proportion of pre-adults throughout the cruise, which is not consistent with synchronized reproduction. For the three species that may have reproduced at full moon, we evaluated by periodic regression analysis whether
- 50 the living depth of the populations could show signs of an ontogenetic vertical migration - a However, when analyzed statistically, G. ruber white, O. universa and G. calida showed no systematic change in depth habitat with progressive maturity of the reproductive cohort. We observe neither that their ALD is correlated withrelationship between the proportion of pre-adults nor that there is any systematic change of ALD of these species during the and lunar cycle. Taken together, the relatively constant proportions of pre-adult and adult 55
  - specimens-day, nor a periodic change in their ALD in the majority of the analyzed species speakphase with the

lunar cycle, which speaks against a strict synchronization of their reproduction<del>, and even in the three species</del> where the proportion of pre-adults was higher during one part of the lunar cycle, there was and provides no evidence for ontogenetic vertical migration. InHence, in the absence of a strong evidence for the control of reproductive or ontogenetic processes on the vertical habitat, and ruling out an effect of daily vertical migration

5 based on detailed observations elsewhere (Meilland et al., 2019), we proceed by interpretinganalyzing if the ALD of the individual species and analyzing if the ALD varied predictably as a function of environmental parameters.

**5.2. Species vertical distribution across the Subtropical South Atlantic**

30

10 The ALD of planktonic foraminifera species shows a consistent depth ranking pattern and a mixture of stable and variable depth habitats (Fig. 5). Since the main feature of the water column relevant for the vertical distribution of non-motile plankton is the mixed-layer depth, we consider the observed ALD against this referencedepth level (Fig. 6). First, we observe a group of species whose ALD was consistently in the Upper Mixed Layer (UML). Species in this group had an ALD shallower than 40 m and a low VD (up to 30 m) that 15 usually was belowdid not range deeper than 40 m. This depth range corresponds to the extent of the warm thermally mixed surface layerUML (Fig  $\frac{22c}{2}$ ). The most abundant species in this group were G. ruber (pink and white), G. conglobatus, O. universa and T. sacculifer. These species all bear algal endosymbionts (Takagi et al., 2019) and), which likely explains their observed habitat is consistent with the high abundances in the high light and low nutrient conditions inof the SML. TheUML, Such a shallow depth habitat of G. ruber was also observed 20 by Berger (19691968) and Rebotim et al. (2017), but the observed ALDALDs for O. universa and G. conglobatus and partly also for T. sacculifer iswere shallower in the South Atlantic than in many other studies/regions, the North Atlantic (Schiebel et al, 2002; Rebotim et al., 2017; Jentzen et al, 2018), indicating that globally this species has a more variable habitat than observed in the studied section. Clearly, Overall, our results confirm observations elsewhere that the dominant habitats of these species are not reaching below the seasonal 25 thermocline, indicating that they may be thermallyare constrained to the uppermost summer mixed layer, as light is not limited. photic zone within the UML (Kemle-von-Mücke and Oberhänsli, 1999; Kuroyanagi and Kawahata, 2004: Rebotim et al. 2017: Jentzen et al. 2018).

A second group of species also inhabits preferentially the upper water layer (ALD usually aboveshallower than 50 m), but their larger VD (dispersionup to 50 m) indicates that their vertical habitat comprises the Whole Mixed Layer (WML). The most abundant species in this group are *C. nitida*, *G. glutinata*, *G. calida*, *T. iota*, *G.*

*rubescens, G. tenellus* and *G. menardii.* With the exception of *T. iota*, these species alsoall bear algal endosymbionts (Takagi et al., 2019), which is consistent with their habitat within the photic zone, but they. However, in contrast to the UML species group, these species appear less tightly linked to the surface layer, implying either a broader thermal tolerance or adaptation of their symbionts to lower light levels. Whereas for most species the observed habitat is comparable with previous work (Rebotim et al., 2017, Kemle-von-Mücke and Oberhänsli, 1999; Rebotim et al., 2017), the shallow habitat of *T. iota* is at odds with its concentration maximum around 300 m in the NE Atlantic reported by Rebotim et al. (2017). Clearly, the ecology of this species requires further investigation. For the remaining species of the UML and WML groups, it is clear that their habitat is above the deep chlorophyll maximum, suggesting that their vertical distribution is influenced by other parameters than the availability of fresh phytoplankton.(2017), indicating a highly variable depth habitat that requires further investigation.

In contrast, species of the **Lower Mixed Layer (LML**) have a habitat that is still dominantly within the seasonal thermocline (above the permanent thermocline), but whose vertical distribution overlaps with the deep chlorophyll maximum (Fig.  $\frac{22c}{2}$ ). The species in this group hadexhibit an ALD between 50 and 100 m and a VD

- similar to the UML group. The most abundant species in this group were *G. elongatus*, *N. dutertrei*, *N. pachyderma*, *N. incompta*, *G. crassaformis* and *G. inflata*. With exception of *G. elongatus*, these species occurred in the Eastern Subtropical South AtlanticAgulhas Leakage and Benguela stations, indicating that this group may respond primarily to variations in productivity, since here the chlorophyll a concentration was higher, the DCM shallower and the water column less stratified than on the western side of the gyreat these stations (Fig. 2). This is consistent with the majority of these species being non-symbiotic (Takagi et al., 20192c). At station
- 192, which is influenced by Benguela upwelling, the planktonic foraminifera community was dominated by the LML group at all depths.vertical levels. The species *G. elongatus* was the only LML elassified species in the Subtropical Gyre. This species was the most abundant form of the *Globigerinoides* plexus, but we caution against a too strict interpretation of its apparently substantially deeper habitat than the remaining *G. ruber* morph

typesother from the *Globigerinoides* plexus because thesethe species cancould only be distinguished in theirits adult stagesstage (Aurahs et al., 2011). Morard et al., 2019), and its apparently deeper habitat may reflect the habitat of adult specimens only. Nevertheless, the observed depth stratification between *G. elongatus* and *G. ruber* (white) is consistent in sign with previous studies based on observations in the plankton (Kuroyanagi and Kawahata, 2004) and oxygen isotope and Mg/Ca ratios (Wang, 2000, Steinke et al., 2005). Thus, at least in

summer, adult G. elongatus live below the warm and stable SMLUML in the South Atlantic.

5

A Species of the **Thermocline** group show a distinctly different ALD distribution, with the largest part of the population livingoccurring below the mixed layer was shown by species of the **Thermocline** group seasonal thermocline. Species in this group have a variable ALD within the permanent thermocline (belowdeeper than

- 10 100 m) and do not show a clear relationship withthat is in general independent of the position of the DCM. In fact, most of their populations occur often below the DCM. The most abundant species belonging to this group are *T. fleisheri*, *G. truncatulinoides*, *G. scitula* and *H. pelagica*. Thermocline dwelling species represent a clearly defined clustergroup distinct from other species assemblages (Fig. 6), suggesting that they constitute a distinct community that may not be shaped by surface processes. Since within their habitat photosynthesis is inhibited
- 15 due to insufficient light, within their habitat, species of the thermocline species group are likely to feed on either zooplankton or sinking organic matter (Schiebel and Hemleben, 2017). Thus, species of the thermocline group may respond not only to productivity that supplies food, but also to processes related to organic matter degradation below the DCM, as dissolved oxygen concentration and pH are influenced depending on the organic matter budget. Little is known about the ecological preferences of *T. fleisheri* because this species is small and
- 20 hence often overlooked since most studies only investigate the specimens >150 μm. Rebotim et al (2017) classified *T. fleisheri* as a surface to subsurface dweller. However, the species was rare in their study, rendering this classification uncertain. In our, transect, *T. fleisheri* was abundant between 100 and 300 m at all Subtropical Gyre stations with specimens up to 200 μm in size (see platePlate 5, 11-13). Within the Subtropical Gyre, the ALD of *T. fleisheri* was close to the DCM in ten out of 13 stations within the subtropical Gyre, suggesting a link between depth habitat and the depth of the chlorophyll maximum for this species (Fig. 5).
- \_The ALD of *G. scitula* was the deepest (235 m) and most variable (VD 109 m) in our study. The mesopelagic habitat of *G. scitula* has also been observed in others regions such as Atlantic and Indian Oceanselsewhere (Bé and Tolderlund, 1971; Itou et al. 2001; Field, 2004; Storz et al. 2009; Rebotim et al. 2017<del>), NE Pacific off California (Field, 2004) and Western North Pacific off Japan (Itou et al., 2001).</del> Jentzen et al. 2018). This
- demonstrates that the species is a elearconsistent permanent thermocline dweller, living below the deep chlorophyll maximum, where it must feed on sinking organic matter. The species *G. truncatulinoides* had an ALD near 170 m with a relatively low VD (55 m), which associates it to the lower part of the DCM and the permanent thermocline below. The cytoplasm color of *G. truncatulinoides* was similar to *G. scitula* (see platesPlates 4 and 5 in the supplementary material) perhaps indicating a similar diet. TheHowever, the spatial distribution of the two species shows that *G. truncatulinoides* replaces *G. scitula* towards the west at stations in
- the Subtropical Gyre. Dissolved oxygen and pH were : similar to the most zonally variable environmental parameters withinAzores Current System where *G. truncatulinoides* linked to North Atlantic Subtropical Gyre is replaced by *G. scitula* when transitional waters from the permanent thermoeline layer (Fig. 2). Consequently, the zonal distinction of the two species may be due to variations of processes expressed in these two environmental parameters. Azores Front migrated southwards (Storz et al, 2009). The habitat of *H. pelagica* is dominantly subsurface but highly variable, likelyperhaps reflecting the presence of both the surface and the subsurface
  - genetic types of this species (c.f. Weiner et al., 2012).

**5.3. Depth hierarchy of relationships between planktonic foraminifera and environmental parameters**

Plankton net samples represent a snapshot of the plankton state in an exact placespace and during an exact time.
 In this way they allow us to observe the direct response of the plankton community to environmental parameters. This is fundamentally different from relationships extracted from sedimentary assemblages, which represent long-term (years to millennia) integrated fluxes of species (Jonkers et al., 2019). Since temperature consistentlyTemperature appears to be the single and dominant parameter explaining variation in community composition in sediment assemblages (Morey et al., 2015), our results can be interpreted as evidence for the effect of temperature on 2005). However, sedimentary assemblages being the result of from vertical and temporal (seasonal-superposition of assemblages driven by a more diverse set of) integration of assemblages, and the effect of temperature on their composition may therefore be indirect. Indeed, our results confirm earlier suggestions (Schiebel and Hemleben, 2000; Schiebel et al, 2001) that abiotic and biotic parameters, varying seasonally and, as our analysis shows, also with depth.

A consideration of other than temperature play an important role in explaining the distribution of planktonic foraminifera in the water column. Both the habitat depth <del>and their</del> variability among species (Fig. 5, 6) as well as the distribution of the species along the transect (Fig. 7), provides clear hints for the effect of multiple environmental parameters on the habitat and the abundance of the species. <del>This observation is</del>These hints are reinforced by the results of the CCA (Fig. 8). Since), where the foraminiferal communities showed a strongerfirst axis sorts the community according to the productivity in the mixed layer. The subtropical gyre stations are

further separated along the second dimension of the CCA, which appears to be related to the vertical than horizontalgradient in the water column. This separation is mirrored by a distinct grouping of species characterizing the end-members of the CCA gradient (Fig. 8). SSH does not appear to systematically affect the species distribution, indicating that the observed spatial pattern and many of the considered environmental parameters only varied at certain depths, weis not affected by the presence of eddies or Agulhas Rings with distinct fauna (Peeters et al., 2004) carried out a series of CCAs far into the South Atlantic.

5

The analysis of species distribution along the transect (without the two easternmost stations) separately for each of the nine depth intervals (Fig. layers provides further details on how the vertical structure of the water column affects the assemblages. This analysis reveals9). These analyses reveal the presence of a vertical

15 affects the assemblages. This analysis reveals9)-These analyses reveal the presence of a vertical successionhierarchy of environmental parameters drivingaffecting community composition7 (Fig. 9), indicating that the vertical changes in the community composition shown by the cluster analysis (Fig. 7) may be the consequence of vertically varying influenceimportance of different environmental parameters.

20 Because of its vertically stacked design, the analysis in Fig. 9 must not be interpreted as an explanation for the vertical succession of species habitats (Figs 5, 6). Instead, it explains which variable controls the species composition within each depth interval. For example, the factors. The UML fauna inhabits the surface likely because of its has an affinity to high light and high temperature, but variations in the composition of the communities in this layer across the studied transect seem to be driven by chlorophyll concentration at the

- 25 surface. This may hint at differences in the adaptation of the UML species with respect to productivity. Remarkably, below 60 m, i.e. at the depth where the WML and LML fauna occur, temperature, salinity and pH explained most of the variation of the community within the layer. This likely reflects the effect of the thickness of the seasonal mixed layer, which is most expressed at those depths (Fig. 2). Below 100 m, the communities appear to be most influenced by the rate of degradation of organic matter produced in the mixed layer, which is
- 30 reflected by pH and dissolved oxygen concentration. This is logical, since these communities are dominated by asymbiotic species that live below the permanent thermocline, where temperature variations are subdued. We note that the distribution of these species appeared not to be linked to the deep chlorophyll maximum and that variability in chlorophyll *a* concentrations do not explain much of the variability in species composition at those depths. Dietary preferences of these species are poorly known, although indirect observations on
- 35 *Neogloboquadrina* indicate an affinity to marine snow (Fehrenbacher et al, 2018). Our observations also suggest that, rather than feeding on fresh organic matter, the species at these depths feed on degraded organic matter. Below 300 m, oxygen concentration shows highest explanatory power (50 % of variation), which could be linked to direct effect of low oxygen on metabolic processes. Under such scenarios, the small and flat *G. scitula* should be better adapted to oxygen limitation than *G. truncatulinoides*, which is consistent with the observed
- distribution of these thermocline dwelling species (Fig. 4).the uppermost layer along the studied transect seems to be driven by chlorophyll-*a* concentration. This may hint at differences in the adaptation of the UML fauna with respect to productivity. From the LML, the influence of physical oceanographic parameters on the composition of the WML and LML faunas become visible, with temperature and salinity displaying higher loadings on the first CCA axis. The importance of chlorophyll-*a* concentrations is reduced below 60-80 m and the variation in the LML and Thermocline faunas is less well explained by the CCA models (inertia explained dropping to below 40 %) and requires combinations of multiple variables, including DO, with temperature and salinity linked to DO and chlorophyll-*a* with different sign at different depth levels.

Considering the actual DO values, the role of this parameter must be indirect, as O2 concentrations throughout the studied water column are far above levels at which foraminifera species begins to display differential
 adaptations (~3.5 mmol/l for benthic foraminifera, Kaiho, 1999; see also Kuroyanagi et al., 2013) and therefore cannot have a direct physiological effect. The relationship therefore likely arises from collinearity with other environmental and/or biotic factors. Indeed, the apparent importance of chlorophyll *a* at depths where it no longer can reflect primary production and the effect of DO at concentrations which cannot incur a direct physiological effect both point to processes related to food availability. Such processes (e.g., aggregate composition and abundance) are not captured directly by any of the tested variables and the presence of

additional, untested controls on the assemblage is indicated by the lower portion of the faunal variance explained by the ordinations at those depths (Fig. 9). TheWe therefore infer that abundance and quality of food, and perhaps the interaction with other plankton, likely play an important role in shaping planktonic foraminifera community within the permanent thermocline. Dietary preferences of thermocline-dwelling species are poorly known, although indirect observations on *N. dutertrei* indicate an affinity with marine snow (Schiebel and Hemleben, 2017; Fehrenbacher et al, 2018).

Whatever the exact environmental and biotic controls on community composition at depth may be, the existence of a vertical succession of parameters best explaining constraining community composition of planktonic foraminifera implies thatadds a new dimension to ecological models derived from sedimentary assemblages integrate seasonally and vertically separate assemblages, whose composition is driven by different processes. It

- 10 integrate seasonally and vertically separate assemblages, whose composition is driven by different processes. It means that SST reconstructions using census counts of planktonic foraminifera (e.g. (transfer functions) may not directly reflect temperature, as well as the inverse efforts to model planktonic foraminifera assemblages from upper ocean properties. Our results reinforce the notion that even though statisticallynear sea surface temperature appears the most important explanatory variablebest predictor of sedimentary species composition (Telford et al)
- 20 truncatulinoides along the transectobserved between eastern and western thermocline faunas is clearly not driven by temperature (Fig. 2; 7) and if this replacement is preserved across the seasonal cycle, it could represent a powerful proxy for organic matter degradation below the surfacebe an indication for the properties of subsurface waters in the South Atlantic.

**6. Conclusions**

35

40

45

50

- 25 We investigated the zonal and vertical distribution of planktonic foraminifera at 1417 stations across the subtropical South Atlantic region-during the late austral summer 2014. Using in-situ vertically resolved environmental data from, CTD profiles and satellite observations, we accessedassessed which factors drive the observed foraminifer species distribution:
- Species specific standing stocks varied regionally, with the most pronounced differences among
   communities observed between the Benguela (station 192) and Subtropical Gyre (stations from 394 to 239) and with an intermediate community in the Agulhas Leakage region (stations from 227 to 202).
  - The highest standing stock was observed in the upper 60 m of the water column, whereas the numbers of dead specimens (no cytoplasm) -increased below 100 m. The highest concentrations of planktonic foraminifera occurred at stations in the oligotrophic western Subtropical Gyre, indicating that the total standing stock is not positively correlated with productivity during the summer.
- The species G. ruber, G. calida and O. universa had a high number of pre-adults consistent with reproductive cycle at full moon. However, the average living depth (ALS) of those species did not show a significant lunar periodicity, suggesting that environmental factors are the prime drivers of their depth habitat variability.
  - We observed no strong evidence for synchronized reproduction affecting the species distribution, and a distinct pattern of species-specific habitat depth, resulting in the existence of distinct vertically stratified faunas.
    - The permanent thermocline layer (200 700 m) hashosts a planktonic foraminifera community distinct from -the mixed layer. In the western South Atlantic, high abundances of *G. truncatulinoides*, *T. fleisheri* and *G. scitula* were observed, whereas *G. scitula* dominated the community in the eastern South Atlantic. Zonal differences in the mixed layer communities were less pronounced.
      - TheOverall, the zonal distribution of species was primarily affected by the inverse relationship between chlorophyll-a and water temperature, and secondarily by the amountvertical structure of exported organic matter from the mixed layer to thermocline that reflects specially pH differences between western and eastern South Atlantic within the thermocline layerwater column in the Subtropical Gyre.
  - The vertical distribution of planktonic foraminifera showed a clear depth-dependent hierarchy in the environmental parameters explaining abundanceassemblage variability. The variability in the upper 4060 m was mainly influenced by productivity (chlorophyll-a-concentration (productivity), followed). Below 60 m, the community composition was more difficult to explain by physic-chemical the tested

variables (and we infer that next to temperature, salinity and pH) between 40 and 100 m. Below 100 m, variables related to microbial respiration (and salinity) the abundance and quality of sinking organic matter were the mostfood, or other biotic interactions, likely played an important to determine species distribution with pH influencing between 100 and 300 m and dissolved oxygen between 200 and 500 m role.

5

Overall, planktonic foraminifera communities of the subtropical South Atlantic seem to respond to a horizontally and vertically variable combination of environmental parameters. This should be taken initiate account when interpreting sedimentary assemblages for paleoceanographyand in efforts to model foraminifera production from environmental parameters.

**10 7. Data availability**

The foraminifera concentration data and the accompanying environmental data from CTD casts have been deposited onat PANGAEA (www.pangaea.de).

 CTD data from the M124 cruise is available on PANGAEA: https://doi.pangaea.de/10.1594/PANGAEA.895426
 and https://doi.org/10.1594/PANGAEA.863015. SSALTO/DUACS sea surface height data were obtained from: https://www.aviso.altimetry.fr/en/my-aviso.html.

Competing interests: The authors declare that they have no conflict of interest.

**8. Author contributions**

20 Douglas V. O. Lessa analyzed the planktonic foraminifera fauna and was responsible for the manuscript writing.

Raphaël Morard was responsible for collection of planktonic foraminifera during the cruise and contributed to the oceanographic data analyses.

Lukas Jonkers contributed to the statistical analyses.

Igor M. Venancio contributed to the interpretation of the lunar periodicity on planktonic foraminifera species.

25 Runa Reuter and Adrian Baumeister contributed to the collection of planktonic foraminifera during the cruise.

Ana Luiza S. Albuquerque headed the Paleoceano project that funded the Douglas Lessa's internship at MARUM (Bremen University).

Michal Kucera designed the research and contributed to the taxonomy and statistical analyses of planktonic foraminifera.

30 All authors contributed to the writing of the manuscript.

**9. Acknowledgements**

We thank all crew members and scientists for their help in the collection of planktonic foraminifera during the M124 cruise onboard the FS Meteor. This study was financed in part by the Coordenação de Aperfeiçoamento de Pessoal de Nível Superior - CAPES/Brazil - Finance Code 001. Douglas V. O. Lessa acknowledges financial support from CAPES/Paleoceano project (23038.001417/2914-71<del>),</del>) and the CAPES Programas Estratégicos - DRI (PE 99999.000042/2017-00<del>), and</del>). Further support was provided by the Deutsche Forschungsgemeinschaft (DFG) through Germany's Excellence Strategy (EXC-2077, grant no 390741603) to the Cluster of Excellence "The Ocean Floor – Earth's Uncharted Interface". We thank Antje Voeker and Ralf Schiebel for their important contributions to manuscript improvement.

| Oksanen, J., Blancher, F.G., Friendly, M., Kindt, R., Legendre, R., McGlinn, D., Minchin, P.R., Solymos, P., |
|--------------------------------------------------------------------------------------------------------------|
| Stevens, M.H.H., Szoecs, E., and Wagner, H.: vegan: Community Ecology Package. R package version 2.5-6.      |
| 2019, https://CRAN.R-project.org/package=vegan.                                                              |

Ortiz, J. D., Mix, A. C., and Collier, R. W.: Environmental control of living symbiotic and asymbiotic
 foraminifera of the California Current. Paleoceanography, 10(6), 987-1009, 1995.

Ottens, J. J.: April and August Northeast Atlantic surface water masses reflected in planktic foraminifera, Neth. J. Sea Res., 28, 261–283, 1992.

Peeters, F. J., Acheson, R., Brummer, G. J. A., De Ruijter, W. P., Schneider, R. R., Ganssen, G. M., Ufkes, E., and Kroon, D.: Vigorous exchange between the Indian and Atlantic oceans at the end of the past five glacial periods. Nature, 430(7000), 661-665, 2004.

Peterson, R. G., and Stramma, L.: Upper-level circulation in the South Atlantic Ocean, Progress in oceanography, 26(1), 1-73, 1991.

10

Provost, C., Escoffier, C., Maamaatuaiahutapu, K., Kartavtseff, A., and Garçon, V.: Subtropical mode waters in the South Atlantic Ocean. Journal of Geophysical Research: Oceans, 104(C9), 21033-21049, 1999.

15 R Core Team.: R: A language and environment for statistical computing. R Foundation for Statistical Computing, Vienna, Austria. 2019, URL https://www.R-project.org/.

Rebotim, A., Voelker, A. H. L., Jonkers, L., Waniek, J. J., Meggers, H., Schiebel, R., Fraile, I., Schulz, M., and Kucera, M.: Factors controlling the depth habitat of planktonic foraminifera in the subtropical eastern North Atlantic, Biogeosciences, 14, 827-859, 2017.

20 Schiebel, R., and Hemleben, C. (EdsAuthors): Planktic Foraminifers in the Modern Ocean, Springer-Verlag, Berlin Heidelberg, 366 p., 2017.

Schiebel, R., and Hemleben, C.: Interannual variability of planktic foraminiferal populations and test flux in the eastern North Atlantic Ocean (JGOFS). Deep Sea Research Part II: Topical Studies in Oceanography, 47(9-11), 1809-1852, 2000.

25 Schiebel, R., Schmuker, B., Alves, M., and Hemleben, C.: Tracking the Recent and late Pleistocene Azores front by the distribution of planktic foraminifers. Journal of Marine Systems, 37(1-3), 213-227, 2002.

Schiebel, R., Waniek, J., Bork, M., and Hemleben, C.: Planktic foraminiferal production stimulated by chlorophyll redistribution and entrainment of nutrients. Deep Sea Research Part I: Oceanographic Research Papers, 48(3), 721-740, 2001.

30 Siccha, M., Trommer, G., Schulz, H., Hemleben, C., and Kucera, M.: Factors controlling the distribution of planktonic foraminifera in the Red Sea and implications for the development of transfer functions. Marine Micropaleontology, 72(3-4), 146-156, 2009.

Sousa, S. H. M., de Godoi, S. S., Amaral, P. G. C., Vicente, T. M., Martins, M. V. A., Sorano, M. R. G. S., Gaeta, S. A., Passos, R. F., and Mahiques, M. M.: Distribution of living planktonic foraminifera in relation to oceanic processes on the southeastern continental Brazilian margin (23 S–25 S and 40 W–44 W), Continental Shelf Research, 89, 76-87, 2014.

Spindler, M., Hemleben, C., Bayer, U., Bé, A. W. H., and Anderson, O. R.: Lunar periodicity of reproduction in the planktonic foraminifer *Hastigerina pelagica*. Marine Ecology Progress Series, 61-64, 1979.

Steinke, S., Chiu, H. Y., Yu, P. S., Shen, C. C., Löwemark, L., Mii, H. S., and Chen, M. T.: Mg/Ca ratios of two
 Globigerinoides ruber (white) morphotypes: Implications for reconstructing past tropical/subtropical surface water conditions. Geochemistry, Geophysics, Geosystems, 6(11), 2005.

Storz, D., Schulz, H., Waniek, J. J., Schulz-Bull, D. E., and Kučera, M.: Seasonal and interannual variability of the planktic foraminiferal flux in the vicinity of the Azores Current, Deep Sea Research Part I: Oceanographic Research Papers, 56(1), 107-124, 2009.

45 Stramma, L., and England, M.: On the water masses and mean circulation of the South Atlantic Ocean, Journal of Geophysical Research: Oceans, 104(C9), 20863-20883, 1999. Takagi, H., Kimoto, K., Fujiki, T., Saito, H., Schmidt, C., Kucera, M., and Moriya, K.: Characterizing photosymbiosis in modern planktonic foraminifera. Biogeosciences, 16(17), (2019).

- 5 Telford, R. J., and Birks, H. J. B.: The secret assumption of transfer functions: problems with spatial autocorrelation in evaluating model performance. Quaternary Science Reviews, 24(20-21), 2173-2179. 2005.
- ! Telford, R. J., Li, C., and Kucera, M.: Mismatch between the depth habitat of planktonic foraminifera and the calibration depth of SST transfer functions may bias reconstructions, Climate of the Past, 9(2), 859-870, 2013.
- Ufkes, E., Jansen, J. H., and Brummer, A.: Living planktonic foraminifera in the eastern South Atlantic during spring: indicators of water masses, upwelling and the Congo (Zaire) River plume, Marine Micropaleontology, 33, 27-53, 1998.
- Wang, L.: Isotopic signals in two morphotypes of *Globigerinoides ruber* (white) from the South China Sea: implications for monsoon climate change during the last glacial cycle. Palaeogeography, Palaeoclimatology, Palaeoecology, 161(3-4), 381-394, 2000.
- 5 Weiner, A., Aurahs, R., Kurasawa, A., Kitazato, H., and Kucera, M.: Vertical niche partitioning between cryptic sibling species of a cosmopolitan marine planktonic protest, Molecular Ecology, 21(16), 4063-4073, 2012.
- Venancio, I. M., Franco, D., Belem, A. L., Mulitza, S., Siccha, M., Albuquerque, A. L. S., Schulz, M., and Kucera, M.: Planktonic foraminifera shell fluxes from a weekly resolved sediment trap record in the
   southwestern Atlantic: Evidence for synchronized reproduction. Marine Micropaleontology, 125, 25-35, 2016.

Žarić, S., Donner, B., Fischer, G., Mulitza, S., & Wefer, G.: Sensitivity of planktic foraminifera to sea surface temperature and export production as derived from sediment trap data, Marine Micropaleontology, 55(1-2), 75-105, 2005.

20 105,